# Decomposing cortical activity through neuronal tracing connectome-eigenmodes in marmosets

Jie Xia[1,2], Cirong Liu [3], Jiao Li[1,2], Yao Meng [1,2], Siqi Yang[4], Huafu Chen [1,2] ✉ & Wei Liao [1,2] ✉

Deciphering the complex relationship between neuroanatomical connections and functional activity in primate brains remains a daunting task, especially regarding the influence of monosynaptic connectivity on cortical activity. Here, we investigate the anatomical-functional relationship and decompose the neuronal-tracing connectome of marmoset brains into a series of eigenmodes using graph signal processing. These cellular connectome eigenmodes effectively constrain the cortical activity derived from resting-state functional MRI, and uncover a patterned cellular-functional decoupling. This pattern reveals a spatial gradient from coupled dorsal-posterior to decoupled ventral-anterior cortices, and recapitulates micro-structural profiles and macro-scale hierarchical cortical organization. Notably, these marmoset-derived eigenmodes may facilitate the inference of spontaneous cortical activity and functional connectivity of homologous areas in humans, highlighting the potential generalizing of the connectomic constraints across species. Collectively, our findings illuminate how neuronal-tracing connectome eigenmodes constrain cortical activity and improve our understanding of the brain's anatomical-functional relationship.

Human and non-human primate brains are structurally and functionally organized at multiple scales[1,2]. At the macroscale, brain regions are organized into interconnected networks, as revealed by diffusion magnetic resonance imaging (dMRI) tractography[3]. At the mesoscale, neuroanatomical tracers reveal neuronal populations are intricately linked via precise synaptic connections[4]. Understanding how the variation in structural organization underlies the brain's functional profile is a fundamental goal of neuroscience[5–7].

Structural and functional connectivity are tethered[8–11]. High-order interactions among neural populations may give rise to complicated and imperfect correspondence between structure and function[7,12]. Graph signal processing (GSP) innovatively paves the way for probing high-order structural-functional interactions[13–15]. GSP exploits the topographic organization of brain structures to characterize brain activity, presenting a concise and interpretable framework[14,16]. Previous studies have proposed that connectome eigenmodes[16–18] derived from dMRI-based structural connectivity (SC) constraint spatiotemporal patterns of neural dynamics in humans[17,19–21]. Functional activity is decomposed into structurally informed components, representing varying degrees of activity deviation from the underlying anatomical architecture[16,22]. Additionally, the structural-decoupling index is proposed to quantify the regional coupling strength between structure and function[12], revealing a meaningful sensorimotor-to-association gradient[23,24] across the neocortex in humans. The

[1]The Clinical Hospital of Chengdu Brain Science Institute, School of Life Science and Technology, University of Electronic Science and Technology of China, Chengdu 611731, P.R. China. [2]MOE Key Lab for Neuroinformation, High-Field Magnetic Resonance Brain Imaging Key Laboratory of Sichuan Province, University of Electronic Science and Technology of China, Chengdu 611731, P.R. China. [3]Institute of Neuroscience, CAS Key Laboratory of Primate Neurobiology, Center for Excellence in Brain Science and Intelligence Technology, Chinese Academy of Sciences, Shanghai 200031, P.R. China. [4]School of Cybersecurity, Chengdu University of Information Technology, Chengdu 610225, P.R. China. ✉e-mail: chenhf@uestc.edu.cn; weiliao.wl@gmail.com

structure-function coupling based on GSP has promising directions in task decoding[25], individual fingerprinting[25], and brain dysfunction[26–28]. However, given the inherent limitations of dMRI tractography in examining anatomical connections[29,30], the degree to which structural connectivity constrains functional activity remains largely uncharted.

Neuroanatomical tract-tracing techniques stand unrivaled in directly detecting monosynaptic connections[31]. These monosynaptic transneuronal tracers prove indispensable for mapping long-distance connections[32–34] and the mesoscale cellular connectome (CC)[35–39]. Neural tracing elucidates information about directionality of anatomical projections in mice[40], monkeys[32], and marmosets (*Callithrix jacchus*)[35,41]. Notably, the marmosets have the most comprehensive neuronal tract-tracing data among primates[35,41] and have close homology and similar cortical architecture to the humans[42,43], providing an invaluable resource for investigating anatomical constraints on functional activity.

In this study, we aimed to decipher the complex relationship between neuroanatomical connections and functional activity using neuronal-tracing data and blood oxygen level-dependent functional MRI (BOLD-fMRI) during resting-state from awake marmosets. First, we considered cortical activity as graph signals residing in the domain of tracer-based CC and decomposed it into low- and high-frequency components using directed GSP (dGSP)[44] via CC eigenmodes. Second, we quantified cellular-functional decoupling (CFD) in individual cortical areas to reflect the degree of local (de)coupling between cortical activity and monosynaptic connections. Third, we analyzed the spatial correspondence between regional CFD and microscale and macroscale attributes. Finally, we explored the potential of marmoset CC eigenmodes to capture intrinsic brain activity and functional connectivity in humans, determining whether connectional information from marmosets could be applied to the human brain.

## Results

We decomposed tract-tracing CC into spatial eigenmodes and projected them onto BOLD-fMRI in marmosets (see Fig. 1 for study overview). We used retrograde tracing data in adult marmosets ($N = 52$; 21 females and 31 males; 1.4–4.6 years) publicly available from the Marmoset Brain Architecture Project (https://www.marmosetbrain.org/)[35] and BOLD-fMRI data in awake marmosets at rest from two independents datasets (total $N = 19$; 1 female and 18 males; 2–9 years) (https://marmosetbrainmapping.org/data)[34,45].

### Cellular connectome (CC) eigenmodes

Utilizing the Paxinos atlas[46,47] of the left hemisphere (Supplementary Fig. 1; Supplementary Table 1) to extract the mean extrinsic fraction of labeled neurons (FLNe) in marmosets, we constructed a weighted and directed CC matrix (55 source areas × 55 target areas)[35] (Supplementary Fig. 2). We then established a symmetric normalized Laplacian matrix for this directed graph[48] (Supplementary Fig. 3; Supplementary Algorithm 1), which captures the projection directionality and edge density leveraging the random walk operator[49,50]. The CC eigenmodes, also known as eigenvectors, were computed by the eigendecomposition of the normalized directed graph Laplacian (Fig. 1a). Note that the sign (polarity) of CC eigenmode is arbitrary. The CC eigenmodes measured smoothly varying patterns across the marmoset's cortices between positive and negative polarities[17,21,51]. The smoothness of each eigenmode could be conceptualized in terms of graph frequency[49]. The irregularity and localization of CC eigenmode patterns increased with increasing eigenvalue (or frequency) (Supplementary Fig. 4). Specifically, the first CC eigenmode was uniformly distributed throughout the brain. The second one reflected a dorsal-ventral dimension. The third CC eigenmode showed the dimension between the anterior cingulate and other brain areas. The fourth CC eigenmode represented a gradient axis from the sensorimotor cortex

to the visual cortex, resembling the marmosets' principal structural gradient[52](Fig. 1b).

### CC eigenmodes constrain on marmosets' cortical activity

We assessed the extent to which CC eigenmodes may explain brain activity observed in BOLD-fMRI data from marmosets. We began by decomposing cortical activity into a combination of orthogonal CC eigenmodes (Fig. 1d). We next tested the accuracy of CC eigenmodes in capturing spontaneous cortical activity using this decomposition. We found that increasing CC eigenmodes improved the reconstruction accuracy of marmosets' cortical activity concentration (Fig. 2a). Using all CC eigenmodes ($N = 55$), the cortical activity could be entirely reconstructed. The first 20 CC eigenmodes achieved 83% reconstruction accuracy of cortical activity concentration, which was higher than utilizing both rewired CC graph[53,54] ($p_{rewired} < 0.001$, false discovery rate (FDR)-corrected) and the Moran spectral randomization (MSR) surrogate cortical activity[55–57] ($p_{Moran} < 0.001$, FDR-corrected). Similarly, the first 20 CC eigenmodes achieved 88% reconstruction accuracy of functional connectivity (FC) in marmosets (Supplementary Fig. 5). These findings suggest that CC eigenmodes canonically serve as a foundation for a compact description of spontaneous cortical activity and FC.

We next compared the reconstruction accuracy of CC eigenmodes against dMRI-based SC eigenmodes (Supplementary Fig. 6). Despite certain similarities, the spatial patterns of CC eigenmodes were distinct from SC eigenmodes (Supplementary Fig. 7). A direct comparison of the reconstruction accuracy of two distinct basis sets revealed that CC eigenmodes numerically outperform the SC eigenmodes in capturing cortical activity concentration and FC. The CC eigenmodes could provide a more compact description of cortical activity patterns than the SC eigenmodes.

We further examined how CC eigenmodes constrain cortical activity. To this end, we separated the CC eigenmodes into low- (the first $C$ eigenvectors) and high-frequency (the last $N-C$ eigenvectors). Here, we set $C = 12$ as the median-frequency, and provided the sensitivity analyses for filter cut-off selection (Supplementary Fig. 8; Supplementary Section 6). Then, low- and high-frequency eigenmodes were used to spatially filter the BOLD-fMRI amplitudes for each time point, resulting in low- and high-frequency components that characterized to what extent BOLD-fMRI fluctuations were strongly or weakly constrained by the underlying monosynaptic connections[12,16,22]. The significance of the cortical class-level activity concentrations was examined statistically using a permutation test (see Statistical analysis).

Cortical activity constrained by low-frequency eigenmodes was concentrated within dorsal-posterior cortices such as the visual cortex (VC), somatosensory cortex (SS), posterior parietal cortex (PPC), and posterior cingulate, medial, and retrosplenial cortices (PCC) (all $p_{SR} < 0.05$, FDR-corrected; Fig. 2c). The high-frequency components were concentrated in ventral-anterior (frontopolar-temporal) cortices, including the medial prefrontal cortex (mPFC), dorsolateral prefrontal cortex (dlPFC), ventrolateral prefrontal cortex (vlPFC), orbitofrontal cortex (OFC), lateral inferior temporal cortex (LIT), and auditory cortex (AU) (all $p_{SR} < 0.05$, FDR-corrected; Fig. 2d). In summary, the patterns of CC eigenmodes constrain cortical activity were circumscribed by specific brain systems.

### Regional cellular-functional decoupling

We further investigated the cellular-functional relationships of the CC and cortical activity in marmosets. We quantified the binary logarithm form of the ratio between the $L_2$-norm of high-frequency versus low-frequency components over time points[12] to represent CFD. The CFD was used to assess the degree of local (de)coupling between cortical activity and underlying monosynaptic connections. Lower CFD implied a strong coupling of cortical activity to the neural connections,

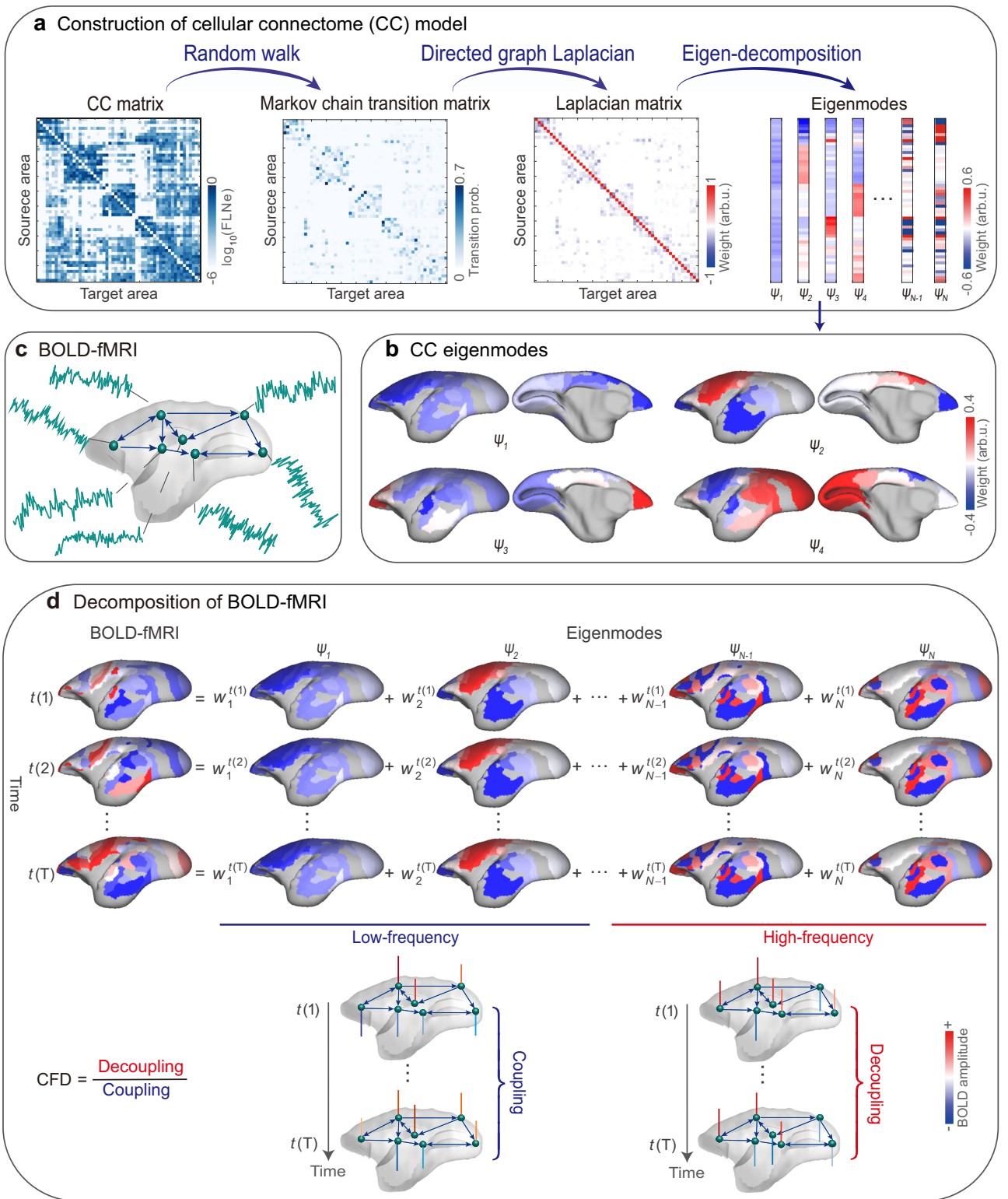

**Fig. 1 | Schematic method overview. a** Workflow for constructing the mesoscale cellular connectome (CC) model. **b** The first four CC eigenmodes ($\psi_1$–$\psi_4$, in ascending order by eigenvalues) were projected onto the marmoset brain surface. Colors visualized arbitrary units (arb. u.), i.e., the weights in eigenvectors. **c** BOLD-fMRI in awake marmosets. **d** Decomposition of BOLD-fMRI. The fMRI data at each time point ($t(i)$) was estimated as the contribution ($w_k^{t(i)}$) of each CC eigenmode ($\psi_k$). Cortical activity was then decomposed into low-frequency components (coupled to CC, i.e., heavily interconnected nodes tend to display similar activity to one another) and high-frequency components (decoupled from CC, i.e., nodes exhibit various activities even if they are heavily connected). Cellular-functional decoupling (CFD) was the ratio between the $L_2$-norm of decoupled and coupled components over time points.

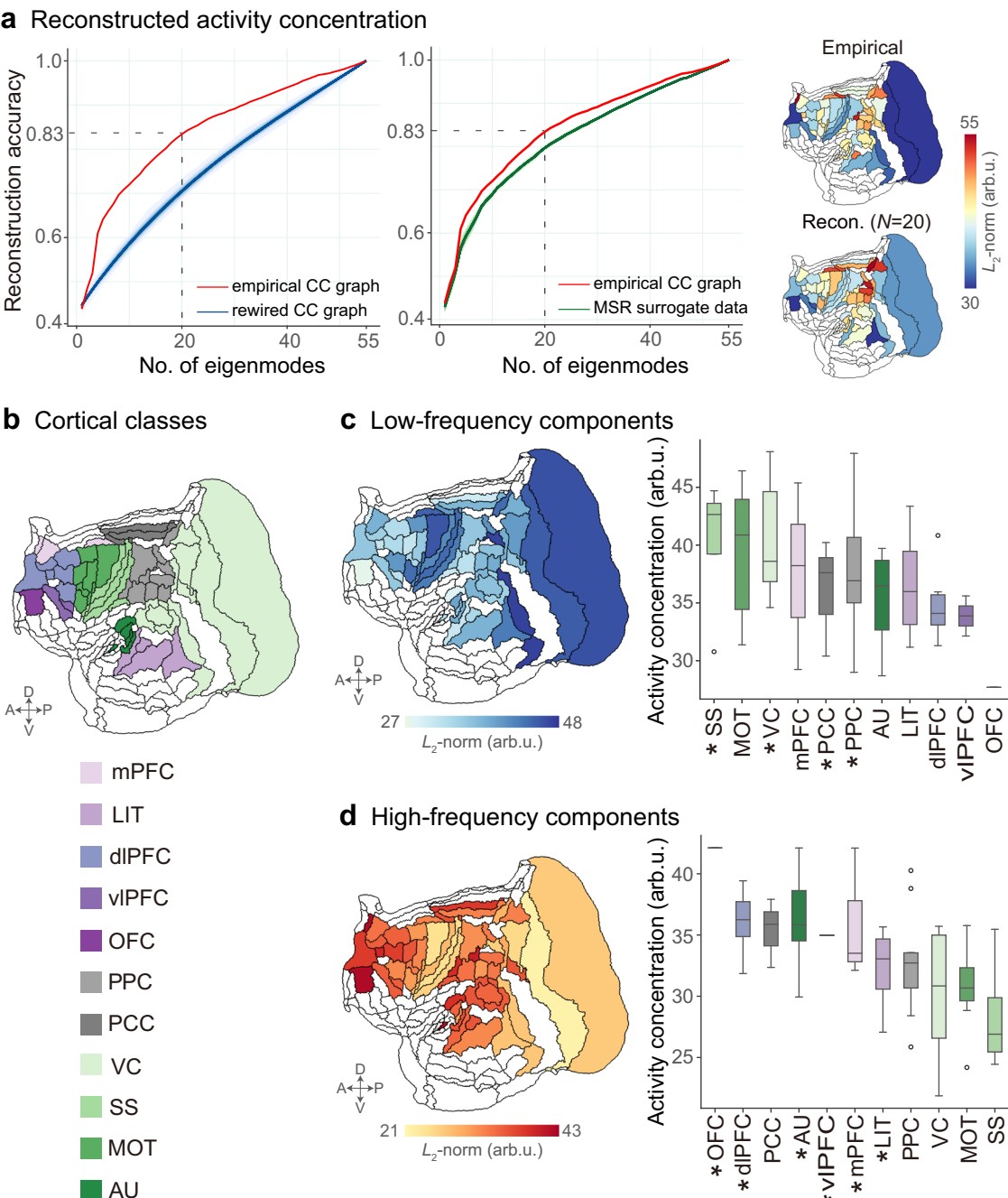

**Fig. 2 | Cellular connectome (CC) eigenmodes constrain cortical activity in marmosets. a** Reconstruction accuracy was quantified as the ratio between empirical and reconstructed cortical activity concentration ($L_2$-norm across time points). The solid line indicates the reconstruction accuracy of the empirical CC graph. The shading lines indicate the reconstruction accuracy using eigenmodes derived from rewired CC graphs (1000 repetitions). The shading lines indicate the reconstruction accuracy using empirical CC eigenmodes to reconstruct the Moran spectral randomization (MSR) surrogate cortical activity (1000 repetitions). The shading indicates the 95th percentile interval of the null distributions. **b** Marmoset cortical classes were parceled according to the Paxinos atlas[46,47]. **c, d** The spatial patterns of low- and high-frequency components. The low- and high-frequency components from each of the eleven cortical classes were presented as box plots ordered by the median values. Box plots represent the 25th (lower), 50th (median), and 75th (upper) percentiles; the whiskers represent the non-outlier endpoints of the distribution; and the circles represent outliers. The sample size for each boxplot was the number of brain areas in each marmoset cortical class and summarized in Supplementary Table 1 ($n_{AU} = 5$, $n_{dlPFC} = 6$, $n_{LIT} = 4$, $n_{mPFC} = 3$, $n_{MOT} = 7$, $n_{OFC} = 1$, $n_{PCC} = 3$, $n_{PPC} = 10$, $n_{SS} = 4$, $n_{vlPFC} = 2$, $n_{VC} = 10$). Asterisks denote statistically significant activity concentration in each cortical class compared to the null distributions generated from graph spectral randomization (SR) (1000 repetitions, *$p_{SR} < 0.05$, $p_{SR}(SS) = 0.0037$, $p_{SR}(VC) = 0.0001$, $p_{SR}(PPC) < 0.0083$, $p_{SR}(PCC) = 0.0037$, $p_{SR}(OFC) = 0.0001$, $p_{SR}(dlPFC) = 0.0001$, $p_{SR}(AU) = 0.0001$, and $p_{SR}(vlPFC) = 0.0001$, $p_{SR}(mPFC) = 0.0001$, and $p_{SR}(LIT) = 0.0018$, one-sided, FDR-corrected). AU auditory cortex, dlPFC dorsolateral prefrontal cortex, LIT lateral and inferior temporal cortex, mPFC medial prefrontal cortex, MOT motor and premotor cortex, OFC orbitofrontal cortex, PCC posterior cingulate, medial and retrosplenial cortex, PPC posterior parietal cortex, SS somatosensory cortex, vlPFC ventrolateral prefrontal cortex, VC visual cortex.

whereas higher CFD indicated the reverse. The spatial pattern of CFD was regionally heterogeneous (Fig. 3a, left). Regional CFD revealed a gradient organization that ranged from coupling areas in the dorsal-posterior cortices (visual, sensorimotor, motor, and premotor cortex (MOT), and PPC) to decoupling areas in the ventral-anterior cortices (the temporal, prefrontal, and orbitofrontal cortex) (Fig. 3a, right).

Furthermore, we provided the sensitivity analyses for filter cut-off selection for patterned CFD (Supplementary Fig. 9). The permutation testing approach was used to localize brain areas where the CFD significantly differed from the graph spectral randomization surrogate activity (Supplementary Fig. 10). Areas exhibiting coupling (Fig. 3a, middle), which significantly deviated from null permutations, were primarily found in the VC (V1, V2, V3A, V4, V6, A19DI), SS (A1-2, A3a, A3b), MOT (A6Va, A8C, A4ab), and PPC (PE, PG, LIP). Areas with significant decoupling (Fig. 3a, middle) were primarily located in the OFC (A11), dlPFC (A10, A9, A8b, A46D), vlPFC (A45), and mPFC (A32V). Collectively, a gradual divergence between the CC and cortical activity in marmosets was observed, transitioning from the dorsal-posterior to the ventral-anterior cortices.

We replicated the regional CFD pattern using an independent dataset from the Institute of Neuroscience (ION) cohort[34] (Fig. 3b). We then conducted a spatial correlation of the CFD pattern between the NIH and ION site datasets. The results showed a statistically significant positive correlation ($\rho = 0.83$, $p_{SMASH} = 0.0001$; Supplementary Fig. 11). Furthermore, the Dice coefficient was 0.73 and 0.70 for significantly decoupled and coupled areas, respectively. The differences in the scanner magnetic fields and the scanning parameters of the two datasets may drive some discrepancies. As a result, these findings confirmed the reproducibility of the CFD patterns.

## Cellular-functional decoupling follows microscale and macroscale hierarchies

We then examined the spatial relationships between CFD and microscale and macroscale spatial profiles. At the microscale levels (Fig. 4a, b), regional CFD was strongly negatively correlated with myelin content[47] ($\rho = 0.40$, $p_{SMASH} = 0.007$) and neuronal counts[58] ($\rho = 0.43$, $p_{SMASH} = 0.008$). The prefrontal, lateral parietal, lateral temporal, and medial parietal cortices, which are weakly myelinated and involved in higher cognitive and affective activities[59], showed strong CFD. Additionally, a primary axis connecting the posterior and medial areas to the anterior and lateral areas followed the density of neurons[58]. Consequently, the spatial pattern of CFD, ranging from coupling to decoupling, was inversely related to the transmodal-to-unimodal cortical hierarchy at the microscale level.

At the macroscale level (Fig. 4c), regional CFD positively correlated with the second functional gradient[52] ($\rho = 0.48$, $p_{SMASH} = 0.004$). The CFD's spatial organization, represented as coupling to decoupling, aligned well with the cortical hierarchy from the unimodal to transmodal cortex. Collectively, the patterned cellular-functional relationship resembled the cortical hierarchical axis on the micro- and macroscale levels.

## Generalizability of marmoset CC eigenmodes to the human brain

We next explored whether human cortical activity and FC could be inferred from marmoset-derived CC eigenmodes. We chose 11 possible homologous areas[60] (Fig. 5a; Supplementary Table 2) from the Paxinos marmoset parcellation scheme[46,47] and the HCP-MMP1.0 human cortical atlas[61], taking into account the differences between humans and

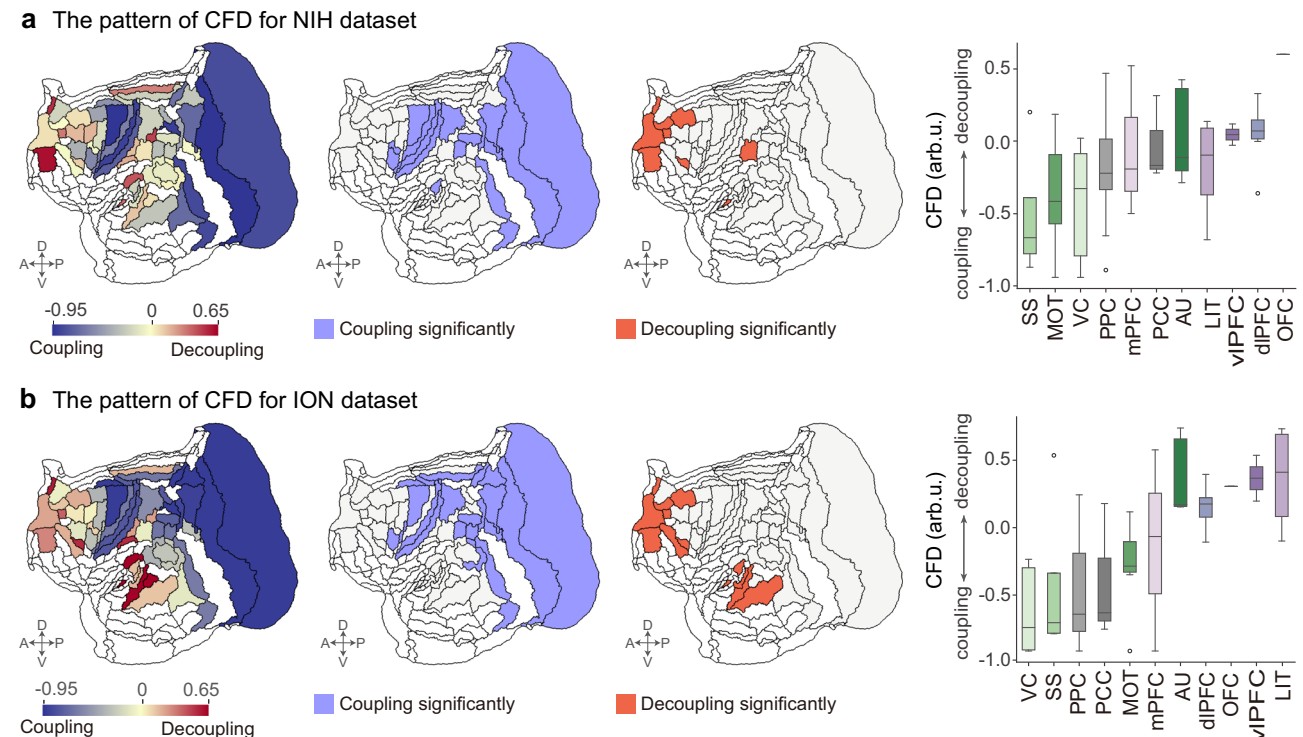

**a** The pattern of CFD for NIH dataset

-0.95   0   0.65
Coupling      Decoupling

Coupling significantly

Decoupling significantly

**b** The pattern of CFD for ION dataset

-0.95   0   0.65
Coupling      Decoupling

Coupling significantly

Decoupling significantly

**Fig. 3 | Regional cellular-functional decoupling (CFD) in two independent datasets. a** (Left) The pattern of CFD in the National Institutes of Health (NIH) cohort was plotted. (Middle) The statistically significant areas were grouped into cellular-functional coupling and decoupling patterns (Binomial test with a significance level $\alpha = 0.05$, two-sided, corrected for multiple comparisons across 55 cortical areas). (Right) Box plots represent the CFD values from the eleven classes ordered by the median values. **b** The pattern of CFD and the statistically significant areas in the Institute of Neuroscience (ION) cohort. Box plots represent the 25th (lower), 50th (median), and 75th (upper) percentiles; the whiskers represent the distribution's endpoints; and the circles represent outliers. $n_{AU} = 5$, $n_{dlPFC} = 6$, $n_{LIT} = 4$, $n_{mPFC} = 3$, $n_{MOT} = 7$, $n_{OFC} = 1$, $n_{PCC} = 3$, $n_{PPC} = 10$, $n_{SS} = 4$, $n_{vlPFC} = 2$, $n_{VC} = 10$. AU auditory cortex, dlPFC dorsolateral prefrontal cortex, LIT lateral and inferior temporal cortex, mPFC medial prefrontal cortex, MOT motor and premotor cortex, OFC orbitofrontal cortex, PCC posterior cingulate, medial and retrosplenial cortex, PPC posterior parietal cortex, SS somatosensory cortex, vlPFC ventrolateral prefrontal cortex, VC visual cortex.

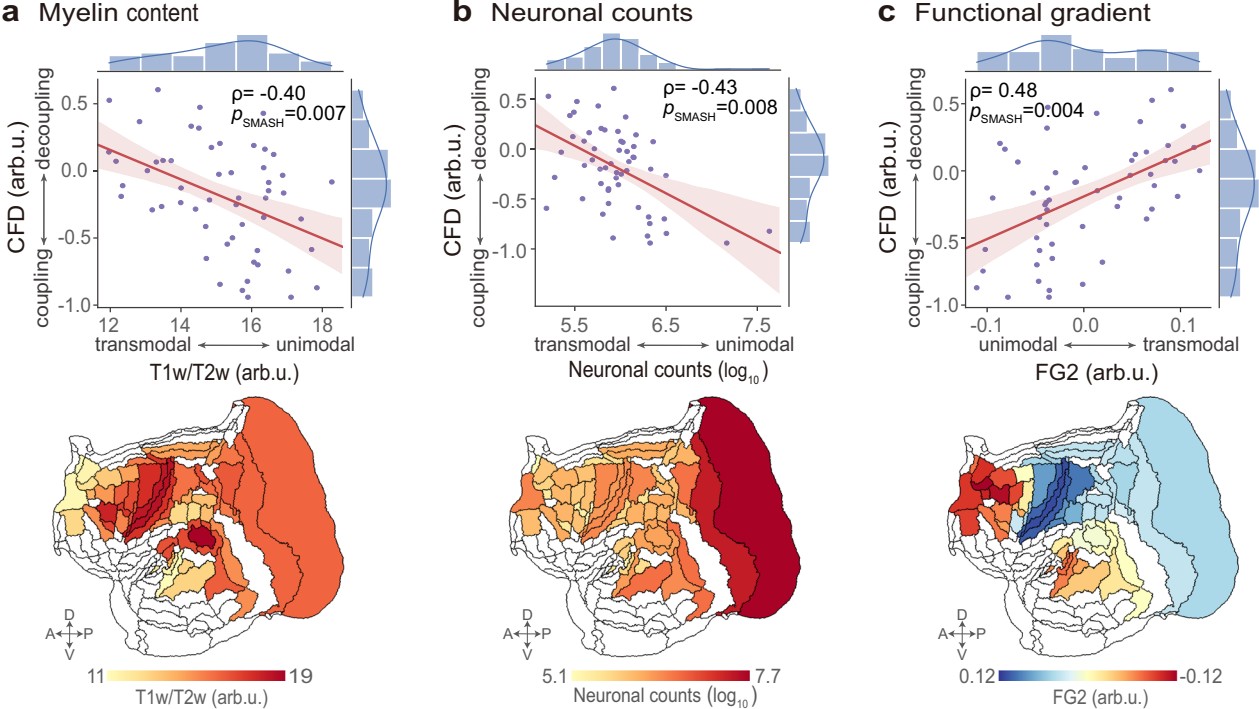

**Fig. 4 | Marmoset cellular-functional decoupling (CFD) is related to micro-structures and macroscale hierarchical organization. a, b** The pattern of regional CFD (n = 55 brain areas) negatively correlates with microstructures, including marmoset myelin content (T1w/T2w ratio) and neuronal counts. **c** Patterned CFD positively correlates with the second functional gradient (FG2). The regression lines are shown for both relationships. Shaded bands represent the 95% confidence intervals; and histograms correspond to each variable. The significance ($p_{SMASH}$) of Spearman's correlation coefficients (ρ) is evaluated using spatial autocorrelation-preserving surrogate brain maps (1000 repetitions, two-sided).

marmosets' cerebral cortices. We then constructed a CC matrix using these 11 homologous markers. We used a basis set generated from the CC matrix of homologous areas to reconstruct localized brain activity patterns in marmosets (Supplementary Fig. 12; Supplementary Section 7), implying that localized CC eigenmodes potentially reflect activity patterns in marmosets.

We found that increasing CC eigenmodes of homologous areas improved the reconstruction accuracy of brain activity patterns in humans (Fig. 5b). The first five CC eigenmodes achieved 90% and 86% reconstruction accuracy of brain activity concentration and FC, respectively. Reconstructing human brain activity concentration (Fig. 5b) and FC (Fig. 5c) using homologous CC eigenmodes was more accurate than rewired CC connectomes ($p_{rewired}$ < 0.001, FDR-corrected) and randomly selected non-homologous areas' cortical activity ($p_{perm}$ < 0.001, FDR-corrected). These findings suggest that CC eigenmodes derived from marmosets may help estimate human spontaneous brain activity and FC of homologous areas.

Additionally, we calculated the relationship between the marmoset's CFD and the three canonical hierarchies in humans described by myelin content, allometric scale, and cortical gene expression across 11 homologous areas to determine the correspondence of hierarchical organization across species between humans and marmosets. We discovered that the CFD pattern in homologous areas of marmosets matched the myelin content[62] (ρ = −0.69, $p_{SMASH}$ = 0.02; Fig. 5d), allometric scaling[63] (ρ = 0.81, $p_{SMASH}$ = 0.002; Fig. 5e), and the first principal component of gene expression[64] (ρ = 0.88, $p_{SMASH}$ = 0.0001; Fig. 5f) in humans. These findings showed a dimension with comparable hierarchical structure in marmosets and humans, suggesting that this dimension may be phylogenetically preserved[24].

## Discussion

We quantified how the marmosets' intrinsic cortical activity derived from BOLD-fMRI data was bounded by CC-eigenmodes from

retrograde tracing. The cellular-functional relationships were gradually decoupled from the dorsal-posterior to ventral-anterior cortices, following microscale and macroscale hierarchies. Furthermore, the marmoset's CC eigenmodes would capture human cortical activity and FC in homologous areas, underlining the potential of generalizing the connectomic constraints across species.

Extended previous structural-functional coupling focused on a single summary process[8,10,11], we considered these couplings contribute to multiple repertoires through eigenmode decomposition[12,14]. Eigenmodes provide a powerful framework for connecting brain anatomy with the spatiotemporal patterns of neural dynamics. Previous studies have extensively utilized eigenmode approaches to understand human brain function[15–17,19,65]. The SC eigenmodes obtained from dMRI-based undirected networks capture local gray matter and white matter fiber connections, which serve as the foundation for human functional networks[16–18]. Furthermore, geometric eigenmodes derived from the brain's geometry (e.g., its shape) capture local spatial relations, representing the underlying anatomical restrictions on human brain function[15]. Our work with marmosets used a similar approach, but the fundamental anatomical features differed. The neuroanatomical tracing connectome represents an asymmetric (directed) network. Graph signal models based on asymmetric network operators may be better for signal and information processing on directed projections[44]. The CC eigenmodes produced from Laplacian's normalized directed graph may capture the projection directionality and edge density, emphasizing the importance of projection directionality information. Different methods of generating anatomical connections and deriving eigenmodes may lead to different spatial patterns of CC and SC eigenmodes. Importantly, CC eigenmodes could be the fundamental building blocks for reconstructing cortical activity patterns with different frequency oscillations, facilitating the establishment of relationships between temporal fluctuations and underlying anatomy in the marmoset brain.

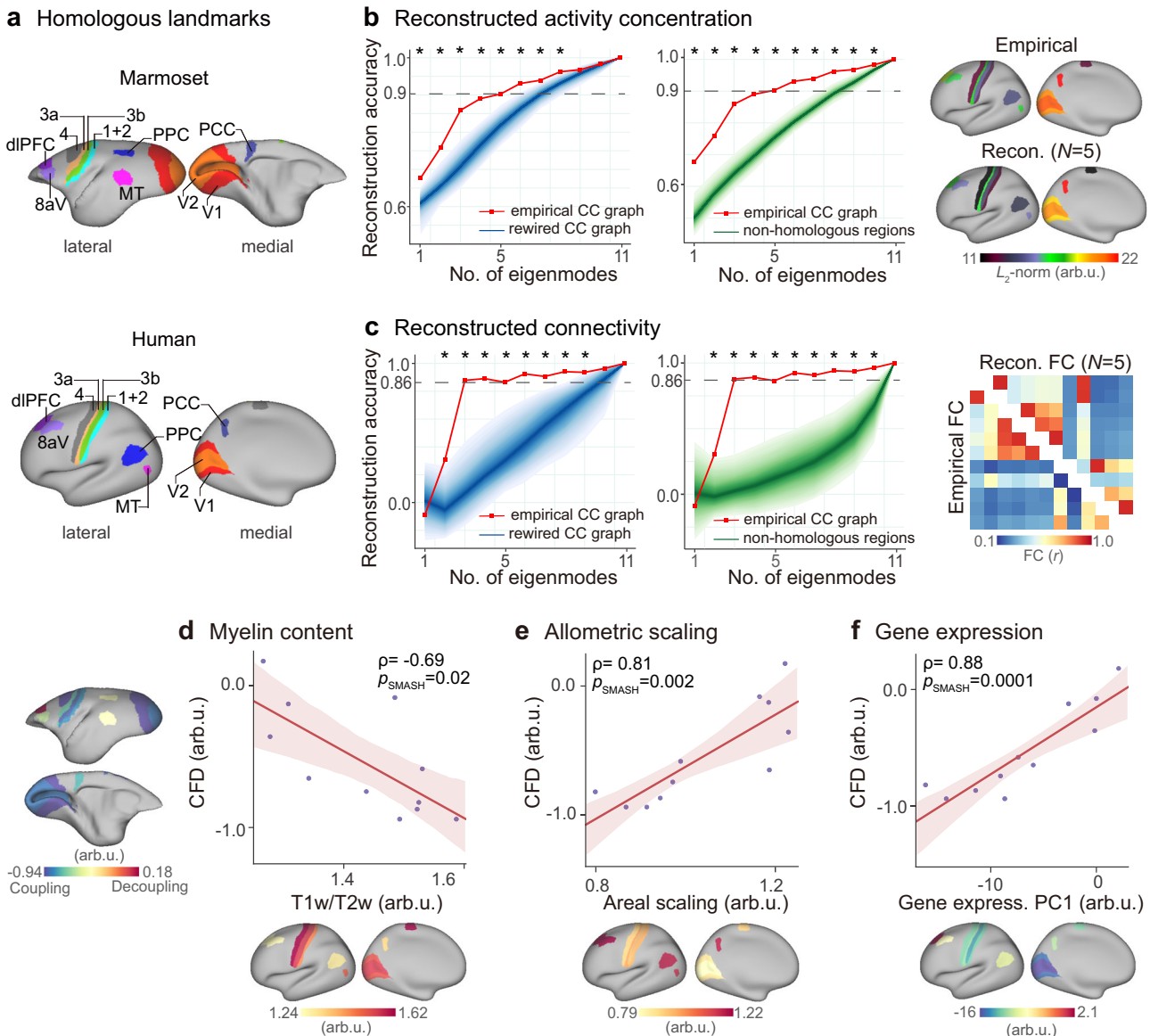

**Fig. 5 | Generalizability of marmoset-derived eigenmodes to cortical activity patterns in humans. a** Humans and marmosets share 11 common homologous landmarks. **b** Reconstruction accuracy of cortical activity concentration ($L_2$-norm across time points) in humans across 11 homologous areas. **c** FC reconstruction accuracy in humans across 55 edges ($C_{11}^2 = 55$). The observed reconstruction accuracy (solid lines) was compared to the accuracy obtained from rewired CC connectomes (shading lines, 1000 repetitions) and randomly selected non-homologous areas' cortical activity (shading lines, 1000 repetitions). The shading indicates the 95th percentile interval of the null distributions. Asterisks denote a statistically significance level at $p < 0.05$ (one-sided, FDR-corrected). The exact

$p$-values are provided in the Source data. **d, e, f** Cross-species association. The association between the CFD pattern of 11 homologous areas in marmosets and the myelin content quantified by T1w/T2w ratio (**d**), allometric scaling quantified as the relative extent of areal scaling with scaling of overall brain size (**e**), and the first principal component (PC1) of gene expression (**f**) in humans. The regression lines are shown for both relationships. Shaded bands represent the 95% confidence intervals. The significance ($p_{SMASH}$) of Spearman's correlation coefficients (ρ) is evaluated using spatial autocorrelation-preserving surrogate brain maps (1000 repetitions, two-sided).

We provided a critical viewpoint on how CC constrains functional activity. Previous studies have used correlation and regression analyses[34,37], and whole-brain computational modeling[52] to assess the local or global correspondence between FC and CC in marmosets. However, in these investigations, FC was represented by BOLD-fMRI temporal correlations measured throughout time periods, which may give a degraded depiction of brain connectivity[22,66]. In contrast, our work decomposed cortical activity into CC eigenmodes-informed components, which may represent how BOLD-fMRI amplitudes were strongly or weakly constrained by monosynaptic connections at a specific time point. Our findings showed that monosynaptic connections strongly constrained cortical activity, particularly in the dorsal-posterior cortices such as the

VC, SS, PPC, and PCC, which were engaged in sensation, perception, and action processes[67]. Furthermore, weakly constrained cortical activity was primarily concentrated within frontopolar-temporal cortices, potentially requiring more diversified functional communication relative to the underlying wiring diagram[68,69]. Notably, the activity of the posterior core regions (PPC and PCC) in the default mode network (DMN)[45] was strongly constrained by connections, whereas the activity of the anterior DMN (dlPFC)[45] was weakly restricted by connections, indicating that the DMN architecture in marmosets may have a divergent anterior-posterior axis. According to a recent cross-species study, the anterior DMN areas have weak connections and spatially irregular connection topology compared to the posterior DMN areas[60]. Consequently, the patterns of

marmoset brain activity constrained by CC eigenmodes were confined by specific brain systems.

We further introduced a CFD index to quantify the cellular-functional relationship per brain area in marmosets, enhancing our knowledge of the associations between the mesoscopic connectome and macroscale function. The approach differed from traditional pairwise correlations between anatomical and functional networks[37]. Instead, we exploited the topology of the anatomical network to inform neural activity, which could improve the statistical properties of BOLD-fMRI fluctuation[20]. We found that the spatial layout of CFD was non-uniform and appeared to vary gradually across the marmoset cortex. The regional CFD showed a macroscale gradient from coupled dorsal-posterior cortices to decoupled ventral-anterior cortices. Coupling in the dorsal-posterior cortices might suggest that monosynaptic connections directly supported functional communication. One potential explanation was that the dorsal-posterior cortices require a quick, accurate response to external and internal stimuli[12,70]. In contrast, the decoupling of ventral-anterior cortices, including the OFC (A11), dlPFC (A10, A9, A8b, A46D), vlPFC (A45), and mPFC (A32V), suggests that functional processing was not bounded by anatomical architecture, likely reflecting functional flexibility and information integration[11,24]. One recent study reported that the frontopolar A10 received projections from a broad area of the rostral temporal association cortex extending toward the temporal pole and might be a DMN candidate[68]. Area 10 is associated with higher-level planning forms, abstract reasoning, and processing multiple competing task demands[68]. The OFC, vlPFC, and mPFC are essential functions in emotional regulation and decision-making in marmosets[71]. Hence, these likely provided the interpretation that ventral-anterior cortices with high-level cognitive processing were more decoupled from structural wiring.

Cortical hierarchy is a significant organizational feature of primate cortical anatomy and function[23,24]. Our regionally heterogeneous CFD pattern was consistent with previous studies that have described similar hierarchical organization between microstructural and functional attributes in the marmoset brain[51,58,59,68]. Indeed, the spatial variations of CFD were inversely associated with micro-architectural properties, such as intracortical myelin[59] and neuronal counts[58], suggesting micro-circuit and histological underpinnings for cellular-functional interaction. Moreover, regional variations of CFD were positively correlated to macroscale functional gradients. Recent studies found that functional gradients in marmosets revealed brain network hierarchies ranging progressively outward from primary cortices to high-order multimodal association areas[34,52,68]. The functional organization was highly constrained by structural wiring in large-scale gradient aspects[52]. Here, we provided evidence that these cortical hierarchies were partially determined constraints on cortical activity imposed by monosynaptic connections. Our findings might contribute to understanding the hierarchical axis of the marmoset cortex that encompasses structural and functional variation at micro- and macro-spatial scales.

Our study was crucial to facilitate the translation of ground truth connectional knowledge from marmoset to human. Non-human primate neuroimaging integrated monosynaptic connections and neuroanatomical tracing to overcome the constraints of human neuroimaging[34] and could advance our understanding of the evolution of the brain[33,59]. Human neuroanatomical connections were derived primarily from homology studies in non-human experimental models[72]. The identification of probable homologous landmarks has been used to make cross-species comparisons[52,60,73]. Our results showed that marmoset's CC eigenmodes may reflect human spontaneous cortical activity in homologous areas, which were not driven by general mathematical properties of basis set expansions but could be derived by some biologically relevant evolutionary process. These findings suggested that CC may be used to restrict brain function

across species. In addition, the CFD pattern of homologous areas in marmosets recapitulated three classical hierarchies in humans described by intracortical myelin[62], allometric scale[63], and brain-wide gene expression[64]. A cross-species comparison of the topographic organization across marmosets and humans was phylogenetically conserved, which might provide critical insight into the evolution of brain organization[23,24,73]. Thus, translational studies of marmoset neuroimaging could compensate for the constraints of ground truth connections in humans, which would contribute to uncovering the correspondence between mesoscale connectome and macroscale function.

The current study has several limitations. First, retrograde tracer injections did not cover entire cortical areas, and subcortical information was missing[35]. Thus, neuronal tracing data with greater cortical coverage will facilitate mapping the complete cellular-resolution connectome and accurately quantify the relationship between the cellular-scale connectome and functional activity. Second, although retrograde tracer accurately maps cellular-level connectome by staining cell bodies, it cannot reveal local gray matter connectivity and white matter axonal fibers[17,18]. Neuronal tracing data might be integrated into cortical surface mesh and fiber tractography in future investigations[39]. Another caution is that the normalized directed graph Laplacian is suitable for strongly connected graphs, guaranteeing that the random walk's transition probability matrix is ergodic[48]. If the directed network was not strongly connected, the PageRank algorithm[74] may be utilized[49,50]. Fourth, our research was based on a restricted number of mapped areas due to the absence of documented marmoset-human homologous areas. Fifth, BOLD-fMRI collection for the marmosets concerning the discovery dataset (all males) and replication cohort (11 males vs. 1 female) was sex-biased. Finally, BOLD-fMRI images used a consistent parcellation for all participants, which assumed that each participant's brain areas could be mapped to the exact spatial location. Future works should consider the effect of individual variation in functional boundaries[7,10].

In summary, we quantified the degree of correspondence between cellular connectome and cortical functional activity in marmosets. The cellular-functional relationships revealed a spatial gradient, resembling microstructural profiles and macroscale brain organization. Notably, the connectional information derived from cellular-resolution tracing in marmosets was translatable to the human brain. These findings contribute to our understanding of how neuronal tracing connectome shapes cortical activity.

## Methods
### Neuronal tract-tracing data in marmosets
We included the retrograde neuroanatomical tract-tracing data in marmosets from the publicly available Marmoset Brain Architecture Project (https://www.marmosetbrain.org/)[35]. All experiments procedures were approved by the Monash University Animal Experimentation Ethics Committee. This dataset consists of 143 retrograde tract-tracing experiments in marmosets ($N = 52$; 21 females and 31 males; 1.4-4.6 years), injected with six fluorescent retrograde tracers[35,41]. The strength of the tract-tracing CC from the labeled neurons (source) to the injection area (target) was quantified by the FLNe[32].

Localization of the tracer injections and labeled neurons was based on the Paxinos parcellation[46,47] with 116 cortical areas of the left hemisphere. The injection areas were concentrated in 55 of 116 areas, including the premotor, prefrontal, superior temporal, parietal, and occipital complexes[35]. An asymmetric CC matrix ($55 \times 116$) described the $\log_{10}$(FLNe) value for each target-source pair across injections within the same target area (Supplementary Fig. 2a). To reflect the main features of the whole interareal network[36,75], we used an edge-complete weighted asymmetrical matrix $A$ ($55 \times 55$), which included only 55 injected areas with pairwise-complete connection values (Supplementary Fig. 2b). Furthermore, those 55 injection areas were

assigned into eleven cortical classes[46,47] (Supplementary Fig. 1; Supplementary Table 1). The CC matrix $A$ quantified according to the Paxinos atlas[46,47] of the left hemisphere was used to construct a directed graph $G_{55\times55}$ for the following analysis, unless otherwise stated.

## MRI data in marmosets

We collected MRI data from the Marmoset Brain Mapping project (https://marmosetbrainmapping.org/data). All experimental procedures were approved by the Animal Care and Use Committee (ACUA) of the National Institute of Neurological Disorders and Stroke. Seven marmosets (all males; 3–9 years) from the National Institutes of Health (NIH) cohort were scanned using a 7 T/30 cm horizontal MRI to obtain structural information and resting-state BOLD-fMRI data[34,45]. All rs-fMRI data were collected in ParaVision 6.0.1 software using a 2D gradient echo planar imaging (EPI) sequence (TR = 2 s, 512 volumes (17 min) per run). After each rs-fMRI session, a T2-weighted image was scanned for spatial registration. Furthermore, in vivo diffusion MRI (dMRI) data were acquired using a 2D diffusion-weighted spin-echo EPI sequence. A detailed account of image acquisition protocol can be found in Supplementary Section 1.1.

All rs-fMRI data were corrected for slice-timing, head motion, and EPI distortion. Further preprocessing procedures included regression covariates, band-pass filtering (0.01–0.1 Hz), and registration of the preprocessed images to the template space of Marmoset Brain Atlas Version-3 (MBMv3, https://marmosetbrainmapping.org/v3.html)[47]. Finally, fMRI data were smoothed using a 1 mm full-width at half-maximum (FWHM) Gaussian kernel. A detailed account of fMRI data preprocessing was described in Supplementary Section 1.2.

After preprocessing and quality control of each scan, all preprocessed fMRI data were parcellated into 116 cortical areas based on the Paxinos atlas[46,47]. The averaged time series were obtained for each brain area. We extracted the time series of the 55 injected areas corresponding to the CC matrix. We showed the temporal mean, standard deviation, and activity concentration distribution of regional BOLD-fMRI signals for each marmoset (Supplementary Fig. 13). The standard deviations of the BOLD-fMRI signals exhibited a non-uniform distribution across regions. To overcome the activity concentration bias at the region level, we normalized the BOLD-fMRI signals by subtracting their temporal mean and dividing them by their temporal standard deviation (i.e., $z$-score). Note that normalization is unnecessary to avoid introducing bias if the signals in all regions have similar amplitudes (or at least standard deviation). Finally, the FC matrix was constructed by estimating the Pearson correlation coefficient ($r$) of the time series between paired brain areas.

All dMRI data were preprocessed using the MRtrix3 package (v3, https://www.mrtrix.org/)[76]. Briefly, probabilistic diffusion tractography was used to reconstruct each individual's structural connectivity (SC) matrices based on the Paxinos atlas[46,47]. The number of streamlines normalized by the total streamlines corresponds to the connection weights. More information regarding the individual SC reconstructions is available at ref. 34. By averaging all individual SC matrices, a group SC matrix $A_{SC}$ was generated. We thresholded the SC matrix to construct a connection matrix that matched the density of CC since the connection density of CC and SC varied. Finally, SC eigenmodes were derived according to previous methods[12,16,20], which were obtained by eigen-decomposition of normalized undirected graph Laplacian[13,14], $\mathscr{L}_{undir} = I - D^{-\frac{1}{2}} A_{SC} D^{-\frac{1}{2}}$, where $D$ stands for the diagonal degree matrix of the adjacency matrix $A_{SC}$, and $I$ is the identity matrix.

## MRI data in humans

We used resting-state BOLD-fMRI datasets from the 100 unrelated subjects ($N = 100$; 54 females and 46 males; 22–36 years) provided by the Human Connectome Project (HCP) (https://db.humanconnectome.org/)[77]. The HCP data was acquired using protocols approved by the Washington University Institutional Review Board. All participants were volunteers and provided informed consent. We analyzed rs-fMRI data acquired in the first scanning session using a left-to-right (LR) encoding direction. The scan lasted 14.4 min (TR = 0.72 s) with 1200 time points. A detailed description of the image acquisition protocol is available at ref. 77.

The BOLD-fMRI data was preprocessed according to the HCP minimal preprocessing pipelines[78]. The rs-fMRI data was adjusted for gradient nonlinearity, head motion, and geometric distortions. Further preprocessing procedures included registration of the corrected images to the T1 weighted images, brain extraction, global intensity normalization, high-pass filtering (cut-off at 2000 s)[79], and elimination of residual confounds through the ICA-FIX method[80]. We did not conduct any additional preprocessing steps. Finally, the preprocessed time series were parcellated into 180 cortical areas of the left hemisphere using the HCP-MMP1.0 parcellation[61].

## Graph signal processing on the marmoset data

**Connectome Laplacian and eigenmodes of the CC.** We assumed the CC as a weighted directed graph $\mathscr{G} = (V, E, A)$, where $V = \{v_1, v_2, \cdots, v_N\}$ is a finite set of nodes, $N$ is the number of nodes, $E \in V \times V$ is a set of directed edges, and adjacency matrix $A = \{A_{ij}\}_{1 \leq i,j \leq N} \in R_+^{N \times N}$ is defined as the FLNe value of the CC.

The graph Laplacian linked a bridge between spectral graph theory and signal processing[13,14,16]. It was difficult to generalize the typical normalized undirected graph Laplacian to directed graphs since its symmetries were no longer verified[44]. A suitable reference operator had to be found to expand the Laplacian-based Fourier analysis from undirected graphs to directed graphs.

As an acceptable reference operator for expanding the signal processing framework to directed graphs, the random walk operator on graphs was presented[49]. The random walk operator was related to the concept of diffusion on graphs, which could transform any graph into a Markov chain[50]. Chung[48] defined a symmetric normalized Laplace operator for strongly connected directed graphs. Notably, the normalized directed graph Laplacian, which has been used in spectral clustering[81], graph embedding[82], and classification applications[83], may represent the graph's directionality and edge density by leveraging the random walk operator.

The normalized directed graph Laplacian[48] is defined as (Supplementary Section 2.1),

$$\mathscr{L} = I - \frac{\Pi^{1/2} P \Pi^{-1/2} + \Pi^{-1/2} P^T \Pi^{1/2}}{2}, \tag{1}$$

where $P = D^{-1}A$ is the transition matrix of the ergodic Markov chain (Supplementary Fig. 14; Supplementary Section 5)[84,85], $D = \sum_{j=1}^{N} A_{ij}$ denotes the diagonal matrix of the out-degrees of $A$, $\Pi = diag\{\pi(v_1), \cdots, \pi(v_N)\}$ is the diagonal matrix of the stationary distribution $\pi$ (Supplementary Fig. 15), i.e., $\pi P = \pi$ of the random walk with the transition matrix $P$, and $I$ is an identity matrix. The Laplacian satisfies $\mathscr{L}^T = \mathscr{L}$, i.e., $\mathscr{L}$ is a symmetric matrix.

The CC eigenmodes were obtained by the eigen-decomposition of normalized directed graph Laplacian $\mathscr{L}$,

$$\mathscr{L}\Psi = \Lambda\Psi, \tag{2}$$

where $\Lambda = diag(\lambda_1, \lambda_2, \cdots, \lambda_N)$ is the eigenvalues of $\mathscr{L}$ ordered according to $0 \leq \lambda_1 \leq \lambda_2 \leq \cdots \leq \lambda_N$[86], and associated $\Psi = \{\psi_k\}_{k=1}^{N}$ is a set of orthogonal eigenvectors. The Dirichlet energy of the eigenvectors of the random walk operator on a directed graph is associated with eigenvalues, which can be regarded as graph frequencies[49]. Hence, the eigenvalues can be interpreted as frequencies, and the eigenvectors are known as connectome eigenmodes. Low-frequency eigenmodes

vary slowly over the graph, whereas high-frequency eigenmodes vary more rapidly[16,87].

**Connectome eigenmode decomposition of fMRI in marmosets.** The cortical activity over the node $v_i$ at time-point $t$ was denoted as $f_{v_i}(t) \in R^{N \times T}$, $i = \{1, 2, \cdots, N\}, t \in \{1, 2, \cdots, T\}$. Then, we used the eigenvector matrix $\Psi$ to define the graph Fourier transform (GFT) of the graph signal $f_{v_i}(t)$ as $w_k(t) = \Psi^T f_{v_i}(t)$. The GFT coefficient $w_k(t)$ describes how much each CC eigenmode contributes to the observed functional activity $f_{v_i}(t)$[22].

The original signal $f_{v_i}(t)$ can be decomposed as a linear combination of the set of CC eigenmodes. The inverse GFT (IGFT) of $w_k(t)$ to $\mathscr{L}$ is defined as (Fig. 1d)

$$f_{v_i}(t) = w_1(t)\psi_1 + w_2(t)\psi_2 + \cdots + w_N(t)\psi_N = \sum_{k=1}^{N} w_k(t)\psi_k(v_i) \quad (3)$$

**Graph energy spectral density and activity concentration.** The magnitude of each CC eigenmode $\{\psi_k\}_{k=1}^{N}$ in the cortical activity pattern at any given time point $t$ was called graph energy spectral density (ESD). The ESD describes the energy present in each connectome eigenmode during a graph time-varying signal, which was computed as the absolute square of the amplitudes for a specific connectome eigenmode $\psi_k$: $ESD(\psi_k, t) = |w_k(t)|^2$.

In addition, to measure the cortical activity concentration of the graph frequency component at the brain area $v_i$, we defined the $L_2$-norm of BOLD-fMRI signal $f_{v_i}(t)$ across all time points: $E_{v_i} = \|f_{v_i}(t)\|_2$, which provided an interpretation of energy for each graph frequency component[12,16,22].

**Measuring cellular-functional decoupling**
Given a graph signal $f_{v_i}(t)$ with graph spectral coefficients $w_k(t)$, cortical activities can be isolated into low-frequency components (coupled to the CC) and high-frequency components (decoupled from the CC)[12,22]. To determine the cut-off frequency $C$, we used the graph spectrum dichotomization method[12] to divide the graph spectra into two parts with equal energy based on average ESD (across time and subjects). The graph low-pass filter matrix $\Psi^{(low)} \in R^{N \times N}$ only keeps the first $C$ eigenvectors (columns of $\Psi$) and sets other components to 0. The graph high-pass filter matrix $\Psi^{(high)} \in R^{N \times N}$ keeps the last $N - C$ eigenvectors (see Supplementary Section 6 for robustness of results to parameter selection). Therefore, the filtered low-frequency $f_{v_i}^{low}(t) \in R^{N \times T}$ and high-frequency $f_{v_i}^{high}(t) \in R^{N \times T}$ components (Fig. 1d) are expressed as

$$f_{v_i}^{low}(t) = \Psi^{(low)} \Psi^T f_{v_i}(t) \quad (4)$$

$$f_{v_i}^{high}(t) = \Psi^{(high)} \Psi^T f_{v_i}(t) \quad (5)$$

The cellular-functional decoupling (CFD) of each brain area can be quantified as the ratio between the $L_2$-norm of high-frequency versus low-frequency components across time points[12], resulting in

$$CFD_{v_i} = \log_2 \left( \frac{\|f_{v_i}^{high}(t)\|_2}{\|f_{v_i}^{low}(t)\|_2} \right) \quad (6)$$

We chose the binary logarithmic form of this index, so CFD values of 0 represent a perfect balance between cellular-functional coupling and decoupling. CFD values around −1 imply that cortical activity is strongly coupled to underlying monosynaptic connections, whereas CFD values around 1 indicate the reverse.

**Relation to cortical microstructure profiles and macroscale gradients**
To reveal the potential biological interpretation of CFD at different biological scales, we further analyzed its relation to the cortical microstructural attributes and connectivity gradients. We used Spearman correlation to measure the spatial correspondences across brain areas. The significance of correlations was assessed using BrainSMASH, a spatial autocorrelation (SA)-preserving surrogate method[88] (Supplementary Section 4).

First, we characterized the relationships between the CFD and cortical myeloarchitecture. The map of marmoset myelin content was acquired from MBMv3 (https://marmosetbrainmapping.org/v3.html)[47]. The myelin content was quantified as the ratio of T1-weighted and T2-weighted in cortical gray matter (T1w/T2w)[59]. Second, we assessed the spatial concordance between the CFD and neuronal distribution. The map of marmoset neuronal counts was acquired via a freely accessible repository (http://www.marmosetbrain.org/cell_density)[58].

We further quantified the spatial concordance between CFD and macroscale functional gradients in the marmoset brain. We used diffusion embedding mapping[23,89,90] to identify spatial axes of interregional functional changes via BrainSpace (http://github.com/MICA-MNI/BrainSpace)[91] (Supplementary Section 3).

**Generalizing marmoset-derived eigenmodes to the human brain**
We attempted to use the marmoset's CC eigenmodes to capture intrinsic brain activity and connectivity in the human brain. First, we obtained the homologous landmarks, including 11 candidate cortical areas across humans and marmosets (Supplementary Table 2). Specific details regarding homologous landmark selection can be found in ref. 60. Second, we extracted an asymmetrical weighted homologous connection matrix ($11 \times 11$) from the marmoset's tracer-based CC. Third, we estimated the connectome eigenmodes of this homologous CC matrix (Supplementary Fig. 11a). Then, the human intrinsic brain activity was decomposed into a combination of the marmoset's CC eigenmodes. Using this decomposition, we reconstructed human cortical activity at each time point via marmoset's CC eigenmodes (ordered by eigenvalues), and further generated an area-to-area FC matrix. Finally, we quantified reconstruction accuracy by calculating the ratio between the observed and reconstructed cortical activity concentration, and the correlation between the empirical and reconstructed FC matrix.

**Statistical analysis**
We performed a nonparametric permutation test to examine the spatial significance of the cortical class-level activity concentrations. We generated 1000 permutations for low- and high-frequency components using graph spectral randomization (SR)[16] (Supplementary Section 2.2) and computed a null distribution of mean activity concentrations for each cortical class. The mean activity concentration was greater than 95% (>95th percentile) of the null permutations, which was identified to be significantly concentrated in a given class[16]. Significance was set at $p_{SR} < 0.05$ with FDR-correction for multiple comparisons across eleven cortical classes.

We performed a nonparametric permutation test to localize empirical CFD of each cortical area that was significantly more coupled or decoupled than the graph surrogate activity[12]. At a significance level of $\alpha = 1/(19+1) = 0.05$, we first generated 19 graph spectral randomization surrogate signals to threshold the CFD for every animal. Furthermore, the binomial distribution $P(n)$ with $n(n = 100)$ tests was utilized to threshold the group average CFD across individuals, correcting for multiple comparisons across 55 cortical areas.

## Sensitivity and reproducibility analyses

The cut-off frequency of CC eigenmodes was determined using the graph spectrum dichotomy approach[12]. We also examined separating the CC eigenmodes into low, medium, and high components[22,87] to further validate the sensitivity of parameter selection on the results. Specifically, we chose different lowest $K_L$ and highest $K_H$ CC eigenmodes to decompose the observed BOLD-fMRI into low- and high-frequency components. The spatial correlations of low- and high-frequency components, as well as the CFD patterns between the original and robustness analyses were compared.

To examine the reproducibility of patterned CFD, we included another independent rs-fMRI data ($N = 12$; 1 female and 11 males; 2–4 years) from ION, China (https://marmosetbrainmapping.org/data). All experimental procedures were approved by the ACUA of the Institute of Neuroscience, Chinese Academy of Sciences. The marmosets from the ION cohort were scanned in a 9.4 T/30 cm horizontal MRI scanner. Additional information regarding image acquisition protocol is available at ref. 34. We repeated the CFD pattern in this data and then estimated the spatial correlation of regional CFD between the original and replication analysis. Moreover, we calculated the Dice coefficient to assess the overlap of significantly coupled or decoupled areas in the two datasets.

## Reporting summary

Further information on research design is available in the Nature Portfolio Reporting Summary linked to this article.

## Data availability

The retrograde neuroanatomical tract-tracing data is publicly available from the Marmoset Brain Architecture Project (https://www.marmosetbrain.org/)[35]. The NIH and ION awake marmosets MRI datasets are available from the Marmoset Brain Mapping (https://marmosetbrainmapping.org/data)[34]. The HCP dataset[77] is publicly available at https://db.humanconnectome.org/. The Paxinos marmoset parcellation is publicly available from the MBMv3 resource (https://marmosetbrainmapping.org/v3.html)[47]. The HCP-MMP1.0 human cortical atlas[61] is publicly available at https://balsa.wustl.edu/study/show/RVVG. The myelin content, allometric scale, and cortical gene expression in human is available through neuromaps (https://github.com/netneurolab/neuromaps)[92]. Source data are provided in this paper.

## Code availability

The code used to conduct the main results in this study is available at https://github.com/weiliao81/Marmoset_CFD and on Zenodo (https://doi.org/10.5281/zenodo.10728317)[93]. The code for spatial autocorrelation-preserving surrogate brain maps can be implemented through the brainSMASH toolbox (https://github.com/murraylab/brainsmash)[88]. The code for gradient analysis and the Moran spectral randomization can be performed via BrainSpace (http://github.com/MICA-MNI/BrainSpace)[91]. The brain surfaces were visualized using Connectome Workbench (v1.5.0, https://www.humanconnectome.org/software/connectome-workbench)[94] and Python script (https://github.com/netneurolab/marmoset_connectome)[75].

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

## Acknowledgements

We thank International Science Editing (https://www.internationalscienceediting.com/) for editing this manuscript. This study was supported by the National Key R&D Program of China (2022YFC2009906 (W.L.) and 2022YFC2009900 (W.L.)), the National Natural Science Foundation of China (62036003 (H.C.), 82121003 (H.C.), and 82202250 (J.L.)), the Fundamental Research Funds for the Central Universities (ZYGX2022YGRH008 (W.L.)), and the Medical-Engineering Cooperation Funds from University of Electronic Science and Technology of China (ZYGX2021YGLH201 (H.C.)).

## Author contributions

W.L. and H.C. supervised the study. J.X. and W.L. designed the methodology and implemented the visualization. C.L. preprocessed and organized the MRI data. J.L., Y.M., and S.Y. contributed and implemented the investigation. J.X., W.L., and C.L. wrote and edited the manuscript.

## Competing interests

The authors declare no competing interests.
