## [Peer Review File · Nature Communications]

Decomposing Cortical Activity through Neuronal Tracing Connectome-eigenmodes in MarmosetsREVIEWER COMMENTS

Reviewer #1 (Remarks to the Author):

Review

Decoding Cortical Activity through Neuronal Tracing Connectome-Harmonics in Marmosets

The manuscript presents an analysis of the marmoset's brain structure-function relationship using a recent graph decomposition framework that has gained significant interest in the neuroscience and engineering communities over the past few years. Specifically, this framework is often being applied to healthy and disease/disrupted brain conditions but its anatomical underpinnings rely either on publicly available human datasets or subject-specific tractography derived from diffusion-MRI. Some major caveats of these techniques are the undifferentiation of anatomical substrate at sub-millimetre scale (at best) and the lack of directionality resulting from the fiber-tracking protocol. To this extent, the connectome harmonics (or connectome eigenmodes) frameworks suffer from these limitations like any other connectomic approach.

The presented study offers the opportunity to assess the validity of such connectome eigenmode frameworks to a non-human primate species, whereby directionality of connections is known, and its potential generalization across primates. Particularly, the utilization of marmosets is well-suited to investigate a scale-down version of primate brain with sub-millimetre (up to cellular) definition in-vivo.

The study describes a hierarchical organization of the marmoset brain through the lens of connectome eigenmodes, using directed connectomes derived from neuronal tracing, that conforms to the hierarchical organization observed in the human brain using the same framework using undirected graphs. This result is well exposed and stands out as solid (Figs 1-4). Its significance for the field is highly valuable.

The authors then expose associations from marmoset to human brain measures that support a generalization across species based on the mapping of 11 cortical regions (Fig 5). While being a valuable translation for those concerned regions, the claims presented in this section concerning whole brain might be perceived as overstated given the small number of mapped regions. These results are still worth reporting but would benefit some clarifications about methodology and acknowledgement of its limitations.

Finally, the methods for deriving discrete Laplacians from directed graphs are of global interest to the neuroscience community. To the best of my knowledge, all human studies use undirected graphs derived from dMRI and tractography. The derivation of a discrete Laplacian using directed graph from cellular tracing have mathematical intricacies that would be worth mentioning and may impact the interpretation of such discrete Laplace operator.

Overall, I would fully endorse the publication if the following comments can be addressed:

Main text:

- Fig 1B: The colorcode in scale of eigenvalue is confusing. Are the eigenvector amplitudes scaled by their corresponding eigenvalue? Since there is a single eigenvalue associated to each eigenvector, an eigenvector cannot take several eigen-values. What is the vertical bar besides the cortical surfaces? Please annotate as eigenvector if it's the case.

- p7: In some Connectome Eigenmodes/Harmonics literature, there is a nomenclature distinction between connectome harmonics and connectome eigenmodes (Naze et al. 2021; Pang et al, 2023). Connectome Harmonics as introduced in Atasoy et al. (2016) are created from a mix of cortical mesh (local gray matter) and tractography (long-range white matter) connections. Connectome Eigenmode are derived only using parcellated brain atlases and therefore do not incorporate local mesh-derived interactions (e.g. Huang et al. 2018). Geometric connectomes (Pang et al. 2023) use only mesh connections. It might be useful to follow this nomenclature and discuss how this specific work bridges the gap between the 2 approaches using a millimeter resolution of white-matter tracing.
- p7: A note on polarities: how was the polarity of each eigenvector assigned? In the case of symmetric connectomes, therefore with only positive eigenvalues, the sign of the eigenvector is rather arbitrary. How does it differ in the case of asymmetric connectome presented here?
- p12, l.209-210: the reported statistics in text do not match the ones in the figures.
- p.12: strong cellular-functional dependency should rather be aligned/liberal than liberal/aligned, I suggest renaming it cellular-functional decoupling throughout the manuscript, or it be renamed CFI: cellular-functional independence.
- p14: l.250-252: *“According to these results, the marmoset CC harmonics concisely and parsimoniously describe human resting-state cortical activity and connectivity”* may be overstated based on only 11 mapped regions.
- p.17: l.308-318: the reason of having the OFC and IPFC in both aligned and liberal components might be due to normalizing the BOLD time series (see first comment about methods). This normalization has consequences when computing the energy $|w_i|^2$ present in each mode. This should be discussed.
- l.363-365: *“We found that marmoset’s CC harmonics provided a more parsimonious representation of human cortical activity than human’s dMRI-based SC harmonics”*: overstated when based on only 11 regions.
- l.377-386: the limitation section could tackle technical aspects of the calculation of the Laplacian on directed graph, and the role of white matter vs. gray matter connectivity in formulating the connectome.

Methods:

- l.425-426: normalizing time series have implication for the GSP analysis, particularly when computing nodal energy after graph filtering.
- l.464: subscript i is not defined, would be better defining node activity v_i .
- l.485-486: would read better *“cortical activities can be isolated into aligned and liberal components”*.
- l.488: spectral => spectra
- l.496-500: related to p.12 above, I find the naming of the metric counter-intuitive and make the interpretation of the results convoluted. Either call the metric cellular-function decoupling, or use aligned/liberal components so that a larger CFD would

mean a larger structure-function coupling.

- l.504: connective => connectivity

Supplementary materials:

SI-2:

- Suppl. Materials p.4, eq.(1) : please add textbook reference for this equality.
- Suppl. Materials, p.5, is there any test or condition on P that guarantee that X_i is ergodic (irreducible and aperiodic)?
- Suppl. Materials p.5, eq.(2) : Please expand on how pi is calculated and how it can be interpreted. This normalized Laplacian formula adds P and its transpose, (scaled by the Perron vector pi), therefore making it symmetric, why is this necessary?

Adding a visualisation of pi (e.g. a bar plot) could be informative to some readers.

Supplementary Results (p.8-9) and associated Suppl. Figs 8 & 9 are not referred anywhere in main text. How are they relevant?

References:

Atasoy, Selen, Isaac Donnelly, and Joel Pearson. "Human Brain Networks Function in Connectome-Specific Harmonic Waves." *Nature Communications* 7 (January 21, 2016): 10340. <https://doi.org/10.1038/ncomms10340>.

Huang, W., T. A. W. Bolton, J. D. Medaglia, D. S. Bassett, A. Ribeiro, and D. Van De Ville. "A Graph Signal Processing Perspective on Functional Brain Imaging." *Proceedings of the IEEE* 106, no. 5 (May 2018): 868–85. <https://doi.org/10.1109/JPROC.2018.2798928>.

Naze, Sébastien, Timothée Proix, Selen Atasoy, and James R. Kozloski. "Robustness of Connectome Harmonics to Local Gray Matter and Long-Range White Matter Connectivity Changes." *NeuroImage* 224 (January 1, 2021): 117364. <https://doi.org/10.1016/j.neuroimage.2020.117364>.

Pang, James C., Kevin M. Aquino, Marianne Oldehinkel, Peter A. Robinson, Ben D. Fulcher, Michael Breakspear, and Alex Fornito. "Geometric Constraints on Human Brain Function." *Nature* 618, no. 7965 (June 2023): 566–74. <https://doi.org/10.1038/s41586-023-06098-1>.

Reviewer #2 (Remarks to the Author):

The paper investigates the relationship between structure and function in marmoset brains. For that purpose, the authors decompose the tract-tracing based connectome into harmonics, and look for the possibility to decode resting-state functional MRI brain activity using such basis. Overall, the results show that connectome harmonics effectively decode brain activity. The pattern of structure-function relationship recapitulates the hierarchical organization of the brain, i.e., a unimodal to transmodal gradient, which is observed across various properties (myelin, neuronal count, connectivity...). At the end, the authors make use of the homology with the human brain to demonstrate a potential general principle of cross-species harmonic constraints.

The manuscript treats an important topic in neuroscience. It is overall well-written, providing sufficient background. However, I have a couple of significant concerns about the analyses and interpretations that I hope the authors can address.

First at all, while the study seems seriously conducted, the results do not appear spectacular and new. It has been repeatedly reported the presence of such organizational axis in the brain across species (also using the CFD measure). The fact that it is also observed in marmosets is rather expected.

For the marmoset activity, what is the accuracy of the reconstruction? It would be informative to have similar figures as 5 and S7 for marmoset results. There are also concerns and debates about the use of such harmonic basis for explaining empirical functional data (see, e.g., <https://doi.org/10.1038/s41586-023-06098-1> and <https://doi.org/10.1101/2023.07.20.549785>). What is the gain compared to null models for both species? Is a random harmonic basis would fit the empirical as well? The CFD is rather a very aggregate measure, what are the true contributing harmonics?

The energy analysis is quite hard to follow, would it be possible to compare directly low versus high frequency classes in figure 2, a bit like figure S4? Absolute numbers are meaningless. The discussion about OFC and dlPFC appears exacerbated because if you look at figure S4, we can also consider auditory, pmACC and mPFC with the same trend, then the

reasoning does not hold anymore. Same story when talking about the CFD gradient, coupling appears in unimodal regions, yes, but what about PCC? Also, decoupling appears in trans-modal regions but to the same extent as the coupling, except for OFC?

I wonder whether the human-marmoset comparison is fair enough. Could it be that the slight increase in prediction of CC compared to SC is simply due to the fact that CC is directed? Or maybe they differ in terms of density? Also, I might have overlook some aspects, but why all reconstruction accuracies reach 1 when considering all harmonics?

I find the term cellular-function confusing, it makes me think about something which has nothing to do with connectivity.

Reviewer #3 (Remarks to the Author):

This paper used graph signal processing to study the structure-function relationship in marmosets. The authors used neuronal-tracing data to create the underlying anatomical connectivity (cellular connectome) and showed that their method can disentangle cortical activity according to a hierarchy of coupled and decoupled regions, following known hierarchical organization of the brain. In most respects I found the paper to be interesting. However, I do have several major and minor concerns/issues, which are itemized below.

Major concerns:

1. On methodology

a.) CC is a non-symmetric (directed) matrix, hence its Laplacian is also non-symmetric. Mathematically, the eigenvectors of non-symmetric matrices do not necessarily form a complete orthonormal basis set. Hence, they may not be linearly independent, which is important for basis set expansions as done in Eq. (3). The authors need to quantitatively verify that the CC harmonics are linearly independent, otherwise using them to decompose cortical activity data may not be valid.

b.) The authors should explain why transforming the CC to a graph Laplacian via the transition matrix is necessary. Otherwise, non-expert readers may confuse this step as a

standard approach for all cases, including undirected matrices, which is not true.

c.) The analysis revolves around the dichotomy between low- and high-frequency harmonics, which heavily relies on the filter cutoff that preserves the total energy in each band. I know past studies have done the same, but I find this approach and the hard cutoff ill-defined. The authors need to provide a more compelling reason for this approach and/or show robustness to the choice of the filter cut-off.

d.) In the calculation of CFD, the authors should explain why the logarithmic transformation is needed. It is also unclear from Eq. (6) how the temporal component disappears. More importantly, the physical meaning of CFD is difficult to comprehend. For example, how does it inform the correspondence between functional signals and CC (e.g., Lines 178-179)?

e.) The authors used the spin test for null testing, which I think is inappropriate because there are a lot of discontinuities/missing regions. There is a greater chance of reassigning missing data to the nearest region, hence the null statistics are not stringent enough. A parametric approach such as variogram matching is more suitable, which the authors should look at.

f.) I also found it bizarre that the authors used human functional data from a dataset different from HCP. HCP has one of the best quality data out there, so why not just use it all throughout? In fact, I highly suggest they check the robustness of their results on HCP data.

2. On analysis and results

a.) Figure 2 is really confusing to read. Statements in Lines 145 to 152 are not supported by the figure either. For example, the authors said that there is high energy concentration of low graph frequency in PmACC, but its value is comparable to those of dIPFC and vIPFC. There is also a lot of differentiation within cortical classes to make overarching statements like these. The authors also need to do some statistical testing in comparing the different cortical classes.

b.) The authors said that results were replicated across datasets. Whilst the correlations of unthresholded CFD are high, there are a lot of differences in the significant regions found in the two datasets. How do they reconcile this?

3. The authors need to take extra care in making inferences from their human results. There are several issues with the current approach.

a.) The dimensionality of the data and harmonics was reduced to 11 to restrict analysis to 11 homologous regions between the species. This removes a lot of important connections to other regions; hence, the resulting harmonics cannot account for true brain activity. In fact, this becomes problematic even for the marmoset results because the authors are producing harmonics from a subset of their original harmonics, so that the spatial patterns found in Fig. 1b are difficult to reconcile with those in Fig. S7A.

b.) It is unfair to compare the performance of the marmoset CC harmonics to human diffusion-based harmonics as the latter is inherently noisier. Moreover, the case would have been stronger if the authors follow my Comment 1f.

c.) The authors should also test whether the CC harmonics can predict 11 non-homologous (e.g., randomly chosen) regions in humans. If they find that CC harmonics perform equally well, then there is a problem here, with the results trivially driven by the mathematics of basis set expansions and not some biologically relevant evolutionary process.

4. On discussion

a.) There is an article in Nature that came out this year showing that the geometry of the human brain fundamentally constrains brain function, not just complex connectivity, using the same framework in this study. The authors should discuss how their work ties with this emerging idea and whether the same geometric influence can be observed in marmosets.

b.) Lines 302 to 308. The authors argued that the current work addresses traditional approaches of averaging temporal data to obtain FC, which inadvertently removes the rich

characteristics of brain dynamics. I found this baffling as the authors also took the average of their CFD across time, so in principle, they are also removing temporal dynamics.

c.) I don't understand Lines 365 to 368 and Lines 373 to 374. How could CC harmonics inform human connectivity, especially given that the CC harmonics calculated in the work did not have full brain coverage and that not all marmoset brain regions can be homologously mapped to humans? The authors should tone down these inferences.

d.) The authors also claimed that CC could inform diffusion-based connectivity. The marmoset brain mapping resource has diffusion MRI available, so the authors can easily compare the performance of marmoset CC and diffusion-based connectome to test this empirically.

e.) Because of the sparsity of regions, the CC harmonics do not match typical spatial patterns and their natural ordering found in the literature (e.g., second harmonics typically follows the anterior-posterior gradient). The authors should discuss this discrepancy.

Minor concerns:

5. The authors need to be careful with terminologies such as "decoding", "complexity", and "energy". They have different connotations in different fields. In fact the term "energy" is not properly defined. I would also refrain from using the terms "liberal" and "aligned" as they are misleading. Just stick with low- and high-frequency.

6. There are a couple of things that need further clarification.

a.) Is the CC matrix data only unihemispheric?

b.) What do the graphs at the bottom of Figure 1D mean?

c.) Wrong caption in Figure 1B colorbar. It is the eigenvector not eigenvalue.

d.) Line 132: principal gradient of what?

e.) Provide details of preprocessing of human data.

f.) How to get the diagonal matrix Π in Eq. (1)?

The length of this report is not short, thus, we made a table of contents to make it convenient to check our responses to the comments. The modified text in the manuscript is highlighted in yellow.

Table of contents

Comments from the Reviewer #1 (18 points)	2
Comments from the Reviewer #2 (5 points)	28
Comments from the Reviewer #3 (23 points)	45

General changes:

1. We clarified the methodology and take-home messages (highlighted by Reviewer #1 Q16-17 and #3 Q1-6).
2. We complemented the parsimony and specificity of cellular connectome eigenmodes to explain brain function in marmosets in Fig. 2 (highlighted by Reviewer #2 Q2 and #3 Q15).
3. The low- and high-frequency components were re-analyzed and discussed in Fig. 2 (highlighted by Reviewer #1 Q7, #2 Q3, and #3 Q7).
4. The HCP dataset was used to re-analyze the generalization of marmoset-derived eigenmodes to humans in Fig. 5 (highlighted by Reviewer #3 Q6).

Comments from the Reviewer #1 (18 points)

Decoding Cortical Activity through Neuronal Tracing Connectome-Harmonics in Marmosets

The manuscript presents an analysis of the marmoset's brain structure-function relationship using a recent graph decomposition framework that has gained significant interest in the neuroscience and engineering communities over the past few years. Specifically, this framework is often being applied to healthy and disease/disrupted brain conditions but its anatomical underpinnings rely either on publicly available human datasets or subject-specific tractography derived from diffusion-MRI. Some major caveats of these techniques are the undifferentiation of anatomical substrate at sub-millimetre scale (at best) and the lack of directionality resulting from the fiber-tracking protocol. To this extent, the connectome harmonics (or connectome eigenmodes) frameworks suffer from these limitations like any other connectomic approach.

The presented study offers the opportunity to assess the validity of such connectome eigenmode frameworks to a non-human primate specie, whereby directionality of connections is known, and its potential generalization across primates. Particularly, the utilization of marmosets is well-suited to investigate a scale-down version of primate brain with sub-millimetre (up to cellular) definition in-vivo.

The study describes a hierarchical organization of the marmoset brain through the lens of connectome eigenmodes, using directed connectomes derived from neuronal tracing, that conforms to the hierarchical organization observed in the human brain using the same framework using undirected graphs. This result is well exposed and stands out as solid (Figs 1-4). Its significance for the field is highly valuable.

The authors then expose associations from marmoset to human brain measures that support a generalization across species based on the mapping of 11 cortical regions (Fig 5). While being a valuable translation for those concerned regions, the claims presented in this section concerning whole brain might be perceived as overstated given the small number of mapped regions. These results are still worth reporting but would benefit some clarifications about methodology and acknowledgement of its limitations.

Finally, the methods for deriving discrete Laplacians from directed graphs are of global interest to the neuroscience community. To the best of my knowledge, all human studies use undirected graphs derived from dMRI and tractography. The derivation of a discrete Laplacian using directed graph from cellular tracing have mathematical intricacies that would be worth mentioning and may impact the interpretation of such discrete Laplace

operator.

Overall, I would fully endorse the publication if the following comments can be addressed:

Response:

We greatly appreciate the insightful comments on our manuscript. We will address the points raised by the Reviewer in the below responses. We hope our responses are satisfactory and improve the quality of the current work.

Main text:

Q1: Fig 1B: The colorcode in scale of eigenvalue is confusing. Are the eigenvector amplitudes scaled by their corresponding eigenvalue?

Since there is a single eigenvalue associated to each eigenvector, an eigenvector cannot take several eigen-values. What is the vertical bar besides the cortical surfaces? Please annotate as eigenvector if it's the case.

Response:

We apologize for miswriting the eigenvector as eigenvalue. We have corrected it in the revision. In the last version, the vertical bars represented the first four CC eigenmodes (eigenvectors) in vector form (in ascending order by eigenvalue), and were projected onto the marmoset brain surface. Colors visualize arbitrary units, i.e., the weights in eigenvectors.

Fig. 1. b The first four CC eigenmodes (ψ_1 – ψ_4 , in ascending order by eigenvalue) were projected onto the marmoset brain surface. Colors visualized arbitrary units, i.e., the weights in eigenvectors.

Q2: p7: In some Connectome Eigenmodes/Harmonics literature, there is a nomenclature distinction between connectome harmonics and connectome eigenmodes (Naze et al. 2021; Pang et al, 2023). Connectome Harmonics as introduced in Atasoy et al. (2016) are created from a mix of cortical mesh (local gray matter) and tractography (long-range white matter) connections. Connectome Eigenmode are derived only using parcellated brain atlases and therefore do not incorporate local mesh-derived interactions (e.g. Huang et al. 2018). Geometric connectomes (Pang et al. 2023) use only mesh connections. It might be useful to follow this nomenclature and discuss how this specific work bridges the gap between the 2 approaches using a millimeter resolution of white-matter tracing.

References:

Atasoy, Selen, Isaac Donnelly, and Joel Pearson. "Human Brain Networks Function in Connectome-Specific Harmonic Waves." *Nature Communications* 7 (January 21, 2016): 10340. <https://doi.org/10.1038/ncomms10340>.

Huang, W., T. A. W. Bolton, J. D. Medaglia, D. S. Bassett, A. Ribeiro, and D. Van De Ville. "A Graph Signal Processing Perspective on Functional Brain Imaging." *Proceedings of the IEEE* 106, no. 5 (May 2018): 868–85. <https://doi.org/10.1109/JPROC.2018.2798928>.

Naze, Sébastien, Timothée Proix, Selen Atasoy, and James R. Kozloski. "Robustness of Connectome Harmonics to Local Gray Matter and Long-Range White Matter Connectivity Changes." *NeuroImage* 224 (January 1, 2021): 117364. <https://doi.org/10.1016/j.neuroimage.2020.117364>.

Pang, James C., Kevin M. Aquino, Marianne Oldehinkel, Peter A. Robinson, Ben D. Fulcher, Michael Breakspear, and Alex Fornito. "Geometric Constraints on Human Brain Function." *Nature* 618, no. 7965 (June 2023): 566–74. <https://doi.org/10.1038/s41586-023-06098-1>.

Response:

Thanks for the reviewer's suggestions. We have modified the term harmonics to eigenmodes. Then, we discussed the connection and difference between cellular connectome (CC)-eigenmodes, dMRI-structural connectome (SC)-eigenmodes, and geometric eigenmodes.

(1) The discrepancy in nomenclature between connectome harmonics and eigenmodes

Atasoy et al. pioneered the concept of "connectome harmonics", which are calculated as the eigenvector of connectome Laplacian of the combined local links between mesh vertices (local gray matter) and white matter SC reconstructed from dMRI (long-range white matter) (Atasoy et al., 2016 & 2018; Naze et al., 2021). Because "connectome

eigenmodes” are exclusively determined using parcellated brain atlases and therefore do not incorporate local mesh-derived interactions (Wang et al., 2017; Huang et al., 2018; Robinson et al., 2021). More recently, Pang et al. have discovered frequency-specific representations of brain's shape, dubbed “geometric eigenmodes” (Pang et al., 2023). However, there is a distinction in nomenclature between connectome harmonics and eigenmodes.

The concept of eigenmodes describes the inherent vibration or wave pattern of a system. Harmonics, on the other hand, depicts waves or vibrations at a specific frequency, which is generally an integer multiple of the original frequency. Eigenmodes can contain harmonics in some instances, but they are not equivalent. “Connectome eigenmodes” of SC have been referred to as “connectome harmonics” (Prete et al., 2019; Rué-Queralt et al., 2021). However, harmonics imply integer frequency ratios, which are not necessarily guaranteed for brain-derived modes (Pang et al., 2023). As a result, the term “Connectome eigenmodes” was used in the revision.

(2) The performance of marmoset’s CC eigenmodes vs marmoset’s SC eigenmodes

In reconstructing marmosets' cortical activity and functional connectivity (FC), we compared the reconstruction accuracy of CC eigenmodes against SC eigenmodes. We reconstructed SC matrices using probabilistic diffusion tractography. A group SC matrix was generated by averaging all individual SC matrices. Because the connection density of CC and SC differed (the density of CC was 0.6842; the density of SC was 0.9912), we thresholded the SC matrix to construct a connection matrix that matched the density of CC (Fig. R1). The CC has proven to be an indispensable tool for mapping long-distance connections and obtaining the direction of information flow (Majka et al., 2016 & 2019). The SC was transmitted between cortical areas either via linkage of the dense local connections or rare, extraordinarily privileged long-range connections (Rosen et al., 2022). One limitation of dMRI was the difficulty in achieving fine enough spatial resolution to map small but essential fiber pathways.

Furthermore, the SC eigenmodes were derived from the normalized undirected graph Laplacian according to previous methods (Huang et al., 2018; Prete et al., 2019; Rué-Queralt et al., 2021) (Fig. S6a). Although the CC eigenmodes account for the projection directionality and edge density, they do not reveal local gray matter connections and white matter fiber pathways. The SC eigenmodes do not take fiber directionality into account, but they capture white matter connections measured with dMRI (Naze et al., 2021; Pang

et al., 2023). A direct comparison of the reconstruction accuracy of two distinct basis sets revealed that CC eigenmodes numerically outperform SC eigenmodes in capturing cortical activity (Fig. S6b) and FC (Fig. S6c). Hence, the CC eigenmodes provided a more parsimonious description of cortical activity than the SC eigenmodes.

A prior mouse research discovered that the CC provides a better ability to shape brain function than SC (Melozzi et al., 2019), owing to the fact that neural tracing may reveal information about the direction of information flow and long-distance connections.

The relevant statement has been included in the revision, see “**CC eigenmodes constrain on marmoset cortical activity**”.

Page 8 in the main text (Lines 153-157):

“We next compared the reconstruction accuracy of CC eigenmodes against dMRI-based SC eigenmodes (Fig. S6). A direct comparison of the reconstruction accuracy of two distinct basis sets revealed that CC eigenmodes numerically outperform the SC eigenmodes in capturing cortical activity and FC. The CC eigenmodes provided a more parsimonious description of cortical activity than the SC eigenmodes.”

Fig. R1. (This figure is just shown in response to reviewers). **Marmoset tracer-based cellular connectome (CC) (a) and dMRI-based structural connectome (SC) (b) matrix.**

Fig. S6. Cellular connectome (CC) eigenmodes were compared to structural connectome (SC) eigenmodes. **a** The parts of SC eigenmodes were projected onto the marmoset brain surface. Blue–white–red colors represented negative–zero–positive values. Despite certain similarities, the spatial patterns of SC eigenmodes differed from CC eigenmodes. **b** Reconstruction accuracy of marmoset cortical activity achieved by SC and CC eigenmodes. Reconstruction accuracy was quantified as the energy ratio between empirical and reconstructed cortical activity. **c** Reconstruction accuracy (i.e., Pearson correlation coefficients between empirical and reconstructed marmoset FC matrices across edges) of marmoset FC achieved by SC and CC eigenmodes. The green line indicates the reconstruction accuracy of the empirical SC graph. The orange shading lines (upper) indicate the reconstruction accuracy using eigenmodes derived from rewired SC graphs (1000 repetitions) to reconstruct unperturbed activity. The purple shading lines (middle) indicate the reconstruction accuracy using empirical SC eigenmodes to reconstruct permuted functional activity by randomly permuting the raw rs-fMRI time series across brain regions (1000 repetitions). The red line (bottom) indicates the reconstruction accuracy of the empirical CC graph.

(3) Discuss how the current work bridges the gap and two approaches

Previous studies have extensively utilized eigenmode approaches to understand human brain function (Atasoy et al., 2016; Huang et al., 2018; Pang et al., 2023). The SC eigenmodes obtained from dMRI-based undirected networks capture local gray matter and white

matter fiber connections, which serve as the foundation for human functional networks (Atasoy et al., 2016 & 2018; Preti et al., 2019; Rué-Queralt et al., 2021). Furthermore, geometric eigenmodes derived from the brain's geometry (e.g., its shape) capture local spatial relations, representing the underlying anatomical restrictions on human brain function (Pang et al., 2023). Our work with marmosets utilized a similar framework, but the underlying anatomical properties differed (dMRI-based SC, geometric shape, tracer-based CC). The neuroanatomical tracing connectome represents an asymmetric (directed) network. Graph signal models based on asymmetric network operators may be better for signal and information processing on directed projections. The CC eigenmodes produced from Laplacian's normalized directed graph may capture the projection directionality and edge density, emphasizing the importance of projection directionality information.

The revision has been updated accordingly:

Page 17 in the main text (Lines 314-326):

“Previous studies have extensively utilized eigenmode approaches to understand human brain function^{15-17,19,67}. The SC eigenmodes obtained from dMRI-based undirected networks capture local gray matter and white matter fiber connections, which serve as the foundation for human functional networks¹⁶⁻¹⁸. Furthermore, geometric eigenmodes derived from the brain's geometry (e.g., its shape) capture local spatial relations, representing the underlying anatomical restrictions on human brain function¹⁵. Our work with marmosets used a similar approach, but the fundamental anatomical features differed. The neuroanatomical tracing connectome represents an asymmetric (directed) network. Graph signal models based on asymmetric network operators may be better for signal and information processing on directed projections⁴⁴. The CC eigenmodes produced from Laplacian's normalized directed graph may capture the projection directionality and edge density, emphasizing the importance of projection directionality information.”

Q3: p7: A note on polarities: how was the polarity of each eigenvector assigned?

In the case of symmetric connectomes, therefore with only positive eigenvalues, the sign of the eigenvector is rather arbitrary. How does it differ in the case of asymmetric connectome presented here?

Response:

The eigenvalues of a symmetric matrix are positive in the case of symmetric

connectomes, and the accompanying eigenvectors are real and orthogonal to each other. Because any constant can scale the eigenvectors, the signs (polarities) of the eigenvectors are arbitrary.

A directed graph is naturally represented by a non-symmetric adjacency matrix. The graph Laplacian provides a bridge between spectral graph theory and signal processing. It is no longer appropriate to use the graph Laplacian directly for directed graphs. The conventional undirected graph Laplacian (normalized $\mathcal{L}_{undir} = I - D^{-\frac{1}{2}}AD^{-\frac{1}{2}}$, where A is the adjacency matrix, and D is the diagonal matrix of degrees of A) is invalid because its symmetries are no longer verified. To expand the Laplacian-based Fourier analysis from undirected graphs to directed graphs, a suitable reference operator must be found.

To avoid the complications of non-diagonalizable matrices, Chung (Chung, 2005) defined a symmetric normalized Laplacian matrix for directed graphs with strongly connected (There is a path in each direction between each pair of vertices of the graph). This Laplace operator is defined as a self-adjoint operator using the transition probability operator and the Perron vector. We transform a weighted directed cellular connectome (CC) matrix into a symmetric normalized directed Laplacian matrix by the random walk operator (Chung, 2005). The eigenvalues derived from the eigen-decomposition of the normalized directed Laplacian are positive, i.e., $0 \leq \lambda_1 \leq \lambda_2 \leq \dots \leq \lambda_N$ (Chung, 2006). It has a complete set of orthonormal eigenvectors associated with eigenvalues. As a result, the signs (polarities) of the eigenvector are quite arbitrary. The CC eigenvectors (eigenmodes) measured smoothly varying patterns across the marmoset's cortices between positive and negative polarities.

Q4: p12, l.209-210: the reported statistics in text do not match the ones in the figures.

Response:

Thanks for the reviewer's careful checks and corrections. We feel sorry for our carelessness and for miswriting reported statistics. The revision has been updated accordingly:

Page 13 in the main text (Lines 237-239):

“Regional CFD was strongly negatively correlated with myelin content⁵⁷ ($\rho = 0.40$, $P_{SMASH} = 0.007$) and neuronal counts⁵⁸ ($\rho = 0.43$, $P_{SMASH} = 0.008$).”

Q5: p.12: strong cellular-functional dependency should rather be aligned/liberal than liberal/aligned, I suggest renaming it cellular-functional decoupling throughout the manuscript, or it be renamed CFI: cellular-functional independence.

Response:

Thanks for the reviewer's idea. To characterize the cellular-functional decoupling (CFD) of each brain area, we measured the binary logarithm form of the energy ratio of high-frequency (cortical activity decoupled to cellular connectome (CC)) to low-frequency (cortical activity coupled to CC) components. The CFD was used to assess the degree of local (de)coupling between cortical activity and underlying monosynaptic connections. Lower CFD implied a strong coupling of cortical activity to neural connections, whereas higher CFD indicated the reverse. "Cellular-functional dependency" has been renamed "cellular-functional decoupling" throughout the manuscript. The revision has been amended as follows:

Page 29 in the main text (Lines 594-601):

"The cellular-functional decoupling (CFD) of each brain area can be quantified as the ratio between the norms (L_2 -norm) of high-frequency versus low-frequency components across time points¹², resulting in

$$CFD_{v_i} = \log_2\left(\frac{\|f_{v_i}^{high}(t)\|_2}{\|f_{v_i}^{low}(t)\|_2}\right) \quad (6)$$

We chose the binary logarithmic form of this index, so CFD values of 0 represent a perfect balance between cellular-functional coupling and decoupling. CFD values around -1 imply that cortical activity is strongly coupled to underlying monosynaptic connections, whereas CFD values around 1 indicate the reverse."

Q6: p14: 1.250-252: "According to these results, the marmoset CC harmonics concisely and parsimoniously describe human resting-state cortical activity and connectivity" may be overstated based on only 11 mapped regions.

Response:

Predefined homologous areas across species were used to generalize marmoset-derived eigenmodes to the human brain. Because marmosets have only recently gained traction as an animal model, established human-marmoset homologous areas have been lacking (Ngo et al., 2023). Our study was based on a modest sample size of 11 mapped areas. As a

result, we reduced the inferences and acknowledged the shortcomings of the methodology. The revision has been amended as follows:

Page 15 in the main text (Lines 277-279):

“These findings suggest that CC eigenmodes derived from marmosets may help estimate human spontaneous brain activity and FC of homologous areas.”

Page 22 in the main text (Lines 423-425):

“Fourth, our research was based on a restricted number of mapped areas due to the absence of documented marmoset-human homologous areas.”

Q7: p.17: l.308-318: the reason of having the OFC and IPFC in both aligned and liberal components might be due to normalizing the BOLD time series (see first comment about methods). This normalization has consequences when computing the energy $|w_i|^2$ present in each mode. This should be discussed.

l.425-426: normalizing time series have implication for the GSP analysis, particularly when computing nodal energy after graph filtering.

Response:

Thanks for your helpful suggestions. Normalization is crucial in GSP because it ensures that time series or signals on nodes have a consistent scale. It is necessary for fair comparisons and to avoid signals of larger magnitude from dominating the analysis. Thus, we normalized the BOLD-fMRI time series by subtracting its temporal mean. To investigate the spatial significance of the cortical class-level low- and high-frequency components, we used a nonparametric permutation test. Then, we re-discussed the patterns of low- and high-frequency components.

(1) The patterns of low- and high-frequency components

We examined how CC eigenmodes constrain cortical activity. To this end, we divided the CC eigenmodes into low- and high-frequency filters. The BOLD-fMRI amplitudes for each time point were spatially filtered by low- and high-frequency eigenmodes, resulting in low- and high-frequency components that characterized to what degree BOLD-fMRI fluctuations were strongly or weakly constrained by the underlying monosynaptic connections.

We further performed a nonparametric permutation test to examine the spatial significance of the cortical class-level activity concentrations. We generated 1000 permutations for low- and high-frequency components using graph spectral randomization (SR) (Pirondini et al., 2016; Huang et al., 2018) and computed a null distribution of mean activity concentrations for each cortical class. The mean activity concentration was greater than 95% (> 95th percentile) of the null permutations, which were identified to be significantly concentrated in a given class. Significance was set at $P_{SR} < 0.05$ with false discovery rate (FDR)-correction for multiple comparisons across eleven cortical classes.

Cortical activity constrained by low-frequency eigenmodes was concentrated within dorsal-posterior cortices such as the visual (VC), somatosensory (SS), posterior parietal cortex (PPC), and posterior cingulate, medial, and retrosplenial cortices (PCC) (all $P_{SR} < 0.05$, FDR-corrected; Fig. 2c). The high-frequency components were concentrated in ventral-anterior (frontopolar-temporal) cortices, including the medial prefrontal cortex (mPFC), dorsolateral prefrontal cortex (dlPFC), ventrolateral prefrontal cortex (vlPFC), orbitofrontal cortex (OFC), lateral inferior temporal cortex (LIT), and auditory (AU) (all $P_{SR} < 0.05$, FDR-corrected; Fig. 2d). In summary, the patterns of CC eigenmodes constrain cortical activity were circumscribed by specific brain systems.

In the revision, we have added the relevant statement, see “**Statistical analysis**” and “**CC eigenmodes constrain on marmoset cortical activity**”.

Page 9 in the main text (Lines 168-177):

“Cortical activity constrained by low-frequency eigenmodes was concentrated within dorsal-posterior cortices such as the visual cortex (VC), somatosensory cortex (SS), posterior parietal cortex (PPC), and posterior cingulate, medial, and retrosplenial cortices (PCC) (all $P_{SR} < 0.05$, FDR-corrected; **Fig. 2c**). The high-frequency components were concentrated in ventral-anterior (frontopolar-temporal) cortices, including the medial prefrontal cortex (mPFC), dorsolateral prefrontal cortex (dlPFC), ventrolateral prefrontal cortex (vlPFC), orbitofrontal cortex (OFC), lateral inferior temporal cortex (LIT), and auditory cortex (AU) (all $P_{SR} < 0.05$, FDR-corrected; **Fig. 2d**). In summary, the patterns of CC eigenmodes constrain cortical activity were circumscribed by specific brain systems.”

Fig. 2. b Marmoset cortical classes were parceled according to the 3D digital Paxinos atlas. **c, d** The spatial patterns of low- and high-frequency components. The low- and high-frequency components from each of the eleven cortical classes are presented as box plots ordered by the median value. Outliers are shown by the gray circles. Asterisks denote statistically significant activity concentration in each cortical class compared to the null permutation distributions (1000 repetitions, $*P_{SR} < 0.05$, one-tailed, FDR-corrected). AU = Auditory cortex, dlPFC = dorsolateral prefrontal cortex, LIT = lateral and inferior temporal cortex, mPFC = medial prefrontal cortex, MOT = motor and premotor cortex, OFC = orbitofrontal cortex, PCC = posterior cingulate, medial, and retrosplenial cortex, PPC = posterior parietal cortex, SS = Somatosensory cortex, vlPFC = ventrolateral prefrontal cortex, VC = Visual cortex.

(2) Discussion of the patterns of low- and high-frequency components

Our findings showed that monosynaptic connections strongly constrained cortical activity, particularly in the dorsal-posterior cortices, including VC, SS, PPC, and PCC, which were involved in sensation, perception, and action processes (Fukushima et al., 2019). Furthermore, weakly constrained cortical activity was mostly concentrated within frontopolar-temporal cortices, potentially requiring more diversified functional communication relative to the underlying wiring diagram (Buckner et al., 2019; Watakabe et al., 2023). Notably, the activity of the posterior core regions (PPC and PCC) in the default mode network (DMN) (Liu et al., 2023) were strongly constrained by connections, whereas

the activity of the anterior DMN (dIPFC) was weakly restricted by connections, indicating that the DMN architecture in marmosets may have a divergent anterior-posterior axis. According to a recent cross-species study, the anterior DMN regions exhibited weak connections and spatially irregular connection topology when compared with the posterior DMN areas (Ngo et al., 2023). Consequently, the patterns of marmoset brain activity constrained by CC eigenmodes were confined by specific brain systems.

In the revision, we have added the relevant statement, see “**Discussion**”.

Page 18 in the main text (Lines 339-352):

“Our findings showed that monosynaptic connections strongly constrained cortical activity, particularly in the dorsal-posterior cortices such as the VC, SS, PPC, and PCC, which were engaged in sensation, perception, and action processes⁶⁹. Furthermore, weakly constrained cortical activity was primarily concentrated within frontopolar-temporal cortices, potentially requiring more diversified functional communication relative to the underlying wiring diagram^{60,70}. Notably, the activity of the posterior core regions (PPC and PCC) in the default mode network (DMN)⁴⁵ was strongly constrained by connections, whereas the activity of the anterior DMN (dIPFC)⁴⁵ was weakly restricted by connections, indicating that the DMN architecture in marmosets may have a divergent anterior-posterior axis. According to a recent cross-species study, the anterior DMN areas have weak connections and spatially irregular connection topology compared to the posterior DMN areas⁶¹. Consequently, the patterns of marmoset brain activity constrained by CC eigenmodes were confined by specific brain systems.”

Q8: 1.363-365: “We found that marmoset’s CC harmonics provided a more parsimonious representation of human cortical activity than human’s dMRI-based SC harmonics”: overstated when based on only 11 regions.

Response:

Thanks for the reviewer's suggestions. Because human dMRI-based structural connectivity (SC) eigenmodes are inherently noisier, comparing marmoset CC eigenmodes to human SC eigenmodes would be unfair. As a result, we did not contemplate comparing marmoset’s CC eigenmodes and human’s SC eigenmodes to reconstruct human activity patterns in the revision.

Predefined homologous areas across species were used to generalize marmoset-derived

eigenmodes to the human brain. Because marmosets have only recently gained traction as an animal model, established human-marmoset homologous areas have been lacking (Ngo et al., 2023). Our study was based on a modest sample size of 11 mapped areas. As a result, we reduced the inferences and acknowledged the shortcomings of the methodology. The revision has been amended as follows:

Page 21 in the main text (Lines 397-403):

“The identification of probable homologous landmarks has been used to make cross-species comparisons^{51,61,74}. Our results showed that marmoset’s CC eigenmodes may reflect human spontaneous cortical activity in homologous areas, which was not driven by general mathematical properties of basis set expansions but could be derived by some biologically relevant evolutionary process. These findings suggested that CC may be used to restrict brain function across species.”

Q9: 1.377-386: the limitation section could tackle technical aspects of the calculation of the Laplacian on directed graph, and the role of white matter vs. gray matter connectivity in formulating the connectome.

Response:

The normalized directed graph Laplacian is suitable for strongly connected graphs to ensure that the random walk’s transition probability matrix is ergodic. If the directed network was not strongly connected, the PageRank algorithm (Langville et al., 2006) may be utilized (Sevi et al., 2023; Seabrook et al., 2023).

Although retrograde tracer accurately maps cellular-level connectome by staining cell bodies, it cannot reveal local gray matter connectivity and white matter axonal pathways (Atasoy et al., 2016; Naze et al., 2021). Future studies could integrate neuronal tracing data into the cortical surface mesh and fiber tractography (Liu et al., 2020). The revision has been amended as follows:

Page 21 in the main text (Lines 416-425):

“Second, although retrograde tracer accurately maps cellular-level connectome by staining cell bodies, it cannot reveal local gray matter connectivity and white matter axonal fibres^{17,18}. Neuronal tracing data might be integrated into cortical surface mesh and fiber tractography in future investigations³⁹. Another caution is that the normalized directed

graph Laplacian is suitable for strongly connected graphs, guaranteeing that the random walk's transition probability matrix is ergodic⁴⁷. If the directed network was not strongly connected, the PageRank algorithm⁷⁵ may be utilized^{48,49}."

Methods:

Q10: 1.464: subscript i is not defined, would be better defining node activity v_i .

Response:

The subscript v_i denotes the node. The revision has been updated accordingly:

Page 28 in the main text (Lines 560-561):

"The cortical activity over the node v_i at time-point t was denoted as $f_{v_i}(t) \in R^{N \times T}$, $i = \{1, 2, \dots, N\}, t \in \{1, 2, \dots, T\}$."

Q11: 1.485-486: would read better "cortical activities can be isolated into aligned and liberal components".

Response:

Thanks for the reviewer's suggestions. As Reviewer #3 Q17 suggests, using the labels "liberal" and "aligned" is confusing, and it is best to stay with low- and high-frequency signals. The revision has been amended as follows:

Page 28 in the main text (Lines 581-583):

"Given a graph signal $f_{v_i}(t)$ with graph spectral coefficients $w_k(t)$, cortical activities can be isolated into low-frequency components (coupled to the CC) and high-frequency components (decoupled from the CC)^{12,22}."

Q12: 1.488: spectral => spectra.

Response:

We apologized for the wrong use of words. The revision has been updated accordingly:

Page 29 in the main text (Lines 584-586):

"To determine the cut-off frequency C , we used the graph spectrum dichotomization method¹² to divide the graph spectra into two parts with equal energy based on average

ESD (across time and subjects).”

Q13: 1.496-500: related to p.12 above, I find the naming of the metric counter-intuitive and make the interpretation of the results convoluted. Either call the metric cellular-function decoupling, or use aligned/liberal components so that a larger CFD would mean a larger structure-function coupling.

Response:

Thanks for the reviewer's idea. To characterize the cellular-functional decoupling (CFD) of each brain area, we measured the binary logarithm form of the energy ratio of high-frequency (cortical activity decoupled to cellular connectome (CC)) to low-frequency (cortical activity coupled to CC) components. The CFD was used to assess the degree of local (de)coupling between cortical activity and underlying monosynaptic connections. Lower CFD implied a strong coupling of cortical activity to neural connections, whereas higher CFD indicated the reverse. "Cellular-functional dependency" has been renamed "cellular-functional decoupling" throughout the manuscript. The revision has been amended as follows:

Page 29 in the main text (Lines 594-601):

“The cellular-functional decoupling (CFD) of each brain area can be quantified as the ratio between the norms (L_2 -norm) of high-frequency versus low-frequency components across time points¹², resulting in

$$CFD_{v_i} = \log_2 \left(\frac{\|f_{v_i}^{high}(t)\|_2}{\|f_{v_i}^{low}(t)\|_2} \right) \quad (6)$$

We chose the binary logarithmic form of this index, so CFD values of 0 represent a perfect balance between cellular-functional coupling and decoupling. CFD values around -1 imply that cortical activity is strongly coupled to underlying monosynaptic connections, whereas CFD values around 1 indicate the reverse. ”

Q14: 1.504: connective => connectivity

Response:

We apologize for the wrong use of words. The revision has been updated accordingly:

Page 30 in the main text (Lines 605-606):

“We further analyzed its relation to the cortical microstructural attributes and connectivity gradients.”

Supplementary materials:

Q15: Suppl. Materials p.4, eq.(1) : please add textbook reference for this equality.

Response:

We feel sorry for not adding textbook references to Supporting Materials page 4, Eq. (1). The transition matrix $P \in R^{N \times N}$ (Lovász, 1993; Spielman, 2012) is equal to $P = D^{-1}A$, where $D = \sum_{j=1}^N A_{ij}$, denotes the diagonal matrix of the out-degrees of the adjacency matrix A . We have added textbook references for this equality.

Page 4 in the Supporting information:

“From the point of view of graph theory, the transition matrix $P \in R^{N \times N}$ ^{9,10} is equal to

$$P = D^{-1}A, \quad (1)$$

where $D = D_{ii} = \sum_{j=1}^N A_{ij}$, denotes the diagonal matrix of the out-degrees of the adjacency matrix A .”

Q16: Suppl. Materials, p.5, is there any test or condition on P that guarantee that Xi is ergodic (irreducible and aperiodic)?

Response:

We defined a Markov chain $\chi = (X_n)_{n \geq 0}$ with transition matrix P , $P = D^{-1}A$, where $D = \sum_{j=1}^N A_{ij}$ denotes the out-degrees matrix of the adjacency matrix A , $A = \{A_{ij}\}_{1 \leq i, j \leq N} \in R_+^{N \times N}$ is defined as the FLNe value of CC in marmosets. For a strongly connected directed graph \mathcal{G} , the Markov chain χ with transition matrix P is ergodic (irreducible and aperiodic), then it has a unique stationary distribution π (Perron vector of P), i.e., $\pi P = \pi$, with $\sum_{i=1}^N \pi(v_i) = 1, \pi(v_i) \geq 0$ (Sevi et al., 2023; Seabrook et al., 2023).

To guarantee that the Markov chain χ with transition matrix P is ergodic (irreducible and aperiodic), the directed graph needs to be strongly connected graph, that is, there is a (directed) path from any node $v_i \in V$ to any node $v_j \in V$ (Sevi et al., 2023; Seabrook et al., 2023). Below, we provided the detailed process to verify the ergodicity of transition matrix P .

A Markov chain χ is ergodic if it is irreducible and aperiodic (Gallager, 2013). This

condition is equivalent to the transition matrix being a primitive nonnegative matrix. By Wielandt's theorem (Wielandt, 1950), a Markov chain χ is ergodic if and only if all elements of P^s are positive for $s = (N - 1)^2 + 1$, where N is the number of nodes of a directed graph. We used the Matlab function “*isergodic*” to determine the ergodicity of the random walk χ (<https://ww2.mathworks.cn/help/econ/dtmc.isergodic.html>). Visually confirm that the Markov chain χ is ergodic by plotting its eigenvalues on the complex plane (Fig. S12).

By the Perron-Frobenius Theorem (Hom et al., 1985), if the random walk χ is ergodic with a diagonalizable transition matrix P , the diagonalization of P admits a simple dominant eigenvalue $\lambda_{\max} = 1$, and an accompanying nonnegative left eigenvector that normalizes to a unique stationary distribution. All other eigenvalues have a modulus less than 1, which means that all eigenvalues different of λ_{\max} lie within the unit circle (Sevi et al., 2023). Therefore, we verified that the Markov chain $\chi = (X_n)_{n \geq 0}$ with transition matrix P defined on a directed weighted graph $\mathcal{G} = (V, E, A)$ is ergodic (irreducible and aperiodic).

In the revision, we have added the relevant verification, see “**Supplementary Result S5 Verify the ergodicity of the transition probability matrix**”.

Fig. S12. The eigenvalues of the transition matrix of Markov chain χ . The eigplot object function plots the eigenvalues of the transition matrix P . The plot highlights the unit circle, the Perron-Frobenius eigenvalue at $(1,0)$, the circle of second largest eigenvalue magnitude, and the spectral gap between the two circles.

Q17: Suppl. Materials p.5, eq.(2) : Please expand on how π is calculated and how it can be interpreted. This normalized Laplacian formula adds P and its transpose, (scaled by the Perron vector π), therefore making it symmetric, why is this necessary? Adding a visualisation of π (e.g. a bar plot) could be informative to some readers.

Response:

The graph Laplacian connects spectral graph theory and signal processing. The undirected graph Laplacian (normalized $\mathcal{L}_{undir} = I - D^{-\frac{1}{2}}AD^{-\frac{1}{2}}$, where A is the adjacency matrix, and D is the diagonal matrix of degrees of A) is a symmetric and positive semi-definite operator (Chung, 1997). According to the spectral theorem, Laplacian's undirected graph admits a set of eigenvectors forming an orthonormal basis. These eigenvectors can be thought of as Fourier modes, accompanying eigenvalues related to the notion of frequency (Chung, 1997).

The random walk operator on graphs is introduced as an appropriate reference operator for expanding the signal processing framework to directed graphs (Sevi et al., 2023). The random walk operator is related to the concept of diffusion on graphs, which may transform any graph into a Markov chain (Seabrook et al., 2023). To avoid the complications of non-diagonalizable matrices, Chung (Chung, 2005) defined a symmetric normalized Laplacian matrix for directed graphs with strongly connected. This definition is inspired by the fact that the degree matrix D is used in the definition of the undirected Laplacian. The D is the diagonal matrix, which is proportional to the limiting distribution of a random walk on an undirected graph (Spielman, 2012; Seabrook et al., 2023; Sevi et al., 2023). As a result, the normalized directed graph Laplacian is constructed by replacing D with the Perron vector for the random walk with the transition matrix P on the graph (Chung, 2005; Spielman, 2012).

The stationary distribution π of the random walk with the transition matrix P can be obtained by calculating the Perron vector of P . The Perron vector π can be calculated using the Matlab function "eigs" (Fig. S13). The detailed procedure for calculating the normalized directed graph Laplacian is shown in Algorithm S1. An example of computing normalized digraph Laplacian is shown in Fig.S3. The source code for this method is available at https://epfl-lts2.github.io/gspbox-html/doc/utils/gsp_create_laplacian.html. The stationary distribution reflects that the distribution of states converges to a stable distribution as the Markov chain progresses, which provides insights into the system's long-term behavior.

The normalized directed graph Laplacian, which has been used in spectral clustering, graph embedding, and classification applications, may represent the graph's directionality and edge density by leveraging the random walk operator. Notably, the resulting Laplace matrix is symmetric. This approach assures that eigenvectors derived from the Laplacian matrix's eigen-decomposition form a set of orthogonal basics. The Dirichlet energy of a given eigenvector of the random walk operator on a directed graph is associated with eigenvalue, which can be regarded as graph frequencies (Sevi et al., 2023). As a result, eigenvectors are regarded as a suitable Fourier basis for functions defined on directed graphs.

In the revision, we have added the relevant statement, see “**Connectome Laplacian and eigenmodes of the CC**”, and Supporting information “**Normalized directed graph Laplacian**”.

Algorithm S1: Normalized directed graph Laplacian

Input: $A \in R_+^{N \times N}$: weighted directed adjacency matrix

Output: L : normalized directed graph Laplacian

- 1: Compute the diagonal matrix D of the out-degrees of A .
 - 2: Compute the transition matrix P of the random walk, see Eq.(1).
 - 3: Compute the stationary distribution π , i.e. $\pi P = \pi$, with $\sum_{i=1}^N \pi(v_i) = 1, \pi(v_i) \geq 0$. π can be obtained by calculating the Perron vector of P , which can be calculated by use Matlab function “*eigs*”.
 - 4: Calculate the diagonal matrix $\Pi = \text{diag}\{\pi(v_1), \dots, \pi(v_N)\}$ of the stationary distribution π .
 - 5: Calculate the normalized directed graph Laplacian L , see Eq. (2).
 - 6: Return a symmetric matrix L .
-

Fig. S3. An example of computing normalized digraph Laplacian. First, a directed graph is represented an asymmetric weighted adjacency matrix. Second, compute the diagonal matrix D of the out-degrees of A . Third, compute the transition matrix P of the random walk $\chi = (X_n)_{n \geq 0}$. Fourth, compute the stationary distribution π , which can be obtained by calculating the Perron vector of P . Finally, calculate the normalized directed graph Laplacian L .

Fig. S13. Visualization Perron vector π of the transition matrix P of Markov chain χ . The Perron vector π of P can be calculated using Matlab function “eigs”.

Q18: Supplementary Results (p.8-9) and associated Suppl. Figs 8 & 9 are not referred anywhere in main text. How are they relevant?

Response:

In the revision, we moved the Supplementary Result (p.8-9, The neuronal tracing connectome informs cortical activity and connectivity) and associated Fig. S8 from the Supporting information into the main text. Since Fig. S9 is unrelated to the main content, we considered deleting it. In particular, we added the reconstruction results of the marmoset’s cortical activity and FC. Furthermore, we investigated the specificity of the basis sets for understanding brain function using two null benchmarks: rewired connectome and permuted functional activity across brain regions.

(1) CC eigenmodes constrain on marmosets’ brain function

We assessed the degree to which CC eigenmodes may explain cortical activity observed in BOLD-fMRI data from marmosets. We began by decomposing cortical activity into a combination of orthogonal CC eigenmodes. We next tested the accuracy of CC eigenmodes in capturing spontaneous cortical activity using this decomposition. We reconstructed the spatial activity map at each time point with an increasing CC eigenmodes, and then generated a region-region FC matrix.

(2) The specificity of CC eigenmodes to explain cortical activity

We further investigated the specificity of CC eigenmodes to explain brain function (Faskowitz et al., 2023; Pang et al., 2023). First, to illustrate the significance of the specific topology of CC for capturing empirical cortical activity, we derived eigenmodes from

rewired connectomes that preserve the weight, degree, and strength distributions of the CC graph (Rubinov et al., 2010; Váša et al., 2022) (1000 repetitions). Furthermore, to test the specificity of CC eigenmodes in representing observed brain activity, we generated permuted functional activity by randomly permuting the raw rs-fMRI time series across brain regions (Griffa et al., 2022) (1000 repetitions).

(3) Results

We found that increasing CC eigenmodes improved the reconstruction accuracy of marmosets' cortical activity (Fig. 2a). Using all CC eigenmodes ($N = 55$), the cortical activity could be entirely reconstructed. The first 20 CC eigenmodes achieved 83% reconstruction accuracy of cortical activity, which was higher than utilizing both rewired CC graph ($P_{\text{rewired}} < 0.001$, false discovery rate (FDR)-corrected) and cortical activity of permuted areas ($P_{\text{perm}} < 0.001$, FDR-corrected). Similarly, the first 20 CC eigenmodes achieved 88% reconstruction accuracy of functional connectivity (FC) in marmosets (Fig. 2b). These findings suggest that CC eigenmodes canonically serve as a foundation for a compact description of spontaneous cortical activity and FC.

In the revision, we have added the relevant statement, see “**CC eigenmodes constrain on marmoset cortical activity**”.

Page 8 in the main text (Lines 139-152):

“We assessed the extent to which CC eigenmodes may explain brain activity observed in BOLD-fMRI data from marmosets. We began by decomposing cortical activity into a combination of orthogonal CC eigenmodes (Fig. 1d). We next tested the accuracy of CC eigenmodes in capturing spontaneous cortical activity using this decomposition. We found that increasing CC eigenmodes improved the reconstruction accuracy of marmosets' cortical activity (Fig. 2a). Using all CC eigenmodes ($N = 55$), the cortical activity could be entirely reconstructed. The first 20 CC eigenmodes achieved 83% reconstruction accuracy of cortical activity, which was higher than utilizing both rewired CC graph^{52,53} ($P_{\text{rewired}} < 0.001$, false discovery rate (FDR)-corrected) and cortical activity of permuted areas⁵⁴⁻⁵⁶ ($P_{\text{perm}} < 0.001$, FDR-corrected). Similarly, the first 20 CC eigenmodes achieved 88% reconstruction accuracy of functional connectivity (FC) in marmosets (Fig. S5). These findings suggest that CC eigenmodes canonically serve as a foundation for a compact description of spontaneous cortical activity and FC.”

Fig. 2. Cellular connectome (CC) eigenmodes constrain cortical activity in marmosets. a Cortical activity reconstruction accuracy. Reconstruction accuracy was quantified as the energy ratio between empirical and reconstructed cortical activity. **b** FC reconstruction accuracy in marmosets (i.e., Pearson correlation coefficients between empirical and reconstructed marmoset resting-state FC matrices across edges). The red line indicates the reconstruction accuracy of the empirical CC graph. The blue shading lines (*left*) indicate the reconstruction accuracy using eigenmodes derived from rewired CC graphs (1000 repetitions). The green shading lines (*right*) indicate the reconstruction accuracy using empirical CC eigenmodes to reconstruct permuted functional activity by randomly permuting the raw rs-fMRI time series across brain regions (1000 repetitions).

Bibliographies in the Response to Reviewer #1:

- Atasoy, S., et al., (2016). Human brain networks function in connectome-specific harmonic waves. *Nature communications*, 7(1), 10340.
- Atasoy, S., et al., (2018). Harmonic brain modes: a unifying framework for linking space and time in brain dynamics. *The Neuroscientist*, 24(3), 277-293.
- Buckner, R. L., & Margulies, D. S. (2019). Macroscale cortical organization and a default-like apex transmodal network in the marmoset monkey. *Nature communications*, 10(1), 1976.
- Chung, F. R. (1997). *Spectral graph theory* (Vol. 92). American Mathematical Soc.
- Chung, F. (2005). Laplacians and the Cheeger inequality for directed graphs. *Annals of Combinatorics*, 9, 1-19.
- Chung, F. (2006). The diameter and Laplacian eigenvalues of directed graphs. *Electronic journal of combinatorics*, 13, N4.
- Fukushima, M., et al., (2019). Neuroanatomy of the marmoset. In *The common marmoset in captivity and biomedical research* (pp. 43-62). Academic Press.
- Faskowitz, J., et al., (2023). Commentary on Pang et al.(2023) Nature. *bioRxiv*, Preprint at <https://doi.org/10.1101/2023.07.20.549785>.
- Gallager, R. G. (2013). *Stochastic processes: theory for applications*. Cambridge University Press.
- Griffa, A., et al., (2022). The evolution of information transmission in mammalian brain networks, *bioRxiv*, Preprint at <https://doi.org/10.1101/2022.05.09.491115>.
- Huang, W., et al., (2018). A graph signal processing perspective on functional brain imaging. *Proceedings of the IEEE*, 106(5), 868-885.
- Hom, R. A., & Johnson, C. R. (1985). *Matrix analysis*. Cambridge University Press, 455.
- Lovász, L. (1993). Random walks on graphs. *Combinatorics, Paul erdos is eighty*, 2(1-46), 4.
- Langville, A. N., & Meyer, C. D. (2006). *Google's PageRank and beyond: The science of search engine rankings*. Princeton university press.
- Liu, C., et al., (2019). Anatomical and functional investigation of the marmoset default mode network. *Nature communications*, 10(1), 1975.
- Liu, C., et al., (2020). A resource for the detailed 3D mapping of white matter pathways in the marmoset brain. *Nature neuroscience*, 23(2), 271-280.
- Majka, P., et al., (2016). Towards a comprehensive atlas of cortical connections in a primate brain: Mapping tracer injection studies of the common marmoset into a reference digital template. *Journal of Comparative Neurology*, 524(11), 2161-2181.
- Melozzi, F., et al., (2019). Individual structural features constrain the mouse functional connectome. *Proceedings of the National Academy of Sciences*, 116(52), 26961-26969.
- Majka, P., et al., (2020). Open access resource for cellular-resolution analyses of corticocortical connectivity in the marmoset monkey. *Nature Communications*, 11(1), 1133.
- Naze, S., et al., (2021). Robustness of connectome harmonics to local gray matter and long-range white matter connectivity changes. *NeuroImage*, 224, 117364.

- Ngo, G. et al., (2023). Joint-embeddings reveal functional differences in default-mode network architecture between marmosets and humans. *NeuroImage*, 272, 120035.
- Pirondini, E., et al., (2016). A spectral method for generating surrogate graph signals. *IEEE signal processing letters*, 23(9), 1275-1278.
- Preti, M. G., & Van De Ville, D. (2019). Decoupling of brain function from structure reveals regional behavioral specialization in humans. *Nature communications*, 10(1), 4747.
- Pang, J. C., et al., (2023). Geometric constraints on human brain function. *Nature*, 618, 566-574.
- Pang, J. C., et al., (2023). Reply to: Commentary on Pang et al.(2023) Nature. *bioRxiv*, Preprint at <https://doi.org/10.1101/2023.10.06.560797>.
- Rubinov, M., & Sporns, O. (2010). Complex network measures of brain connectivity: uses and interpretations. *Neuroimage*, 52(3), 1059-1069.
- Rué-Queralt, J., et al., (2021). The connectome spectrum as a canonical basis for a sparse representation of fast brain activity. *NeuroImage*, 244, 118611.
- Rosen, B. Q., & Halgren, E. (2022). An estimation of the absolute number of axons indicates that human cortical areas are sparsely connected. *PLoS Biology*, 20(3), e3001575.
- Spielman, D. (2012). Spectral graph theory. *Combinatorial scientific computing*, 18, 18.
- Sevi, H., et al., (2023). Harmonic analysis on directed graphs and applications: From Fourier analysis to wavelets. *Applied and Computational Harmonic Analysis*, 62, 390-440.
- Seabrook, E., & Wiskott, L. (2023). A Tutorial on the Spectral Theory of Markov Chains. *Neural Computation*, 35(11), 1713-1796.
- Váša, F., & Mišić, B. (2022). Null models in network neuroscience. *Nature Reviews Neuroscience*, 23(8), 493-504.
- Wang, M. B., et al., (2017). Brain network eigenmodes provide a robust and compact representation of the structural connectome in health and disease. *PLoS computational biology*, 13(6), e1005550.
- Wielandt, H. (1950). Unzerlegbare, nicht negative Matrizen. *Mathematische Zeitschrift*, 52(1), 642-648.
- Watakabe, A., et al., (2023). Local and long-distance organization of prefrontal cortex circuits in the marmoset brain. *Neuron* 111, 2121–2122.

Comments from the Reviewer #2 (5 points)

The paper investigates the relationship between structure and function in marmoset brains. For that purpose, the authors decompose the tract-tracing based connectome into harmonics, and look for the possibility to decode resting-state functional MRI brain activity using such basis. Overall, the results show that connectome harmonics effectively decode brain activity. The pattern of structure-function relationship recapitulates the hierarchical organization of the brain, i.e., a unimodal to transmodal gradient, which is observed across various properties (myelin, neuronal count, connectivity...). At the end, the authors make use of the homology with the human brain to demonstrate a potential general principle of cross-species harmonic constraints.

The manuscript treats an important topic in neuroscience. It is overall well-written, providing sufficient background. However, I have a couple of significant concerns about the analyses and interpretations that I hope the authors can address.

Response:

We greatly appreciate the reviewer's positive comments and constructive suggestions. We will address the points raised by the Reviewer in the below responses.

Q1: First at all, while the study seems seriously conducted, the results do not appear spectacular and new. It has been repeatedly reported the presence of such organizational axis in the brain across species (also using the CFD measure). The fact that it is also observed in marmosets is rather expected.

Response:

Understanding how the variation in structural organization underlies the brain's functional profile is a fundamental goal of neuroscience (Suárez et al., 2020). Deciphering the complicated relationship between neuroanatomical connections and functional activity in primate brains remains a daunting task, especially regarding the influence of monosynaptic connectivity on cortical activity. In this study, we investigated the anatomical-functional relationship using a graph decomposition framework that has recently sparked substantial attention in the neuroscience and engineering communities.

Graph signal processing (GSP) innovatively paves the way for probing high-order structural-functional interactions (Lioi et al., 2021). GSP exploits the topographic organization of brain structures to characterize brain activity, presenting a concise and interpretable framework (Huang et al., 2018). Previously, Atasoy et al. argued that

connectome eigenmodes derived from dMRI-based structural connectivity (SC) constraint spatiotemporal patterns of brain dynamics in humans (Atasoy et al., 2016). This framework is successfully applied to healthy and disease/disrupted brain conditions. Still, its anatomical underpinnings rely on publicly available human datasets or subject-specific tractography derived from dMRI. Some significant caveats of these techniques are the undifferentiation of the anatomical substrate at the sub-millimeter scale and the absence of directionality resulting from the fiber-tracking protocol (Thomas et al., 2014). To this extent, the connectome eigenmodes frameworks suffer from these limitations like any other connectomic approach. As a result of the limits of dMRI tractography in examining anatomical connections, the extent to which structural connectivity constrains functional activity remains largely uncharted.

The presented study provides a chance to assess the validity of such connectome eigenmode frameworks to a non-human primate species where the directionality of connections is known, as well as its potential generality across primates. Neuroanatomical tract-tracing techniques prove indispensable for mapping long-distance connections and the mesoscale cellular connectome (CC). Notably, the marmoset has the most comprehensive neuronal tract-tracing data among primates, has close homology, and has a cortical architecture similar to humans, providing an invaluable resource for investigating anatomical constraints on functional activity.

The study describes a hierarchical organization of the marmoset brain using CC eigenmodes, which conforms to the hierarchical organization observed in the human brain using the same framework of undirected graphs (Prete et al., 2019). The hierarchical organization is concordant with numerous reports in other model organisms, highlighting conserved architectural features across phylogeny (Sydnor et al., 2021).

Q2: For the marmoset activity, what is the accuracy of the reconstruction? It would be informative to have similar figures as 5 and S7 for marmoset results. There are also concerns and debates about the use of such harmonic basis for explaining empirical functional data (see, e.g., <https://doi.org/10.1038/s41586-023-06098-1> and <https://doi.org/10.1101/2023.07.20.549785>). What is the gain compared to null models for both species? Is a random harmonic basis would fit the empirical as well? The CFD is rather a very aggregate measure, what are the true contributing harmonics?

Response:

Thanks for your valuable suggestions. We added the reconstruction results of the marmoset activity and functional connectivity (FC). Furthermore, we investigated the specificity of the basis sets for explaining brain function using two null benchmarks: rewired connectome and permuted functional activity across brain regions.

(1) CC eigenmodes constrain on marmosets' brain function

We assessed the degree to which CC eigenmodes may explain cortical activity observed in BOLD-fMRI data from marmosets. We began by decomposing cortical activity into a combination of orthogonal CC eigenmodes. We next tested the accuracy of CC eigenmodes in capturing spontaneous cortical activity using this decomposition. We reconstructed the spatial activity map at each time point with an increasing CC eigenmodes, and then generated a region-region FC matrix.

(2) The specificity of CC eigenmodes to explain cortical activity

We further investigated the specificity of CC eigenmodes to explain brain function (Faskowitz et al., 2023; Pang et al., 2023). First, to illustrate the significance of the specific topology of CC for capturing empirical cortical activity, we derived eigenmodes from rewired connectomes that preserve the weight, degree, and strength distributions of the CC graph (Rubinov et al., 2010; Váša et al., 2022) (1000 repetitions). Furthermore, to test the specificity of CC eigenmodes in representing observed brain activity, we generated permuted functional activity by randomly permuting the raw rs-fMRI time series across brain regions (Griffa et al., 2022) (1000 repetitions).

(3) Results

We found that increasing CC eigenmodes improved the reconstruction accuracy of marmosets' cortical activity (Fig. 2a). Using all CC eigenmodes ($N = 55$), the cortical activity could be entirely reconstructed. The first 20 CC eigenmodes achieved 83% reconstruction accuracy of cortical activity, which was higher than utilizing both rewired CC graph ($P_{\text{rewired}} < 0.001$, false discovery rate (FDR)-corrected) and cortical activity of permuted areas ($P_{\text{perm}} < 0.001$, FDR-corrected). Similarly, the first 20 CC eigenmodes achieved 88% reconstruction accuracy of functional connectivity (FC) in marmosets (Fig. 2b). These findings suggest that CC eigenmodes canonically serve as a foundation for a compact description of spontaneous cortical activity and FC.

In the revision, we have added the relevant statement, see “**CC eigenmodes constrain on marmoset cortical activity**”.

Page 8 in the main text (Lines 139-152):

“We assessed the extent to which CC eigenmodes may explain brain activity observed in BOLD-fMRI data from marmosets. We began by decomposing cortical activity into a combination of orthogonal CC eigenmodes (**Fig. 1d**). We next tested the accuracy of CC eigenmodes in capturing spontaneous cortical activity using this decomposition. We found that increasing CC eigenmodes improved the reconstruction accuracy of marmosets’ cortical activity (**Fig. 2a**). Using all CC eigenmodes ($N = 55$), the cortical activity could be entirely reconstructed. The first 20 CC eigenmodes achieved 83% reconstruction accuracy of cortical activity, which was higher than utilizing both rewired CC graph^{52,53} ($P_{\text{rewired}} < 0.001$, false discovery rate (FDR)-corrected) and cortical activity of permuted areas⁵⁴⁻⁵⁶ ($P_{\text{perm}} < 0.001$, FDR-corrected). Similarly, the first 20 CC eigenmodes achieved 88% reconstruction accuracy of functional connectivity (FC) in marmosets (**Fig. S5**). These findings suggest that CC eigenmodes canonically serve as a foundation for a compact description of spontaneous cortical activity and FC.”

Fig. 2. Cellular connectome (CC) eigenmodes constrain cortical activity in marmosets. a Cortical activity reconstruction accuracy. Reconstruction accuracy was quantified as the energy ratio between

empirical and reconstructed cortical activity. **b** FC reconstruction accuracy in marmosets (i.e., Pearson correlation coefficients between empirical and reconstructed marmoset resting-state FC matrices across edges). The red line indicates the reconstruction accuracy of the empirical CC graph. The blue shading lines (*left*) indicate the reconstruction accuracy using eigenmodes derived from rewired CC graphs (1000 repetitions). The green shading lines (*right*) indicate the reconstruction accuracy using empirical CC eigenmodes to reconstruct permuted functional activity by randomly permuting the raw rs-fMRI time series across brain regions (1000 repetitions).

(4) The contribution of CC eigenmodes to CFD

Given a graph signal $f_{v_i}(t)$ with graph spectral coefficients $w_k(t)$, cortical activities can be isolated into low-frequency components (coupled to the CC) and high-frequency components (decoupled from the CC). The cellular-functional decoupling (CFD) of each brain area can be quantified as the ratio between the norms (L_2 -norm) of high-frequency versus low-frequency components across time points, resulting in

$$CFD_{v_i} = \log_2\left(\frac{\|f_{v_i}^{high}(t)\|_2}{\|f_{v_i}^{low}(t)\|_2}\right)$$

The CFD was used to assess the degree of local (de)coupling between cortical activity and underlying monosynaptic connections. Because the energy ratio of high- against low-frequency components is higher than 0, the ratio values between -1 and 1 are transformed using logarithmic transformation. The binary logarithm of CFD facilitates better observation of cortical activity coupling and decoupling patterns with underlying connectome.

Because the index is provided in the binary logarithmic form, CFD values of 0 represent a perfect balance between cellular-functional coupling and decoupling. CFD values around -1 indicate that cortical activity is highly coupled to monosynaptic connections, implying that low-frequency eigenmodes mainly constrain cortical activity. CFD values closer to 1, on the other hand, show that cortical activity is highly decoupled from monosynaptic connections, suggesting that high-frequency eigenmodes constrain cortical activity.

In summary, negative CFD values indicate that low-frequency connectome eigenmodes dominate the cortical activity. Positive CFD values, on the other hand, indicate the domination of high-frequency eigenmodes.

Q3: The energy analysis is quite hard to follow, would it be possible to compare directly low versus high frequency classes in figure 2, a bit like figure S4? Absolute numbers are meaningless. The discussion about OFC and dlPFC appears exacerbated because if you look at figure S4, we can also consider auditory, pmACC and mPFC with the same trend, then the reasoning does not hold anymore.

Same story when talking about the CFD gradient, coupling appears in unimodal regions, yes, but what about PCC? Also, decoupling appears in trans-modal regions but to the same extent as the coupling, except for OFC?

Response:

Thanks for your helpful suggestions. We apologize the ambiguity in our prior version. To investigate the spatial significance of the cortical class-level low- and high-frequency components, we used a nonparametric permutation test. Then, we re-discussed the patterns of low- and high-frequency components. Furthermore, we revised the conclusion of the cellular-functional decoupling (CFD) pattern.

(1) The patterns of low- and high-frequency components

We examined how CC eigenmodes constrain cortical activity. To this end, we divided the CC eigenmodes into low- and high-frequency filters. The BOLD-fMRI amplitudes for each time point were spatially filtered by low- and high-frequency eigenmodes, resulting in low- and high-frequency components that characterized to what degree BOLD-fMRI fluctuations were strongly or weakly constrained by the underlying monosynaptic connections.

We further performed a nonparametric permutation test to examine the spatial significance of the cortical class-level activity concentrations. We generated 1000 permutations for low- and high-frequency components using graph spectral randomization (SR) (Pirondini et al., 2016; Huang et al., 2018) and computed a null distribution of mean activity concentrations for each cortical class. The mean activity concentration was greater than 95% (> 95th percentile) of the null permutations, which were identified to be significantly concentrated in a given class. Significance was set at $P_{SR} < 0.05$ with false discovery rate (FDR)-correction for multiple comparisons across eleven cortical classes.

Cortical activity constrained by low-frequency eigenmodes was concentrated within dorsal-posterior cortices such as the visual (VC), somatosensory (SS), posterior parietal

cortex (PPC), and posterior cingulate, medial, and retrosplenial cortices (PCC) (all $P_{SR} < 0.05$, FDR-corrected; Fig. 2c). The high-frequency components were concentrated in ventral-anterior (frontopolar-temporal) cortices, including the medial prefrontal cortex (mPFC), dorsolateral prefrontal cortex (dlPFC), ventrolateral prefrontal cortex (vlPFC), orbitofrontal cortex (OFC), lateral inferior temporal cortex (LIT), and auditory (AU) (all $P_{SR} < 0.05$, FDR-corrected; Fig. 2d). In summary, the patterns of CC eigenmodes constrain cortical activity were circumscribed by specific brain systems.

In the revision, we have added the relevant statement, see **“Statistical analysis”** and **“CC eigenmodes constrain on marmoset cortical activity”**.

Page 9 in the main text (Lines 168-177):

“Cortical activity constrained by low-frequency eigenmodes was concentrated within dorsal-posterior cortices such as the visual cortex (VC), somatosensory cortex (SS), posterior parietal cortex (PPC), and posterior cingulate, medial, and retrosplenial cortices (PCC) (all $P_{SR} < 0.05$, FDR-corrected; **Fig. 2c**). The high-frequency components were concentrated in ventral-anterior (frontopolar-temporal) cortices, including the medial prefrontal cortex (mPFC), dorsolateral prefrontal cortex (dlPFC), ventrolateral prefrontal cortex (vlPFC), orbitofrontal cortex (OFC), lateral inferior temporal cortex (LIT), and auditory cortex (AU) (all $P_{SR} < 0.05$, FDR-corrected; **Fig. 2d**). In summary, the patterns of CC eigenmodes constrain cortical activity were circumscribed by specific brain systems.”

Fig. 2. **b** Marmoset cortical classes were parceled according to the 3D digital Paxinos atlas. **c, d** The spatial patterns of low- and high-frequency components. The low- and high-frequency components from each of the eleven cortical classes are presented as box plots ordered by the median value. Outliers are shown by the gray circles. Asterisks denote statistically significant activity concentration in each cortical class compared to the null permutation distributions (1000 repetitions, $*P_{SR} < 0.05$, one-tailed, FDR-corrected). AU = Auditory cortex, dlPFC = dorsolateral prefrontal cortex, LIT = lateral and inferior temporal cortex, mPFC = medial prefrontal cortex, MOT = motor and premotor cortex, OFC = orbitofrontal cortex, PCC = posterior cingulate, medial, and retrosplenial cortex, PPC = posterior parietal cortex, SS = Somatosensory cortex, vlPFC = ventrolateral prefrontal cortex, VC = Visual cortex.

(2) Discussion of the patterns of low- and high-frequency components

Our findings showed that monosynaptic connections strongly constrained cortical activity, particularly in the dorsal-posterior cortices, including VC, SS, PPC, and PCC, which were involved in sensation, perception, and action processes (Fukushima et al., 2019). Furthermore, weakly constrained cortical activity was mostly concentrated within frontopolar-temporal cortices, potentially requiring more diversified functional communication relative to the underlying wiring diagram (Buckner et al., 2019; Watakabe et al., 2023). Notably, the activity of the posterior core regions (PPC and PCC) in the default mode network (DMN) (Liu et al., 2023) were strongly constrained by connections, whereas

the activity of the anterior DMN (dIPFC) was weakly restricted by connections, indicating that the DMN architecture in marmosets may have a divergent anterior-posterior axis. According to a recent cross-species study, the anterior DMN regions exhibited weak connections and spatially irregular connection topology when compared with the posterior DMN areas (Ngo et al., 2023). Consequently, the patterns of marmoset brain activity constrained by CC eigenmodes were confined by specific brain systems.

In the revision, we have added the relevant statement, see “**Discussion**”.

Page 18 in the main text (Lines 339-352):

“Our findings showed that monosynaptic connections strongly constrained cortical activity, particularly in the dorsal-posterior cortices such as the VC, SS, PPC, and PCC, which were engaged in sensation, perception, and action processes⁶⁹. Furthermore, weakly constrained cortical activity was primarily concentrated within frontopolar-temporal cortices, potentially requiring more diversified functional communication relative to the underlying wiring diagram^{60,70}. Notably, the activity of the posterior core regions (PPC and PCC) in the default mode network (DMN)⁴⁵ was strongly constrained by connections, whereas the activity of the anterior DMN (dIPFC)⁴⁵ was weakly restricted by connections, indicating that the DMN architecture in marmosets may have a divergent anterior-posterior axis. According to a recent cross-species study, the anterior DMN areas have weak connections and spatially irregular connection topology compared to the posterior DMN areas⁶¹. Consequently, the patterns of marmoset brain activity constrained by CC eigenmodes were confined by specific brain systems.”

(3) Regional cellular-functional decoupling (CFD)

In addition, we investigated the cellular-functional relationships of the CC and cortical activity in marmosets. The CFD was used to assess the degree of local (de)coupling between cortical activity to underlying monosynaptic connections. The spatial pattern of CFD was regionally heterogeneous (Fig. 3, left). Regional CFD revealed a gradient organization that ranged from coupling areas in the dorsal-posterior cortices (visual, sensorimotor, motor, and premotor cortex (MOT), PPC) to decoupling areas in the ventral-anterior cortices (the temporal, prefrontal, orbitofrontal cortex) (Fig. 3, right).

Furthermore, the permutation testing approach was used to localize brain areas where the CFD significantly differed from the graph spectral randomization surrogate activity. Areas exhibiting coupling (Fig. 3a, middle), which significantly deviated from null

permutations, were primarily found in the VC (V1, V2, V3A, V4, V6, A19DI), SS (A1-2, A3a, A3b), MOT (A6Va, A8C, A4ab), and PPC (PE, PG, LIP). Regions with significant decoupling (Fig. 3a, middle) were primarily located in the OFC (A11), dIPFC (A10, A9, A8b, A46D), vIPFC (A45), and mPFC (A32V). Collectively, a gradual divergence between CC and cortical activity in marmosets was observed, transitioning from the dorsal-posterior to the ventral-anterior cortices.

a The pattern of CFD for NIH dataset

Fig. 3. (Left) The pattern of CFD in the National Institutes of Health (NIH) cohort was plotted. (Middle) The statistically significant areas were grouped into cellular-function coupling and decoupling patterns. (Right) The box plots represent the CFD values from eleven classes ordered by the median value. The grey circles represent outliers.

In the revision, we revised the conclusion of the CFD pattern.

Page 11 in the main text (Lines 205-208):

“Regional CFD revealed a gradient organization that ranged from coupling areas in the dorsal-posterior cortices (visual, sensorimotor, motor, and premotor cortex (MOT), PPC) to decoupling areas in the ventral-anterior cortices (the temporal, prefrontal, orbitofrontal cortex).”

Q4: I wonder whether the human-marmoset comparison is fair enough. Could it be that the slight increase in prediction of CC compared to SC is simply due to the fact that CC is directed? Or maybe they differ in terms of density? Also, I might have overlook some aspects, but why all reconstruction accuracies reach 1 when considering all harmonics?

Response:

(1) Comparing the performance of marmoset’s CC eigenmodes to human’s SC eigenmodes is unfair

Because human dMRI-based structural connectivity (SC) eigenmodes are inherently

noisier, comparing the performance of marmoset CC eigenmodes to human SC eigenmodes would be unfair. As a result, we did not contemplate comparing marmoset's CC eigenmodes and human's SC eigenmodes to reconstruct human activity patterns in the revision. We investigated the specificity of the basis sets for understanding human brain function using two null benchmarks: rewired connectome and randomly selected non-homologous cortical activity.

To generalize marmoset-derived eigenmodes to the human brain, we used resting-state BOLD-fMRI datasets from the 100 unrelated subjects ($n = 100$, 54 females, 22-36 years) provided by the Human Connectome Project (HCP) (<https://db.humanconnectome.org/>).

We found that increasing CC eigenmodes of homologous areas improved the reconstruction accuracy of brain activity patterns in humans (Fig. 5b). The first five CC eigenmodes achieved 90% and 86% reconstruction accuracy of brain activity and FC, respectively. Reconstructing human brain activity and FC using homologous CC eigenmodes was more accurate than rewired CC connectomes ($P_{\text{rewired}} < 0.001$, FDR-corrected) and randomly selected non-homologous areas' cortical activity ($P_{\text{perm}} < 0.001$, FDR-corrected). These findings suggest that CC eigenmodes derived from marmosets might help estimate human spontaneous brain activity and connectivity of homologous areas.

In the revision, we have added the relevant statement, see "**Generalizability of marmoset CC eigenmodes to the human brain**".

Page 15 in the main text (Lines 271-279):

"We found that increasing CC eigenmodes of homologous areas improved the reconstruction accuracy of brain activity patterns in humans (Fig. 5b). The first five CC eigenmodes achieved 90% and 86% reconstruction accuracy of brain activity and FC, respectively. Reconstructing human brain activity and FC using homologous CC eigenmodes was more accurate than rewired CC connectomes ($P_{\text{rewired}} < 0.001$, FDR-corrected) and randomly selected non-homologous areas' cortical activity ($P_{\text{perm}} < 0.001$, FDR-corrected). These findings suggest that CC eigenmodes derived from marmosets may help estimate human spontaneous brain activity and FC of homologous areas."

Cartoon
schematics
redacted
from panel
a

Fig. 5. Generalizability of marmoset-derived eigenmodes to cortical activity patterns in humans. a Humans and marmosets share 11 common homologous landmarks. **b** Cortical activity reconstruction accuracy in humans across 11 common homologous areas. **c** FC reconstruction accuracy in humans across 55 edges ($C_{11}^2 = 55$). Resting-state BOLD-fMRI in humans was used to establish empirical cortical activity and FC. The red line indicates the reconstruction accuracy of the empirical CC graph of homologous regions. The blue shading lines (left) indicate the reconstruction accuracy using eigenmodes derived from rewired CC graphs (1000 repetitions) to reconstruct unperturbed activity. The green shading lines (right) indicate the reconstruction accuracy using homologous CC eigenmodes to reconstruct randomly chosen non-homologous cortical activity (1000 repetitions). Asterisks denote statistically significant reconstruction accuracy compared to the null permutation distributions (1000 repetitions, $*P < 0.05$, one-tailed, FDR-corrected).

(2) The performance of marmoset's CC eigenmodes vs marmoset's SC eigenmodes

In reconstructing marmosets' cortical activity and functional connectivity (FC), we compared the reconstruction accuracy of CC eigenmodes against SC eigenmodes. We reconstructed SC matrices using probabilistic diffusion tractography. A group SC matrix was generated by averaging all individual SC matrices. Because the connection density of CC and SC differed (the density of CC was 0.6842; the density of SC was 0.9912), we thresholded the SC matrix to construct a connection matrix that matched the density of CC (Fig. R1). The CC has proven to be an indispensable tool for mapping long-distance connections and obtaining the direction of information flow (Majka et al., 2016 & 2019). The SC was transmitted between cortical areas either via linkage of the dense local connections or rare, extraordinarily privileged long-range connections (Rosen et al., 2022).

One limitation of dMRI was the difficulty in achieving fine enough spatial resolution to map small but essential fiber pathways.

Furthermore, the SC eigenmodes were derived from the normalized undirected graph Laplacian according to previous methods (Huang et al., 2018; Preti et al., 2019; Rué-Queralt et al., 2021) (Fig. S6a). Although the CC eigenmodes account for the projection directionality and edge density, they do not reveal local gray matter connections and white matter fiber pathways. The SC eigenmodes do not take fiber directionality into account, but they capture white matter connections measured with dMRI (Naze et al., 2021; Pang et al., 2023). A direct comparison of the reconstruction accuracy of two distinct basis sets revealed that CC eigenmodes numerically outperform SC eigenmodes in capturing cortical activity (Fig. S6b) and FC (Fig. S6c). Hence, the CC eigenmodes provided a more parsimonious description of cortical activity than the SC eigenmodes.

A prior mouse research discovered that the CC provides a better ability to shape brain function than SC (Melozzi et al., 2019), owing to the fact that neural tracing may reveal information about the direction of information flow and long-distance connections.

The relevant statement has been included in the revision, see “**CC eigenmodes constrain on marmoset cortical activity**”.

Page 8 in the main text (Lines 153-157):

“We next compared the reconstruction accuracy of CC eigenmodes against dMRI-based SC eigenmodes (Fig. S6). A direct comparison of the reconstruction accuracy of two distinct basis sets revealed that CC eigenmodes numerically outperform the SC eigenmodes in capturing cortical activity and FC. The CC eigenmodes provided a more parsimonious description of cortical activity than the SC eigenmodes.”

Fig. R1. (This figure is just shown in response to reviewers). **Marmoset tracer-based cellular connectome (CC) (a) and dMRI-based structural connectome (SC) (b) matrix.**

Fig. S6. Cellular connectome (CC) eigenmodes were compared to structural connectome (SC) eigenmodes. **a** The parts of SC eigenmodes were projected onto the marmoset brain surface. Blue–white–red colors represented negative–zero–positive values. Despite certain similarities, the spatial patterns of SC eigenmodes differed from CC eigenmodes. **b** Reconstruction accuracy of marmoset cortical activity achieved by SC and CC eigenmodes. Reconstruction accuracy was quantified as the energy ratio between empirical and reconstructed cortical activity. **c** Reconstruction accuracy (i.e., Pearson correlation coefficients between empirical and reconstructed marmoset FC matrices across edges) of marmoset FC achieved by SC and CC eigenmodes. The green line indicates the reconstruction accuracy of the empirical SC graph. The orange shading lines (upper) indicate the reconstruction accuracy using eigenmodes derived from rewired SC graphs (1000 repetitions) to reconstruct unperturbed activity. The purple shading lines (middle) indicate the reconstruction accuracy using empirical SC eigenmodes to reconstruct permuted functional activity by randomly permuting the raw rs-fMRI time series across brain regions (1000 repetitions). The red line (bottom) indicates the reconstruction accuracy of the empirical CC graph.

(3) Why reconstruction accuracy reach 1 when considering all eigenmodes?

The BOLD-fMRI signals are transformed into the orthonormal basis (connectome eigenmodes) via the graph Fourier transform (GFT). The spectral representation of the

activity patterns according to GFT weights, i.e., $w_k(t) = \Psi^T f_{v_i}(t)$, which quantifies how much each connectome eigenmode contributes to activity pattern. One such spectral representation is obtained for each point in time. In the inverse direction, each activity pattern can be reconstructed as a weighted sum of the connectome eigenmodes, i.e., $f_{v_i}(t) = w_1(t)\psi_1 + w_2(t)\psi_2 + \dots + w_N(t)\psi_N = \Psi w_k(t)$, without loss of information. Thus, reconstruction accuracies reach 1 when considering all connectome eigenmodes.

Q5. I find the term cellular-function confusing, it makes me think about something which has nothing to do with connectivity.

Response:

We aimed to decipher the relationship between brain structure and function in marmosets. Neuroanatomical tract-tracing techniques stand unrivaled in directly detecting monosynaptic connections. These monosynaptic transneuronal tracers prove indispensable for mapping the mesoscale cellular-level connectome (Oh et al., 2014; Majka et al., 2020). Previously, gold-standard cellular-level tracing was referred to as cellular connectome (CC) (Oh et al., 2014; Lin et al., 2019; Majka et al., 2020; Hori et al., 2020; Skibbe et al., 2023).

Reviewer #1 Q5 and Q13 suggested that we rename “cellular-functional dependency” to “cellular-functional decoupling” throughout the manuscript. The cellular-functional decoupling (CFD) was used to assess the degree of local (de)coupling between cortical activity and monosynaptic connections. The binary logarithm form of the energy ratio of high-frequency (cortical activity decoupled from CC) versus low-frequency (cortical activity coupled to CC) components represents each brain area's CFD. Lower CFD implied a strong coupling of cortical activity to the neural connections, whereas higher CFD indicated the reverse.

Bibliographies in the Response to Reviewer #2:

- Atasoy, S., et al., (2016). Human brain networks function in connectome-specific harmonic waves. *Nature communications*, 7(1), 10340.
- Buckner, R. L., & Margulies, D. S. (2019). Macroscale cortical organization and a default-like apex transmodal network in the marmoset monkey. *Nature communications*, 10(1), 1976.
- Fukushima, M., et al., (2019). Neuroanatomy of the marmoset. In *The common marmoset in captivity and biomedical research* (pp. 43-62). Academic Press.
- Faskowitz, J., et al., (2023). Commentary on Pang et al.(2023) Nature. *bioRxiv*, Preprint at <https://doi.org/10.1101/2023.07.20.549785>.
- Griffa, A., et al., (2022). The evolution of information transmission in mammalian brain networks, *bioRxiv*, Preprint at <https://doi.org/10.1101/2022.05.09.491115>.
- Huang, W., et al., (2018). A graph signal processing perspective on functional brain imaging. *Proceedings of the IEEE*, 106(5), 868-885.
- Hori, Y., et al., (2020). Comparison of resting-state functional connectivity in marmosets with tracer-based cellular connectivity. *Neuroimage*, 204, 116241.
- Liu, C., et al., (2019). Anatomical and functional investigation of the marmoset default mode network. *Nature communications*, 10(1), 1975.
- Lin, M. et al., (2019). A high-throughput neurohistological pipeline for brain-wide mesoscale connectivity mapping of the common marmoset. *Elife*, 8, e40042.
- Lioi, G., et al., (2021). Gradients of connectivity as graph Fourier bases of brain activity. *Network Neuroscience*, 5(2), 322-336.
- Majka, P., et al., (2016). Towards a comprehensive atlas of cortical connections in a primate brain: Mapping tracer injection studies of the common marmoset into a reference digital template. *Journal of Comparative Neurology*, 524(11), 2161-2181.
- Melozzi, F., et al., (2019). Individual structural features constrain the mouse functional connectome. *Proceedings of the National Academy of Sciences*, 116(52), 26961-26969.
- Majka, P., et al., (2020). Open access resource for cellular-resolution analyses of corticocortical connectivity in the marmoset monkey. *Nature Communications*, 11(1), 1133.
- Naze, S., et al., (2021). Robustness of connectome harmonics to local gray matter and long-range white matter connectivity changes. *NeuroImage*, 224, 117364.
- Ngo, G. N., et al., (2023). Joint-embeddings reveal functional differences in default-mode network architecture between marmosets and humans. *NeuroImage*, 272, 120035.
- Oh, S. et al., (2014). A mesoscale connectome of the mouse brain. *Nature*, 508(7495), 207-214.
- Pirondini, E., et al., (2016). A spectral method for generating surrogate graph signals. *IEEE signal processing letters*, 23(9), 1275-1278.
- Preti, M. G., & Van De Ville, D. (2019). Decoupling of brain function from structure reveals regional behavioral specialization in humans. *Nature communications*, 10(1), 4747.
- Pang, J. C., et al., (2023). Geometric constraints on human brain function. *Nature*, 618, 566-574.
- Pang, J. C., et al., (2023). Reply to: Commentary on Pang et al.(2023) Nature. *bioRxiv*, Preprint at

- <https://doi.org/10.1101/2023.10.06.560797>.
- Rubinov, M., & Sporns, O. (2010). Complex network measures of brain connectivity: uses and interpretations. *Neuroimage*, 52(3), 1059-1069.
- Rué-Queralt, J., et al., (2021). The connectome spectrum as a canonical basis for a sparse representation of fast brain activity. *NeuroImage*, 244, 118611.
- Rosen, B. Q., & Halgren, E. (2022). An estimation of the absolute number of axons indicates that human cortical areas are sparsely connected. *PLoS Biology*, 20(3), e3001575.
- Suárez, L. E., et al., (2020). Linking structure and function in macroscale brain networks. *Trends in cognitive sciences*, 24(4), 302-315.
- Sydnor, V. J., et al., (2021). Neurodevelopment of the association cortices: Patterns, mechanisms, and implications for psychopathology. *Neuron*, 109(18), 2820-2846.
- Skibbe, H., et al., (2023). The Brain/MINDS Marmoset Connectivity Resource: An open-access platform for cellular-level tracing and tractography in the primate brain. *PLoS biology*, 21(6), e3002158.
- Thomas, C., et al., (2014). Anatomical accuracy of brain connections derived from diffusion MRI tractography is inherently limited. *Proceedings of the National Academy of Sciences*, 111(46), 16574-16579.
- Váša, F., & Mišić, B. (2022). Null models in network neuroscience. *Nature Reviews Neuroscience*, 23(8), 493-504.
- Watakabe, A., et al., (2023). Local and long-distance organization of prefrontal cortex circuits in the marmoset brain. *Neuron* 111, 2121–2122.

Comments from the Reviewer #3 (23 points)

This paper used graph signal processing to study the structure-function relationship in marmosets. The authors used neuronal-tracing data to create the underlying anatomical connectivity (cellular connectome) and showed that their method can disentangle cortical activity according to a hierarchy of coupled and decoupled regions, following known hierarchical organization of the brain. In most respects I found the paper to be interesting. However, I do have several major and minor concerns/issues, which are itemized below.

Response:

We greatly appreciate the insightful comments on our manuscript. We hope our responses are satisfactory and improve the quality of the current work.

Major concerns:

1. On methodology

Q1: CC is a non-symmetric (directed) matrix, hence its Laplacian is also non-symmetric. Mathematically, the eigenvectors of non-symmetric matrices do not necessarily form a complete orthonormal basis set. Hence, they may not be linearly independent, which is important for basis set expansions as done in Eq. (3). The authors need to quantitatively verify that the CC harmonics are linearly independent, otherwise using them to decompose cortical activity data may not be valid.

Response:

A directed graph is naturally represented by a non-symmetric adjacency matrix. The graph Laplacian provides a bridge between spectral graph theory and signal processing. It is no longer appropriate to use the graph Laplacian directly for directed graphs. The conventional undirected graph Laplacian (normalized $\mathcal{L}_{undir} = I - D^{-\frac{1}{2}}AD^{-\frac{1}{2}}$, where A is the adjacency matrix, and D is the diagonal matrix of degrees of A) is invalid because its symmetries are no longer verified. To expand the Laplacian-based Fourier analysis from undirected graphs to directed graphs, a suitable reference operator must be found.

As an acceptable reference operator for expanding the signal processing framework to directed graphs, the random walk operator on graphs was presented (Sevi et al., 2023). The random walk operator is related to the concept of diffusion on graphs, which may transform any graph into a Markov chain (Seabrook et al., 2023). To avoid the complications of non-diagonalizable matrices, Chung (Chung, 2005) defined a symmetric

normalized Laplacian matrix for directed strongly connected graphs. This Laplace operator is defined as a self-adjoint operator using the transition probability operator and the Perron vector. Notably, the normalized directed graph Laplacian, which has been used in spectral clustering, graph embedding, and classification applications, may represent the graph's directionality and edge density by leveraging the random walk operator (Seabrook et al., 2023).

We assumed the CC as a weighted directed graph and $A = \{A_{ij}\}_{1 \leq i, j \leq N} \in R_+^{N \times N}$ is defined as the FLNe value of the CC. The normalized graph Laplacian for directed graphs (Chung, 2005) is defined as,

$$\mathcal{L}_{dir} = I - \frac{\Pi^{1/2} P \Pi^{-1/2} + \Pi^{-1/2} P^T \Pi^{1/2}}{2}, \quad (1)$$

where $P = D^{-1}A$ is the transition matrix of the ergodic Markov chain, $D = \sum_{j=1}^N A_{ij}$ denotes the out-degrees matrix of A , $\Pi = \text{diag}\{\pi(v_1), \dots, \pi(v_N)\}$ is the diagonal matrix of the stationary distribution π , i.e., $\pi P = \pi$, I is an identity matrix.

The restriction on the underlying graph so that P is ergodic. The stationary probabilities are all non-zero, which is needed for $\Pi^{-1/2}$ to be well-defined. Eq. (1) can be simplified as follows:

$$\begin{aligned} \mathcal{L}_{dir} &= I - \frac{\Pi^{1/2} P \Pi^{-1/2} + \Pi^{-1/2} (\Pi^{-1} P^T \Pi) \Pi^{1/2}}{2} \\ &= I - \Pi^{1/2} \frac{(P + \Pi^{-1} P^T \Pi)}{2} \Pi^{-1/2} \\ &= I - \Pi^{1/2} P_A \Pi^{-1/2}, \end{aligned}$$

where $P_A = \frac{(P + \Pi^{-1} P^T \Pi)}{2}$ is the additive reversibility of the random walk (Seabrook et al., 2023).

Therefore, the Laplacian matrix satisfies $\mathcal{L}_{dir} = \mathcal{L}_{dir}^T$, i.e., \mathcal{L}_{dir} is a symmetric matrix. It ensures CC eigenvectors (eigenmodes) derived from the eigen-decomposition \mathcal{L}_{dir} are orthonormal. We proved the orthogonality by the fact that the product of the eigenvectors of \mathcal{L}_{dir} matrix and its transpose is equal to the identity matrix, i.e., $\Psi \Psi^T = I$. Due to the orthogonality, the CC eigenmodes are linearly independent, so they are a basis set that can decompose spatiotemporal dynamics as a weighted sum. In Eq. (3) (i.e., $f_{v_i}(t) = w_1(t)\psi_1 + w_2(t)\psi_2 + \dots + w_N(t)\psi_N = \Psi w_k(t)$), it would be feasible to represent cortical activity as a combination of CC eigenmodes.

In the revision, we have added the relevant statement, see “**Connectome Laplacian and eigenmodes of the CC.**”

Q2: The authors should explain why transforming the CC to a graph Laplacian via the transition matrix is necessary. Otherwise, non-expert readers may confuse this step as a standard approach for all cases, including undirected matrices, which is not true.

Response:

The graph Laplacian connects spectral graph theory and signal processing. The undirected graph Laplacian (normalized $\mathcal{L}_{undir} = I - D^{-\frac{1}{2}}AD^{-\frac{1}{2}}$, where A is the adjacency matrix, and D is the diagonal matrix of degrees of A) is a symmetric and positive semi-definite operator (Chung, 1997). According to the spectral theorem, Laplacian's undirected graph admits a set of eigenvectors forming an orthonormal basis. These eigenvectors can be thought of as Fourier modes, accompanying eigenvalues related to the notion of frequency (Chung, 1997).

The random walk operator on graphs is introduced as an appropriate reference operator for expanding the signal processing framework to directed graphs (Sevi et al., 2023). The random walk operator is related to the concept of diffusion on graphs, which may transform any graph into a Markov chain (Seabrook et al., 2023). To avoid the complications of non-diagonalizable matrices, Chung (Chung, 2005) defined a symmetric normalized Laplacian matrix for directed graphs with strongly connected. This definition is inspired by the fact that the degree matrix D used in the definition of the undirected Laplacian. The D is the diagonal matrix, which is proportional to the limiting distribution of a random walk on an undirected graph (Spielman, 2012; Seabrook et al., 2023; Sevi et al., 2023). As a result, the normalized directed graph Laplacian is constructed by replacing D with the Perron vector for the random walk with the transition matrix P on the graph (Chung, 2005; Spielman, 2012).

The normalized directed graph Laplacian, which has been used in spectral clustering, graph embedding, and classification applications, may represent the graph's directionality and edge density by leveraging the random walk operator. Notably, the resulting Laplace matrix is symmetric. This approach assures that eigenvectors derived from the Laplacian matrix's eigen-decomposition form a set of orthogonal basics. The Dirichlet energy of a given eigenvector of the random walk operator on a directed graph is associated with eigenvalue, which can be regarded as graph frequencies (Sevi et al, 2023). As a result, eigenvectors are regarded as a suitable Fourier basis for functions defined on directed graphs.

In the revision, we have added the relevant statement, see "**Connectome Laplacian and**

eigenmodes of the CC.”

Q3: The analysis revolves around the dichotomy between low- and high-frequency harmonics, which heavily relies on the filter cutoff that preserves the total energy in each band. I know past studies have done the same, but I find this approach and the hard cutoff ill-defined. The authors need to provide a more compelling reason for this approach and/or show robustness to the choice of the filter cut-off.

Response:

In the revision, we reported results representing the low- and high-frequency components using median-split frequency C ($C = 12$). To verify the robustness of the filter cut-off setting, we also considered separating the CC eigenmodes into low, medium, and high components (Huang et al., 2016; Medaglia et al., 2018). We specifically decomposed the observed BOLD-fMRI signals into low- and high-frequency components using $K_L = 7$ to $K_L = 11$, and $K_L = 13$ to $K_L = 17$ (K_L from five below to five above the cut-off frequency C in the main text), and K_H ranging from $K_H = 31$ to $K_H = 40$. We then computed the spatial correlation of low- and high-frequency components, as well as the CFD patterns between the original and robustness analysis.

We found that the observed signal decomposition was robust to the choice of filter cut-off parameter: for low-frequency components (Fig. S7a), the spatial correlation values ranged from 0.785 to 0.942, all $P_{\text{SMASH}} < 0.0001$, and the mean spatial correlation was 0.868 (S.D. = 0.066); for high-frequency components (Fig. S7b), the spatial correlation values ranged from 0.647 to 0.852, all $P_{\text{SMASH}} < 0.0001$, and the mean spatial correlation was 0.747 (S.D. = 0.068).

Furthermore, we found that the CFD patterns were consistent across the filter cut-off setting (Fig. S8): the spatial correlation coefficients ranged from 0.779 to 0.920, all $P_{\text{SMASH}} < 0.0001$, and the mean spatial correlation was 0.872 (S.D. = 0.051).

In summary, low- and high-frequency components, as well as the CFD patterns, are highly stable regardless of the filter cut-off setting. We have added sensitivity analyses of the filter cut-off setting in the results; see “**Supplementary Information for S6 Sensitivity analyses**”.

Fig. S7. Robustness of low- and high-frequency components to filter cut-off setting. The observed signal decomposition was robust under different low- and high-frequency eigenmodes. The significance of correlation coefficients (P_{SMASH}) is evaluated using 1000 spatial autocorrelation-preserving surrogate brain maps.

Fig. S8. Robustness of regional cellular-functional decoupling (CFD) pattern to filter cut-off setting. The patterns of CFD were stable under different low- and high-frequency eigenmodes. The significance of correlation coefficients (P_{SMASH}) is evaluated using 1000 spatial autocorrelation-preserving surrogate brain maps.

Q4: In the calculation of CFD, the authors should explain why the logarithmic transformation is needed. It is also unclear from Eq. (6) how the temporal component disappears. More importantly, the physical meaning of CFD is difficult to comprehend. For example, how does it inform the correspondence between functional signals and CC (e.g., Lines 178-179)?

Response:

(1) The necessity of logarithmic transformation of CFD

Given a graph signal $f_{v_i}(t)$ with graph spectral coefficients $w_k(t)$, cortical activities can be isolated into low-frequency components (coupled to the CC) and high-frequency components (decoupled from the CC). The cellular-functional decoupling (CFD) of each brain area was quantified as the activity concentration (L_2 -norm) ratio of high-frequency versus low-frequency components across time points, resulting in

$$CFD_{v_i} = \log_2 \left(\frac{\|f_{v_i}^{high}(t)\|_2}{\|f_{v_i}^{low}(t)\|_2} \right)$$

The CFD was used to assess the degree of local (de)coupling between cortical activity and underlying monosynaptic connections. Because the energy ratio of high- against low-frequency components is higher than 0, the ratio values between -1 and 1 are transformed using logarithmic transformation. The binary logarithm of CFD facilitates better observation of cortical activity coupling and decoupling patterns with underlying connectome.

Because the index is provided in the binary logarithmic form, CFD values of 0 represent a perfect balance between cellular-functional coupling and decoupling. CFD values around -1 indicate that cortical activity is highly coupled to monosynaptic connections, implying that low-frequency eigenmodes mainly constrain cortical activity. CFD values closer to 1, on the other hand, show that cortical activity is highly decoupled from monosynaptic connections, suggesting that high-frequency eigenmodes constrain cortical activity.

(2) Illustrate how the temporal component disappears in Eq. (6)

In Eq. (6), we calculated the activity concentration (L_2 -norm) of the filtered high-frequency and low-frequency time series across time points to quantify the CFD pattern for each brain region, resulting in the temporal component disappearing. Specifically, the

filtered low- or high-frequency components are a vector of length T (time points) for each brain area. We calculated the L_2 -norm for this vector, and then it becomes a value.

(3) The physical meaning of CFD

To characterize the cellular-functional decoupling (CFD) of each brain area, we measured the binary logarithm form of the ratio between the norms (L_2 -norm) of high-frequency (cortical activity decoupled to cellular connectome (CC)) to low-frequency (cortical activity coupled to CC) components. The CFD was used to assess the degree of local (de)coupling between cortical activity and underlying monosynaptic connections. Lower CFD implied a strong coupling of cortical activity to neural connections, whereas higher CFD indicated the reverse.

We found that the spatial pattern of CFD was regionally heterogeneous. Regional CFD revealed a gradient organization that ranged from coupling areas in the dorsal-posterior cortices (visual, sensorimotor, motor, and premotor cortex (MOT), PPC) to decoupling areas in the ventral-anterior cortices (the temporal, prefrontal, orbitofrontal cortex).

We have added an explanation of the necessity of logarithmic transformation of the CFD index. In addition, we updated the CFD pattern's conclusion. The revision has been amended as follows:

Page 29 in the main text (Lines 594-601):

“The cellular-functional decoupling (CFD) of each brain area can be quantified as the ratio between the norms (L_2 -norm) of high-frequency versus low-frequency components across time points¹², resulting in

$$CFD_{v_i} = \log_2 \left(\frac{\|f_{v_i}^{high}(t)\|_2}{\|f_{v_i}^{low}(t)\|_2} \right) \quad (6)$$

We chose the binary logarithmic form of this index, so CFD values of 0 represent a perfect balance between cellular-functional coupling and decoupling. CFD values around -1 imply that cortical activity is strongly coupled to underlying monosynaptic connections, whereas CFD values around 1 indicate the reverse.”

Page 11 in the main text (Lines 204):

“The correspondence between empirical functional signals and CC was regionally heterogeneous.”

It would be more appropriate to revise this sentence to “The spatial pattern of CFD was

regionally heterogeneous.”

Q5: The authors used the spin test for null testing, which I think is inappropriate because there are a lot of discontinuities/missing regions. There is a greater chance of reassigning missing data to the nearest region, hence the null statistics are not stringent enough. A parametric approach such as variogram matching is more suitable, which the authors should look at.

Response:

We do agree that the spin test has a greater chance of reassigning missing data to the nearest region. By applying variogram matching to the generative modeling, a parametric approach avoids spatial rotation (Viladomat et al., 2014; Burt et al., 2020).

In the revision, we performed a permutation test to generate spatial autocorrelation (SA)-preserving surrogate brain maps using variogram matching (Viladomat et al., 2014; Burt et al., 2020). This approach can be implemented through the BrainSMASH software (Burt et al., 2020) (<https://github.com/murraylab/brainsmash>). The revision has been amended as follows:

Page 30 in the main text (Lines 607-609):

“The significance of correlations was assessed using BrainSMASH, a spatial autocorrelation (SA)-preserving surrogate method⁸⁹ (Supplementary Information S4).”

Page 7 in the Supplementary Information:

“S4 Spatial correction for brain map similarity

The intrinsic spatial smoothing of two given brain maps may exaggerate the significance of their spatial correlation. To this end, we performed a permutation test to generate spatial autocorrelation (SA)-preserving surrogate brain maps using variogram matching^{23,24}. The parametric method can be subdivided into two main steps. First, the empirical brain map is randomly permuted. Then, the permuted maps are smoothed and rescaled to reintroduce the SA characteristic of the empirical brain map. This approach can be implemented through the brainSMASH software²³ (<https://github.com/murraylab/brainsmash>). Specifically, we used BrainSMASH to generate 1000 surrogate maps of the empirical map. The P_{SMASH} value was calculated as the percentile rank of the surrogate correlations that are more extreme than the observed correlation (<5th or >95th percentile).”

Q6: I also found it bizarre that the authors used human functional data from a dataset different from HCP. HCP has one of the best quality data out there, so why not just use it all throughout? In fact, I highly suggest they check the robustness of their results on HCP data.

Response:

Thanks for the reviewer's suggestions. To generalize marmoset-derived eigenmodes to the human brain, we included resting-state BOLD-fMRI datasets from the 100 unrelated subjects (n = 100, 54 females, 22-36 years) provided by the Human Connectome Project (HCP) (<https://db.humanconnectome.org/>) (Van Essen et al., 2013). The BOLD-fMRI data was preprocessed according to the HCP minimal preprocessing pipelines. We did not conduct any additional preprocessing steps. Finally, the preprocessed time series were parcellated into 180 cortical areas of the left hemisphere using the HCP-MMP1.0 parcellation (Glasser et al., 2016).

In the revision, we have added the relevant statement, see “**Generalizability of marmoset CC eigenmodes to the human brain**”.

Page 15 in the main text (Lines 271-279):

“We found that increasing CC eigenmodes of homologous areas improved the reconstruction accuracy of brain activity patterns in humans (**Fig. 5b**). The first five CC eigenmodes achieved 90% and 86% reconstruction accuracy of brain activity and FC, respectively. Reconstructing human brain activity and FC using homologous CC eigenmodes was more accurate than rewired CC connectomes ($P_{\text{rewired}} < 0.001$, FDR-corrected) and randomly selected non-homologous areas' cortical activity ($P_{\text{perm}} < 0.001$, FDR-corrected). These findings suggest that CC eigenmodes derived from marmosets may help estimate human spontaneous brain activity and FC of homologous areas. ”

Editorial Note:
Cartoon schematics redacted from panel a

Fig. 5. Generalizability of marmoset-derived eigenmodes to cortical activity patterns in humans. a Humans and marmosets share 11 common homologous landmarks. **b** Cortical activity reconstruction accuracy in humans across 11 common homologous areas. **c** FC reconstruction accuracy in humans across 55 edges ($C_{11}^2 = 55$). Resting-state BOLD-fMRI in humans was used to establish empirical cortical activity and FC. The red line indicates the reconstruction accuracy of the empirical CC graph of homologous regions. The blue shading lines (left) indicate the reconstruction accuracy using eigenmodes derived from rewired CC graphs (1000 repetitions) to reconstruct unperturbed activity. The green shading lines (right) indicate the reconstruction accuracy using homologous CC eigenmodes to reconstruct randomly chosen non-homologous cortical activity (1000 repetitions). Asterisks denote statistically significant reconstruction accuracy compared to the null permutation distributions (1000 repetitions, $*P < 0.05$, one-tailed, FDR-corrected).

2. On analysis and results

Q7: Figure 2 is really confusing to read. Statements in Lines 145 to 152 are not supported by the figure either. For example, the authors said that there is high energy concentration of low graph frequency in PmACC, but its value is comparable to those of dlPFC and vlPFC. There is also a lot of differentiation within cortical classes to make overarching statements like these. The authors also need to do some statistical testing in comparing the different cortical classes.

Response:

Thanks for your helpful suggestions. We apologize the ambiguity in our prior version. To investigate the spatial significance of the cortical class-level low- and high-frequency components, we used a nonparametric permutation test. Then, we re-discussed the

patterns of low- and high-frequency components. Furthermore, we revised the conclusion of the cellular-functional decoupling (CFD) pattern.

(1) The patterns of low- and high-frequency components

We examined how CC eigenmodes constrain cortical activity. To this end, we divided the CC eigenmodes into low- and high-frequency filters. The BOLD-fMRI amplitudes for each time point were spatially filtered by low- and high-frequency eigenmodes, resulting in low- and high-frequency components that characterized to what degree BOLD-fMRI fluctuations were strongly or weakly constrained by the underlying monosynaptic connections.

We further performed a nonparametric permutation test to examine the spatial significance of the cortical class-level activity concentrations. We generated 1000 permutations for low- and high-frequency components using graph spectral randomization (SR) (Pirondini et al., 2016; Huang et al., 2018) and computed a null distribution of mean activity concentrations for each cortical class. The mean activity concentration was greater than 95% (> 95th percentile) of the null permutations, which were identified to be significantly concentrated in a given class. Significance was set at $P_{SR} < 0.05$ with false discovery rate (FDR)-correction for multiple comparisons across eleven cortical classes.

Cortical activity constrained by low-frequency eigenmodes was concentrated within dorsal-posterior cortices such as the visual (VC), somatosensory (SS), posterior parietal cortex (PPC), and posterior cingulate, medial, and retrosplenial cortices (PCC) (all $P_{SR} < 0.05$, FDR-corrected; Fig. 2c). The high-frequency components were concentrated in ventral-anterior (frontopolar-temporal) cortices, including the medial prefrontal cortex (mPFC), dorsolateral prefrontal cortex (dlPFC), ventrolateral prefrontal cortex (vlPFC), orbitofrontal cortex (OFC), lateral inferior temporal cortex (LIT), and auditory (AU) (all $P_{SR} < 0.05$, FDR-corrected; Fig. 2d). In summary, the patterns of CC eigenmodes constrain cortical activity were circumscribed by specific brain systems.

In the revision, we have added the relevant statement, see “**Statistical analysis**” and “**CC eigenmodes constrain on marmoset cortical activity**”.

Page 9 in the main text (Lines 168-177):

“Cortical activity constrained by low-frequency eigenmodes was concentrated within dorsal-posterior cortices such as the visual cortex (VC), somatosensory cortex (SS),

posterior parietal cortex (PPC), and posterior cingulate, medial, and retrosplenial cortices (PCC) (all $P_{SR} < 0.05$, FDR-corrected; **Fig. 2c**). The high-frequency components were concentrated in ventral-anterior (frontopolar-temporal) cortices, including the medial prefrontal cortex (mPFC), dorsolateral prefrontal cortex (dlPFC), ventrolateral prefrontal cortex (vlPFC), orbitofrontal cortex (OFC), lateral inferior temporal cortex (LIT), and auditory cortex (AU) (all $P_{SR} < 0.05$, FDR-corrected; **Fig. 2d**). In summary, the patterns of CC eigenmodes constrain cortical activity were circumscribed by specific brain systems.”

Fig. 2. **b** Marmoset cortical classes were parceled according to the 3D digital Paxinos atlas. **c, d** The spatial patterns of low- and high-frequency components. The low- and high-frequency components from each of the eleven cortical classes are presented as box plots ordered by the median value. Outliers are shown by the gray circles. Asterisks denote statistically significant activity concentration in each cortical class compared to the null permutation distributions (1000 repetitions, $*P_{SR} < 0.05$, one-tailed, FDR-corrected). AU = Auditory cortex, dlPFC = dorsolateral prefrontal cortex, LIT = lateral and inferior temporal cortex, mPFC = medial prefrontal cortex, MOT = motor and premotor cortex, OFC = orbitofrontal cortex, PCC = posterior cingulate, medial, and retrosplenial cortex, PPC = posterior parietal cortex, SS = Somatosensory cortex, vlPFC = ventrolateral prefrontal cortex, VC = Visual cortex.

(2) Discussion of the patterns of low- and high-frequency components

Our findings showed that monosynaptic connections strongly constrained cortical

activity, particularly in the dorsal-posterior cortices, including VC, SS, PPC, and PCC, which were involved in sensation, perception, and action processes (Fukushima et al., 2019). Furthermore, weakly constrained cortical activity was mostly concentrated within frontopolar-temporal cortices, potentially requiring more diversified functional communication relative to the underlying wiring diagram (Buckner et al., 2019; Watakabe et al., 2023). Notably, the activity of the posterior core regions (PPC and PCC) in the default mode network (DMN) (Liu et al., 2023) were strongly constrained by connections, whereas the activity of the anterior DMN (dlPFC) was weakly restricted by connections, indicating that the DMN architecture in marmosets may have a divergent anterior-posterior axis. According to a recent cross-species study, the anterior DMN regions exhibited weak connections and spatially irregular connection topology when compared with the posterior DMN areas (Ngo et al., 2023). Consequently, the patterns of marmoset brain activity constrained by CC eigenmodes were confined by specific brain systems.

In the revision, we have added the relevant statement, see “**Discussion**”.

Page 18 in the main text (Lines 339-352):

“Our findings showed that monosynaptic connections strongly constrained cortical activity, particularly in the dorsal-posterior cortices such as the VC, SS, PPC, and PCC, which were engaged in sensation, perception, and action processes⁶⁹. Furthermore, weakly constrained cortical activity was primarily concentrated within frontopolar-temporal cortices, potentially requiring more diversified functional communication relative to the underlying wiring diagram^{60,70}. Notably, the activity of the posterior core regions (PPC and PCC) in the default mode network (DMN)⁴⁵ was strongly constrained by connections, whereas the activity of the anterior DMN (dlPFC)⁴⁵ was weakly restricted by connections, indicating that the DMN architecture in marmosets may have a divergent anterior-posterior axis. According to a recent cross-species study, the anterior DMN areas have weak connections and spatially irregular connection topology compared to the posterior DMN areas⁶¹. Consequently, the patterns of marmoset brain activity constrained by CC eigenmodes were confined by specific brain systems.”

Q8: The authors said that results were replicated across datasets. Whilst the correlations of unthresholded CFD are high, there are a lot of differences in the significant regions found in the two datasets. How do they reconcile this?

Response:

To examine the reproducibility of patterned CFD, we estimated the spatial correlation of regional CFD between the NIH and ION datasets (Fig. 3). In addition, we calculated the Dice coefficient to assess the overlap of significantly coupled or decoupled regions in the two datasets. The results showed a statistically significant positive correlation ($\rho = 0.83$, $P_{SMASH} < 0.001$, Fig. S10). Furthermore, for significantly decoupled and coupled areas, the Dice coefficient was 0.73 and 0.70, respectively. As a result, these findings confirmed the reproducibility of CFD patterns. The revision has been updated accordingly:

Page 13 in the main text (Lines 231-232):

“The results showed a statistically significant positive correlation ($\rho = 0.83$, $P_{SMASH} < 0.001$, Fig. S10). Furthermore, for significantly decoupled and coupled areas, the Dice coefficient was 0.73 and 0.70, respectively. As a result, these findings confirmed the reproducibility of the CFD patterns.”

a The pattern of CFD for NIH dataset**b The pattern of CFD for ION dataset**
Fig. 3. Regional cellular-functional decoupling (CFD) in two independent datasets. **a** (Left) The pattern of CFD in the National Institutes of Health (NIH) cohort was plotted. (Middle) The statistically significant areas were grouped into cellular-function coupling and decoupling patterns. (Right) The box plots represent the CFD values from eleven classes ordered by the median value. Outliers are shown by the gray circles. **b** The pattern of CFD and the statistically significant areas in the Institute of Neuroscience (ION) cohort.

Fig. S10. Repeatability analysis of regional cellular-functional decoupling (CFD) pattern in two independent datasets. The spatial correlation of the cellular-functional decoupling (CFD) pattern between the National Institutes of Health (NIH) sites dataset and the Institute of Neuroscience (ION) sites dataset showed excellent reproducibility. The significance of correlation coefficients (P_{SMASH}) is evaluated using 1000 spatial autocorrelation-preserving surrogate brain maps.

3. The authors need to take extra care in making inferences from their human results. There are several issues with the current approach.

Q9: The dimensionality of the data and harmonics was reduced to 11 to restrict analysis to 11 homologous regions between the species. This removes a lot of important connections to other regions; hence, the resulting harmonics cannot account for true brain activity. In fact, this becomes problematic even for the marmoset results because the authors are producing harmonics from a subset of their original harmonics, so that the spatial patterns found in Fig. 1b are difficult to reconcile with those in Fig. S7A.

Response:

Predefined homologous areas across species were used to generalize marmoset-derived eigenmodes to the human brain. Because marmosets have only recently gained traction as an animal model, established human-marmoset homologous areas have been lacking (Ngo et al., 2023). Our study was based on a modest sample size of 11 mapped areas. We first examined whether local cellular connectome (CC) eigenmodels can reconstruct

activity patterns in marmosets. To do this, we used the same reconstruction framework to reconstruct local cortical activity using a basis set obtained from the CC matrix of homologous regions (Fig. S11a).

As the eigenvalue increased, the spatial distribution of homologous CC eigenmodes became more intricate (Fig.S11b). The first homologous CC eigenmode, in particular, is uniformly distributed throughout the brain. The second one separates the motor cortex from the rest of the brain. The third CC eigenmode shows a dimension between the anterior cingulate and other brain regions. The spatial layout of these eigenmodes is similar to that seen in Fig. 1b.

We found that increasing homologous CC eigenmodes enhanced the reconstruction accuracy of marmoset cortical activity (Fig. S11c) and FC (Fig. S11d). The first five CC eigenmodes reached 86% and 83% reconstruction accuracy of cortical activity and FC in marmosets, respectively. Reconstructing marmoset cortical activity and FC using homologous CC eigenmodes showed better performance than rewired CC connectomes ($P_{\text{rewired}} < 0.001$, FDR-corrected) and randomly chosen non-homologous area' cortical activity ($P_{\text{perm}} < 0.001$, FDR-corrected). These results suggest that eigenmodes anatomically localized can reconstruct brain activity patterns (Patil et al., 2023), which is consistent with the classic neuroscientific paradigm in which cortical activity should be described as “discrete, isolated, and anatomically localized” clusters (Jones et al., 1999).

In the revision, we have added the relevant statement, see Supporting information “**S7 Localized eigenmodes can reconstruct activity patterns in marmosets**”.

Page 14 in the main text (Lines 266-270):

“We then constructed a CC matrix using these 11 homologous markers. We used a basis set generated from the CC matrix of homologous areas to reconstruct localized brain activity patterns in marmosets (Fig. S11), implying that localized CC eigenmodes potentially reflect activity patterns in marmosets.”

Editorial Note:
Cartoon schematic redacted from panel a

Fig. S11. Reconstruction accuracy of marmoset cortical activity and functional connectivity (FC) achieved by homologous cellular connectome (CC) eigenmodes. **a** Homologous landmarks (11 areas) in marmosets. **b** The CC eigenmodes of homologous regions. **c** Reconstruction accuracy of cortical activity in marmosets across 11 common homologous areas. Reconstruction accuracy was quantified as the energy ratio between empirical and reconstructed cortical activity. **d** Reconstruction accuracy (i.e., The Pearson correlation coefficients between empirical and reconstructed FC across 55 edges) of marmoset FC. Resting-state BOLD-fMRI in marmosets was used to establish empirical cortical activity and FC. The red line indicates the reconstruction accuracy of the empirical CC graph of homologous regions. The blue shading lines (left) indicate the reconstruction accuracy using eigenmodes derived from rewired CC graphs (1000 repetitions) to reconstruct unperturbed activity. The green shading lines (right) indicate the reconstruction accuracy using homologous CC eigenmodes to reconstruct randomly selected non-homologous cortical activity (1000 repetitions). Asterisks denote statistically significant of reconstruction accuracy compared to the null permutation distributions ($*P < 0.05$, one-tailed, FDR-corrected).

Q10: It is unfair to compare the performance of the marmoset CC harmonics to human diffusion-based harmonics as the latter is inherently noisier. Moreover, the case would have been stronger if the authors follow my Comment 1f.

Response:

Because human dMRI-based structural connectivity (SC) eigenmodes are inherently noisier, comparing the performance of marmoset CC eigenmodes to human SC

eigenmodes would be unfair. As a result, we did not contemplate comparing marmoset's CC eigenmodes and human's SC eigenmodes to reconstruct human activity patterns in the revision. We investigated the specificity of the basis sets for understanding human brain function using two null benchmarks: rewired connectome and randomly selected non-homologous cortical activity.

To generalize marmoset-derived eigenmodes to the human brain, we used resting-state BOLD-fMRI datasets from the 100 unrelated subjects ($n = 100$, 54 females, 22-36 years) provided by the HCP.

We found that increasing CC eigenmodes of homologous areas improved the reconstruction accuracy of brain activity patterns in humans (Fig. 5b). The first five CC eigenmodes achieved 90% and 86% reconstruction accuracy of brain activity and FC, respectively. Reconstructing human brain activity and FC using homologous CC eigenmodes was more accurate than rewired CC connectomes ($P_{\text{rewired}} < 0.001$, FDR-corrected) and randomly selected non-homologous areas' cortical activity ($P_{\text{perm}} < 0.001$, FDR-corrected). These findings suggest that CC eigenmodes derived from marmosets might help estimate human spontaneous brain activity and connectivity of homologous areas.

In the revision, we have added the relevant statement, see "**Generalizability of marmoset CC eigenmodes to the human brain**".

Page 15 in the main text (Lines 271-279):

"We found that increasing CC eigenmodes of homologous areas improved the reconstruction accuracy of brain activity patterns in humans (Fig. 5b). The first five CC eigenmodes achieved 90% and 86% reconstruction accuracy of brain activity and FC, respectively. Reconstructing human brain activity and FC using homologous CC eigenmodes was more accurate than rewired CC connectomes ($P_{\text{rewired}} < 0.001$, FDR-corrected) and randomly selected non-homologous areas' cortical activity ($P_{\text{perm}} < 0.001$, FDR-corrected). These findings suggest that CC eigenmodes derived from marmosets may help estimate human spontaneous brain activity and FC of homologous areas."

Fig. 5. Generalizability of marmoset-derived eigenmodes to cortical activity patterns in humans. a Humans and marmosets share 11 common homologous landmarks. **b** Cortical activity reconstruction accuracy in humans across 11 common homologous areas. **c** FC reconstruction accuracy in humans across 55 edges ($C_{11}^2 = 55$). Resting-state BOLD-fMRI in humans was used to establish empirical cortical activity and FC. The red line indicates the reconstruction accuracy of the empirical CC graph of homologous regions. The blue shading lines (left) indicate the reconstruction accuracy using eigenmodes derived from rewired CC graphs (1000 repetitions) to reconstruct unperturbed activity. The green shading lines (right) indicate the reconstruction accuracy using homologous CC eigenmodes to reconstruct randomly chosen non-homologous cortical activity (1000 repetitions). Asterisks denote statistically significant reconstruction accuracy compared to the null permutation distributions (1000 repetitions, $*P < 0.05$, one-tailed, FDR-corrected).

Q11: The authors should also test whether the CC harmonics can predict 11 non-homologous (e.g., randomly chosen) regions in humans. If they find that CC harmonics perform equally well, then there is a problem here, with the results trivially driven by the mathematics of basis set expansions and not some biologically relevant evolutionary process.

Response:

We investigated the specificity of the basis sets for explaining human brain function using two null benchmarks: rewired connectome and randomly chosen non-homologous cortical activity. We used homologous CC eigenmodes to reconstruct randomly chosen non-homologous cortical activity (1000 repetitions). We found that homologous CC eigenmodes performed poorly in reconstructing randomly chosen non-homologous

cortical activity patterns (Fig. 5b) and FC of humans (Fig. 5c), demonstrating CC eigenmodes of homologous regions exhibited specificity for explaining brain function in humans.

Our results found that marmoset's CC eigenmodes may reflect human spontaneous cortical activity patterns of homologous regions, which were not driven by general mathematical properties of basis set expansions but could be derived by some biologically relevant evolutionary process. These findings suggested that CC may be used to restrict brain function across species.

In the revision, we have added the relevant statement, see "**Generalizability of marmoset CC eigenmodes to the human brain**" and "**Discussion**".

Page 15 in the main text (Lines 272-277):

"The first five CC eigenmodes achieved 90% and 86% reconstruction accuracy of brain activity and FC, respectively. Reconstructing human brain activity and FC using homologous CC eigenmodes was more accurate than rewired CC connectomes ($P_{\text{rewired}} < 0.001$, FDR-corrected) and randomly selected non-homologous areas' cortical activity ($P_{\text{perm}} < 0.001$, FDR-corrected)."

Page 20 in the main text (Lines 397-403):

"The identification of probable homologous landmarks has been used to make cross-species comparisons^{51,61,74}. Our results showed that marmoset's CC eigenmodes may reflect human spontaneous cortical activity in homologous areas, which was not driven by general mathematical properties of basis set expansions but could be derived by some biologically relevant evolutionary process. These findings suggested that CC may be used to restrict brain function across species."

4. On discussion

Q12: There is an article in Nature that came out this year showing that the geometry of the human brain fundamentally constrains brain function, not just complex connectivity, using the same framework in this study. The authors should discuss how their work ties with this emerging idea and whether the same geometric influence can be observed in marmosets.

Response:

We first examined how geometric eigenmodes may explain cortical activity in marmosets. Then, we discussed how geometric eigenmodes related to our present work.

(1) Geometric eigenmodes constrain on marmoset cortical activity

Following recent research (Pang et al., 2023), we first examined how geometric eigenmodes may explain cortical activity in marmosets. To derive geometric eigenmodes, we used a mesh representation of a population-averaged template of the neocortical surface (Marmoset Brain Mapping V3, <https://marmosetbrainmapping.org/v3.html>, surfFS.lh.graymid.surf.gii) (Liu et al., 2021). The geometric eigenmodes of the marmoset cortex were calculated using the LaPy python library (Wachinger et al., 2015) (Fig. R2a).

Using this decomposition, we evaluated the accuracy of geometric eigenmodes in capturing marmoset cortical activity. We observed that as the number of geometric eigenmodes grows, so does the reconstruction accuracy. The first 100 geometric eigenmodes caught 15.61% reconstruction accuracy of brain activity (Fig. R2b). Similarly, the first 100 geometric eigenmodes achieved 84.19% correlation of marmoset FC (Fig. R2c). These findings indicated that cortical geometric eigenmodes might capture spontaneous cortical activity in marmosets.

Because we obtained the tracer-based CC (region \times region) and dMRI-based SC (region \times region) at the region level rather than the vertex level, the eigen-decomposition of the Laplacian matrix can produce at most connectome eigenmodes as the number of brain regions. On the other hand, geometric eigenmodes can yield as many eigenmodes as the number of brain vertices. As a result, it is currently impossible to directly compare the performance of CC eigenmodes, SC eigenmodes, and geometry eigenmodes.

We will follow previous procedures (Atasoy et al., 2016; Naze et al., 2021) and combine a mix of cortical mesh (local gray matter) and tractography (long-range white matter) connections to generate the weighted connectivity matrices (vertex \times vertex). The hypothesis that geometric eigenmodes may provide a more parsimonious description of cortical activity than connectome eigenmodes in marmosets will next be tested (Pang et al., 2023).

Fig. R2. (This figure is just shown in response to reviewers). **Reconstruction of cortical activity with geometric eigenmodes in marmosets.** **a** Geometric eigenmodes are derived from the cortical surface mesh by solving the eigenvalue problem, $\Delta\Psi = -\lambda\Psi$. The eigenmodes $\psi_1, \psi_2, \psi_3, \dots, \psi_N$ are ordered from low to high spatial frequency. Negative, zero and positive values are coloured blue, white and red, respectively. **b** Reconstruction accuracy of marmoset cortical activity with geometric eigenmodes. Reconstruction accuracy was quantified as the energy ratio between empirical and reconstructed cortical activity. **c** Reconstruction accuracy (i.e., Pearson correlation coefficients between empirical and reconstructed marmoset resting-state FC matrices across edges) of marmoset FC achieved by geometric eigenmodes.

(2) Discuss how geometric eigenmodes tie in with the current work

Previous studies have extensively utilized eigenmode approaches to explain human brain function (Atasoy et al., 2016; Huang et al., 2018; Pang et al., 2023). The geometric eigenmodes produced from the brain's geometry (e.g., its shape) capture local spatial relations, representing the underlying anatomical restrictions on human brain function (Pang et al., 2023). Our work with marmosets utilized a similar framework, but the

underlying anatomical properties differed (geometric shape, tracer-based CC). Tracer-based CC represents an asymmetric (directed) network. Graph signal models based on asymmetric network operators may be better for signal and information processing on directed projections. The CC eigenmodes produced from Laplacian's normalized directed graph may capture the projection directionality and edge density, emphasizing the importance of projection directionality information.

In the revision, we have added the relevant statement, see “**Discussion**”.

Page 17 in the main text (Lines 314-326):

““Previous studies have extensively utilized eigenmode approaches to understand human brain function^{15-17,19,67}. Geometric eigenmodes derived from the brain’s geometry (e.g., its shape) capture local spatial relations, representing the underlying anatomical restrictions on human brain function¹⁵. Our work with marmosets used a similar approach, but the fundamental anatomical features differed. The neuroanatomical tracing connectome represents an asymmetric (directed) network. Graph signal models based on asymmetric network operators may be better for signal and information processing on directed projections⁴⁴. The CC eigenmodes produced from Laplacian's normalized directed graph may capture the projection directionality and edge density, emphasizing the importance of projection directionality information.”

Q13: Lines 302 to 308. The authors argued that the current work addresses traditional approaches of averaging temporal data to obtain FC, which inadvertently removes the rich characteristics of brain dynamics. I found this baffling as the authors also took the average of their CFD across time, so in principle, they are also removing temporal dynamics.

Response:

Previous studies have used correlation and regression analyses (Hori et al., 2019, Tian et al., 2022), and whole-brain computational modeling (Tong et al., 2022) to assess the local or global correspondence between FC and CC in marmosets. However, in these investigations, FC was represented by BOLD-fMRI temporal correlations measured throughout time periods, which may give a degraded depiction of brain connectivity (Marrelec et al., 2016; Medaglia et al., 2018). In contrast, our work decomposed cortical activity into CC harmonics-informed components, which may represent how BOLD-fMRI

amplitudes were strongly or weakly constrained by monosynaptic connections at a specific time point.

The graph Fourier transform provides a distinct perspective from connectivity analyses by allowing the spectrum of the anatomical network to inform estimates of how BOLD-fMRI amplitudes in each observation (not functional connectivity computed over time) are constrained by monosynaptic connections. Specifically, BOLD-fMRI amplitudes for each time point were spatially filtered by low- and high-frequency eigenmodes, resulting in low- and high-frequency components (Fig. 1d).

Low-frequency components revealed that the cortical activity at each time point is tightly restricted by the underlying monosynaptic connections: heavily interconnected nodes exhibit similar functional dynamics to one another. High-frequency components, on the other hand, suggested a divergence between the spatial structure of the functional signal and the underlying network: nodes may exhibit various activities even if they are heavily connected in the monosynaptic connections (Fig.1d). As a result, the GSP framework enables a frame-wise temporal resolution to uncover the structure-informed study of functional brain dynamics.

In addition, we further investigated the cellular-functional relationships of the CC and cortical activity in marmosets. To illustrate cellular-functional decoupling (CFD), we quantified the binary logarithm form of the ratio of the norms of high-frequency versus low-frequency components over time points (Fig. 1d). Importantly, the CFD was particularly useful in determining the degree of local (de)coupling between cortical activity and underlying monosynaptic connections. We did not take the average of the CFD over time. Instead, we calculated the ratio between the high and low-frequency signal norms over time.

Fig. 1. d Decomposition of BOLD-fMRI. The fMRI data at each time point ($t(i)$) was estimated as the contribution ($w_k^{t(i)}$) of each CC eigenmode (ψ_k). Cortical activity was then decomposed into low-frequency components (coupled to CC, i.e., heavily interconnected nodes tend to display similar activity to one another) and high-frequency components (decoupled from CC, i.e., nodes exhibit various activities even if they are heavily connected). Cellular-functional decoupling (CFD) was the ratio between the norms of decoupled and coupled components over time points.

Q14: I don't understand Lines 365 to 368 and Lines 373 to 374. How could CC harmonics inform human connectivity, especially given that the CC harmonics calculated in the work did not have full brain coverage and that not all marmoset brain regions can be homologously mapped to humans? The authors should tone down these inferences.

Response:

Predefined homologous areas across species were used to generalize marmoset-derived eigenmodes to the human brain. Because marmosets have only recently gained traction as an animal model, established human-marmoset homologous areas have been lacking (Ngo et al., 2023). Our study was based on a modest sample size of 11 mapped areas. As a result, we reduced the inferences and acknowledged the shortcomings of the methodology. The revision has been amended as follows:

Page 15 in the main text (Lines 277-279):

“These findings suggest that CC eigenmodes derived from marmosets may help estimate human spontaneous brain activity and FC of homologous areas.”

Page 21 in the main text (Lines 397-403):

“The identification of probable homologous landmarks has been used to make cross-species comparisons^{51,61,74}. Our results showed that marmoset’s CC eigenmodes may reflect human spontaneous cortical activity in homologous areas, which was not driven by general mathematical properties of basis set expansions but could be derived by some biologically relevant evolutionary process. These findings suggested that CC may be used to restrict brain function across species.”

Page 22 in the main text (Lines 423-425):

“Fourth, our research was based on a restricted number of mapped areas due to the absence of documented marmoset-human homologous areas.”

Q15: The authors also claimed that CC could inform diffusion-based connectivity. The marmoset brain mapping resource has diffusion MRI available, so the authors can easily compare the performance of marmoset CC and diffusion-based connectome to test this empirically.

Response:

In reconstructing marmosets' cortical activity and functional connectivity (FC), we compared the reconstruction accuracy of CC eigenmodes against SC eigenmodes. We reconstructed SC matrices using probabilistic diffusion tractography. A group SC matrix was generated by averaging all individual SC matrices. Because the connection density of CC and SC differed (the density of CC was 0.6842; the density of SC was 0.9912), we thresholded the SC matrix to construct a connection matrix that matched the density of CC (Fig. R1). The CC has proven to be an indispensable tool for mapping long-distance connections and obtaining the direction of information flow (Majka et al., 2016 & 2019). The SC was transmitted between cortical areas either via linkage of the dense local connections or rare, extraordinarily privileged long-range connections (Rosen et al., 2022). One limitation of dMRI was the difficulty in achieving fine enough spatial resolution to

map small but essential fiber pathways.

Furthermore, the SC eigenmodes were derived from the normalized undirected graph Laplacian according to previous methods (Huang et al., 2018; Preti et al., 2019; Rué-Queralt et al., 2021) (Fig. S6a). Although the CC eigenmodes account for the projection directionality and edge density, they do not reveal local gray matter connections and white matter fiber pathways. The SC eigenmodes do not take fiber directionality into account, but they capture white matter connections measured with dMRI (Naze et al., 2021; Pang et al., 2023). A direct comparison of the reconstruction accuracy of two distinct basis sets revealed that CC eigenmodes numerically outperform SC eigenmodes in capturing cortical activity (Fig. S6b) and FC (Fig. S6c). Hence, the CC eigenmodes provided a more parsimonious description of cortical activity than the SC eigenmodes.

A prior mouse research discovered that the CC provides a better ability to shape brain function than SC (Melozzi et al., 2019), owing to the fact that neural tracing may reveal information about the direction of information flow and long-distance connections.

The relevant statement has been included in the revision, see “**CC eigenmodes constrain on marmoset cortical activity**”.

Page 8 in the main text (Lines 153-157):

“We next compared the reconstruction accuracy of CC eigenmodes against dMRI-based SC eigenmodes (Fig. S6). A direct comparison of the reconstruction accuracy of two distinct basis sets revealed that CC eigenmodes numerically outperform the SC eigenmodes in capturing cortical activity and FC. The CC eigenmodes provided a more parsimonious description of cortical activity than the SC eigenmodes.”

Fig. R1. (This figure is just shown in response to reviewers). **Marmoset tracer-based cellular connectome (CC) (a) and dMRI-based structural connectome (SC) (b) matrix.**

Fig. S6. Cellular connectome (CC) eigenmodes were compared to structural connectome (SC) eigenmodes. **a** The parts of SC eigenmodes were projected onto the marmoset brain surface. Blue–white–red colors represented negative–zero–positive values. Despite certain similarities, the spatial patterns of SC eigenmodes differed from CC eigenmodes. **b** Reconstruction accuracy of marmoset cortical activity achieved by SC and CC eigenmodes. Reconstruction accuracy was quantified as the energy ratio between empirical and reconstructed cortical activity. **c** Reconstruction accuracy (i.e., Pearson correlation coefficients between empirical and reconstructed marmoset FC matrices across edges) of marmoset FC achieved by SC and CC eigenmodes. The green line indicates the reconstruction accuracy of the empirical SC graph. The orange shading lines (upper) indicate the reconstruction accuracy using eigenmodes derived from rewired SC graphs (1000 repetitions) to reconstruct unperturbed activity. The purple shading lines (middle) indicate the reconstruction accuracy using empirical SC eigenmodes to reconstruct permutated functional activity by randomly permutating the raw rs-fMRI time series across brain regions (1000 repetitions). The red line (bottom) indicates the reconstruction accuracy of the empirical CC graph.

Q16: Because of the sparsity of regions, the CC harmonics do not match typical spatial patterns and their natural ordering found in the literature (e.g., second harmonics typically follows the anterior-posterior gradient). The authors should discuss this discrepancy.

Response:

Previous studies (Atasoy et al., 2016; Preti et al., 2019; Glomb et al., 2020; Ye et al., 2022) have discovered that low-frequency connectome eigenmodes represent brain patterns of global and gradual fluctuations along the key geometrical axes, such as the left-right, anterior-posterior, ventral-to-dorsal, and the medial-peripheral. The graph frequency increases as the eigenvalue increases, resulting in finer-grained subdivisions of the cortex. High-frequency eigenmodes encode patterns that become increasingly complicated, irregular, and localized. We investigated the spatial patterns of SC eigenmodes in humans (Atasoy et al., 2016; Preti et al., 2019; Glomb et al., 2020; Ye et al., 2022) and found that patterns and their natural order are not always matched.

In Atasoy et al. (2016), the first SC eigenmode captures the left-right axis. The second SC eigenmode captures the medial-peripheral axis. The anterior-posterior geometric axis is depicted in the third SC eigenmode (Fig. R3).

Figure redacted

Fig. R3. (This figure is just shown in response to reviewers). The first three SC eigenmodes from (Atasoy et al., 2016).

In Preti et al. (2019), the first SC eigenmode is homogeneous across the brain. The second SC eigenmode captures the left-right axis. The anterior-posterior geometric axis is depicted in the third SC eigenmode (Fig. R4).

Figure redacted

Fig. R4. (This figure is just shown in response to reviewers). The first four SC eigenmodes from (Preti et al., 2019).

In Glomb et al. (2020), the first SC eigenmode is homogeneous across the brain. The second SC eigenmode captures the anterior-posterior axis. The left-right geometric axis is depicted in the third SC eigenmode. The fourth SC eigenmode divides the parietal lobe from the rest of the brain (Fig. R5).

Figure redacted

Fig. R5. (This figure is just shown in response to reviewers). The first four SC eigenmodes from (Glomb et al., 2020).

In Ye et al. (2022), the first SC eigenmode is homogeneous across the brain. The second SC eigenmode captures the anterior-posterior axis. The left-right geometric axis is depicted in the third SC eigenmode. The fourth SC eigenmode divides the parietal lobe from the rest of the brain (Fig. R6).

Figure redacted

Fig. R6. (This figure is just shown in response to reviewers). The first four SC eigenmodes from (Ye et al., 2022).

All human studies used undirected graphs derived from dMRI tractography to calculate SC eigenmodes, which capture local spatial relations between mesh vertices and white matter connections. Our study used a normalized directed graph Laplacian of CC to derive CC eigenmodes, which capture the projection directionality and edge density leveraging the random walk operator.

With increasing eigenvalue (or frequency), the irregularity and localization of CC eigenmode patterns increased. Specifically, the first CC eigenmode is uniformly distributed throughout the brain. The second reflects a dorsal-ventral dimension. The third shows a dimension between the anterior cingulate and other brain regions. The fourth represents a gradient axis from the sensorimotor cortex to the visual cortex, resembling the principal structural gradient in marmosets (Fig. 1b). Despite certain similarities, the spatial patterns of the CC eigenmodes differed from previous human SC eigenmodes.

Fig. 1. b The first four CC eigenmodes (ψ_1 – ψ_4 , in ascending order by eigenvalue) were projected onto the marmoset brain surface. Colors visualized arbitrary units, i.e., the weights in eigenvectors.

In the revision, we have added the relevant statement, see “**Discussion**”.

Page 17 in the main text (Lines 314-326):

“Previous studies have extensively utilized eigenmode approaches to understand human brain function^{15-17,19,67}. The SC eigenmodes obtained from dMRI-based undirected networks capture local gray matter and white matter fiber connections, which serve as the foundation for human functional networks¹⁶⁻¹⁸. Our work with marmosets used a similar approach, but the fundamental anatomical features differed. The neuroanatomical tracing connectome represents an asymmetric (directed) network. Graph signal models based on asymmetric network operators may be better for signal and information processing on directed projections⁴⁴. The CC eigenmodes produced from Laplacian's normalized directed graph may capture the projection directionality and edge density, emphasizing

the importance of projection directionality information.”

5: Minor concerns:

Q17: The authors need to be careful with terminologies such as “decoding”, “complexity”, and “energy”. They have different connotations in different fields. In fact the term “energy” is not properly defined. I would also refrain from using the terms “liberal” and “aligned” as they are misleading. Just stick with low- and high-frequency.

Response:

We have modified the term “decoding” to “decomposing”. The term “complexity” was modified to “irregularity”. The term “energy” was modified to “activity”. The terminologies “liberal” and “aligned” were modified to “low” and “high-frequency”.

We defined the L_2 -norm of BOLD-fMRI signal $f_{v_i}(t)$ across all time points to measure the cortical activity concentration of the graph frequency component at the brain area v_i : $E_{v_i} = \|f_{v_i}(t)\|_2$. We used the L_2 -norm because it provides an interpretation of energy for each graph frequency component (Medaglia et al., 2018; Huang et al., 2018; Preti et al., 2019). In signal processing, computing the L_2 -norm of a signal corresponds to the square root of its energy calculated in the spatial domain.

The revision has been updated accordingly:

Page 6 in the main text (Lines 110-114):

“Decoding BOLD-fMRI. Graph signal at each time point t_i was estimated as the contribution $w_k^{(t_i)}$ of connectome harmonics ψ_k to the cortical activity. Cortical activity was decoded into low-frequency (aligned) and high-frequency (liberal) components.”

It would be more appropriate to revise this sentence to “Decomposition of BOLD-fMRI. The fMRI data at each time point ($t(i)$) was estimated as the contribution ($w_k^{t(i)}$) of each CC eigenmode (ψ_k). Cortical activity was then decomposed into low-frequency components (coupled to CC, i.e., heavily interconnected nodes tend to display similar activity to one another) and high-frequency components (decoupled from CC, i.e., nodes exhibit various activities even if they are heavily connected).”

Page 7 in the main text (Lines 129-130):

“The complexity and localization of CC harmonic patterns increased with increasing

eigenvalue (or frequency).”

It would be more appropriate to revise this sentence to “The irregularity and localization of CC eigenmode patterns increased with increasing eigenvalue (or frequency).”

Page 28 in main text (Lines 575-578):

“In addition, to measure the energy concentration of the graph frequency component at a brain area $i \in V$, we computed the ℓ_2 -norm of f_i^t across all temporal samples of a given subject.”

It would be more appropriate to revise this sentence to “In addition, to measure the cortical activity concentration of the graph frequency component in the brain area v_i , we computed the L_2 -norm of $f_{v_i}(t)$ across all time points of a given subject: $E_{v_i} = \|f_{v_i}(t)\|_2$, which provided an interpretation of energy for each graph frequency component^{12,16,22}.”

6. There are a couple of things that need further clarification.

Q18: Is the CC matrix data only unihemispheric?

Response:

The CC measured according to the Paxinos parcellation of the left hemisphere (Paxinos, G., et al, 2012) was supplied by The Marmoset Brain Connectivity Atlas (<http://marmoset.braincircuits.org/>) (Majka, P., et al, 2020). As a result, we employed the CC matrix based on the Paxinos parcellation of the left hemisphere. The revision has been amended as follows:

Page 23 in main text (Lines 458-460):

“The CC matrix A quantified according to the Paxinos atlas⁴⁶ of the left hemisphere was used to construct a directed graph $G_{55 \times 55}$ for the following analysis, unless otherwise stated.”

Q19: What do the graphs at the bottom of Figure 1D mean?

Response:

The BOLD-fMRI signals for each time point were spatially filtered by low- and high-frequency eigenmodes, resulting in low- and high-frequency components that described how BOLD-fMRI fluctuations are strongly or weakly coupled constrained by the underlying monosynaptic connections (Medaglia et al., 2018; Huang et al., 2018; Preti et al., 2019) (Fig. 1d).

Low-frequency components revealed that the cortical activity at each time point is tightly restricted by the underlying monosynaptic connections: heavily interconnected nodes exhibit similar functional dynamics to one another. High-frequency components, on the other hand, suggested a divergence between the spatial structure of the functional signal and the underlying network: nodes may exhibit various activities even if they are heavily connected in the monosynaptic connections (Fig.1d).

In addition, we investigated the cellular-functional relationships of the CC and cortical activity in marmosets. To illustrate cellular-functional decoupling (CFD), we quantified the binary logarithm form of the ratio of the norms of high-frequency versus low-frequency components over time points (Fig. 1d). Importantly, the CFD was particularly useful in determining the degree of local (de)coupling between cortical activity and underlying monosynaptic connections.

Fig. 1. d Decomposition of BOLD-fMRI. The fMRI data at each time point ($t(i)$) was estimated as the contribution ($w_k^{t(i)}$) of each CC eigenmode (ψ_k). Cortical activity was then decomposed into low-frequency components (coupled to CC, i.e., heavily interconnected nodes tend to display similar activity to one another) and high-frequency components (decoupled from CC, i.e., nodes exhibit various activities even if they are heavily connected). Cellular-functional decoupling (CFD) was the ratio between the norms of decoupled and coupled components over time points.

Q20: Wrong caption in Figure 1B colorbar. It is the eigenvector not eigenvalue.

Response:

We apologize for miswriting the eigenvector as eigenvalue. We have corrected it in the revision. In the last version, the vertical bars represented the first four CC eigenmodes (eigenvectors) in vector form (in ascending order by eigenvalue), and were projected onto the marmoset brain surface. Colors visualize arbitrary units, i.e., the weights in eigenvectors.

Fig. 1. b The first four CC eigenmodes (ψ_1 – ψ_4 , in ascending order by eigenvalue) were projected onto the marmoset brain surface. Colors visualized arbitrary units, i.e., the weights in eigenvectors.

Q21: Line 132: principal gradient of what?

Response:

The fourth CC eigenmode represents a gradient axis from the sensorimotor cortex to the visual cortex, resembling the marmosets' principal structural gradient in marmosets (Tong et al., 2022). The revision has been updated accordingly:

Page 7 in the main text (Lines 133-135):

“The fourth CC eigenmode represented a gradient axis from the sensorimotor cortex to the visual cortex, resembling the marmosets’ principal structural gradient⁵¹”

Q22: Provide details of preprocessing of human data.

Response:

We used resting-state BOLD-fMRI data from the 100 unrelated subjects provided by the Human Connectome Project (HCP) (Van Essen et al., 2013) to generalize marmoset-derived eigenmodes to the human brain. The revision has been updated accordingly:

Page 25 in the main text (Lines 504-518):

“We used resting-state BOLD-fMRI datasets from the 100 unrelated subjects ($n = 100$; 54 females and 46 males; 22-36 years) provided by the Human Connectome Project (HCP) (<https://db.humanconnectome.org/>)⁷⁸. The HCP data was acquired using protocols approved by the Washington University Institutional Review Board. We analyzed rs-fMRI data acquired in the first scanning session using a left-to-right (LR) encoding direction. The scan lasted 14.4 minutes ($TR = 0.72s$) with 1200 time points. A detailed description of the acquisition was previously published⁷⁸.

The BOLD-fMRI data was preprocessed according to the HCP minimal preprocessing pipelines⁷⁹. The rs-fMRI data was adjusted for gradient nonlinearity, head motion, and geometric distortions. Further preprocessing procedures included registration of the corrected images to the T1 weighted images, brain extraction, global intensity normalization, high-pass filtering (cut-off at 2000 s)⁸⁰, and elimination of residual confounds through the ICA-FIX method⁸¹. We did not conduct any additional preprocessing steps. Finally, the preprocessed time series were parcellated into 180 cortical areas of the left hemisphere using the HCP-MMP1.0 parcellation⁶².”

Q23: How to get the diagonal matrix Π in Eq. (1)?

Response:

The stationary distribution π of the random walk with the transition matrix P can be obtained by calculating the Perron vector of P . The Perron vector π can be calculated using the Matlab function “eigs” (Fig. S13). The detailed procedure for calculating the normalized directed graph Laplacian is shown in Algorithm S1. An example of computing normalized digraph Laplacian is shown in Fig.S3. The source code for this method is available at https://epfl-lts2.github.io/gspbox-html/doc/utils/gsp_create_laplacian.html. The stationary distribution reflects that the distribution of states converges to a stable distribution as the Markov chain progresses, which provides insights into the system's long-term behavior.

In the revision, we have added the relevant statement, see Supporting information “**Normalized directed graph Laplacian**”.

Algorithm S1: Normalized directed graph Laplacian

Input: $A \in R_+^{N \times N}$: weighted directed adjacency matrix

Output: L : normalized directed graph Laplacian

- 1: Compute the diagonal matrix D of the out-degrees of A .
 - 2: Compute the transition matrix P of the random walk, see Eq.(1).
 - 3: Compute the stationary distribution π , i.e. $\pi P = \pi$, with $\sum_{i=1}^N \pi(v_i) = 1, \pi(v_i) \geq 0$. π can be obtained by calculating the Perron vector of P , which can be calculated by use Matlab function “eigs”.
 - 4: Calculate the diagonal matrix $\Pi = \text{diag}\{\pi(v_1), \dots, \pi(v_N)\}$ of the stationary distribution π .
 - 5: Calculate the normalized directed graph Laplacian L , see Eq. (2).
 - 6: Return a symmetric matrix L .
-

Fig. S3. An example of computing normalized digraph Laplacian. First, a directed graph is represented an asymmetric weighted adjacency matrix. Second, compute the diagonal matrix D of the out-degrees of A . Third, compute the transition matrix P of the random walk $\chi = (X_n)_{n \geq 0}$. Fourth, compute the stationary distribution π , which can be obtained by calculating the Perron vector of P . Finally, calculate the normalized directed graph Laplacian L .

Fig. S13. Visualization Perron vector π of the transition matrix P of Markov chain χ . The Perron vector π of P can be calculated using Matlab function “eigs”.

Bibliographies in the Response to Reviewer #3:

- Atasoy, S., et al., (2016). Human brain networks function in connectome-specific harmonic waves. *Nature communications*, 7(1), 10340.
- Burt, J. B., et al., (2020). Generative modeling of brain maps with spatial autocorrelation. *NeuroImage*, 220, 117038.
- Chung, F. (2005). Laplacians and the Cheeger inequality for directed graphs. *Annals of Combinatorics*, 9, 1-19.
- Glasser, M. F., et al., (2016). A multi-modal parcellation of human cerebral cortex. *Nature*, 536(7615), 171-178.
- Glomb, K., et al., (2020). Connectome spectral analysis to track EEG task dynamics on a subsecond scale. *NeuroImage*, 221, 117137.
- Huang, W., et al., (2016). Graph frequency analysis of brain signals. *IEEE journal of selected topics in signal processing*, 10(7), 1189-1203.
- Huang, W., et al., (2018). A graph signal processing perspective on functional brain imaging. *Proceedings of the IEEE*, 106(5), 868-885.
- Hori, Y., et al., (2020). Comparison of resting-state functional connectivity in marmosets with tracer-based cellular connectivity. *Neuroimage*, 204, 116241.
- Jones, E. G. (1999). Golgi, Cajal and the neuron doctrine. *Journal of the History of the Neurosciences*, 8(2), 170-178.
- Lovász, L. (1993). Random walks on graphs. *Combinatorics, Paul erdos is eighty*, 2(1-46), 4.
- Liu, C., et al., (2021). Marmoset Brain Mapping V3: Population multi-modal standard volumetric and surface-based templates. *Neuroimage*, 226, 117620.
- Marrelec, G., et al., (2016). Functional connectivity's degenerate view of brain computation. *PLoS computational biology*, 12(10), e1005031.
- Medaglia, J. D., et al., (2018). Functional alignment with anatomical networks is associated with cognitive flexibility. *Nature human behaviour*, 2(2), 156-164.
- Majka, P., et al., (2020). Open access resource for cellular-resolution analyses of corticocortical connectivity in the marmoset monkey. *Nature Communications*, 11(1), 1133.
- Naze, S., et al., (2021). Robustness of connectome harmonics to local gray matter and long-range white matter connectivity changes. *NeuroImage*, 224, 117364.
- Ngo, G. N., et al., (2023). Joint-embeddings reveal functional differences in default-mode network architecture between marmosets and humans. *NeuroImage*, 272, 120035.
- Paxinos, G., et al., (2012). The marmoset brain in stereotaxic coordinates. Elsevier Academic Press.
- Preti, M. G., & Van De Ville, D. (2019). Decoupling of brain function from structure reveals regional behavioral specialization in humans. *Nature communications*, 10(1), 4747.
- Pang, J. C., et al., (2023). Geometric constraints on human brain function. *Nature*, 618, 566-574.
- Patil, K. R., et al., (2023). Commentary on Pang et al.(2023) Nature. *bioRxiv*, Preprint at <https://doi.org/10.1101/2023.10.06.561240>.
- Sevi, H., et al., (2023). Harmonic analysis on directed graphs and applications: From Fourier analysis

- to wavelets. *Applied and Computational Harmonic Analysis*, 62, 390-440.
- Seabrook, E., & Wiskott, L. (2023). A Tutorial on the Spectral Theory of Markov Chains. *Neural Computation*, 35,1713–1796.
- Tian, X., et al., (2022). An integrated resource for functional and structural connectivity of the marmoset brain. *Nature Communications*, 13(1), 7416.
- Tong, C., et al., (2022). Multimodal analysis demonstrating the shaping of functional gradients in the marmoset brain. *Nature Communications*, 13(1), 6584.
- Van Essen, D. C., et al., (2013). The WU-Minn human connectome project: an overview. *Neuroimage*, 80, 62-79.
- Viladomat, J., et al., (2014). Assessing the significance of global and local correlations under spatial autocorrelation: a nonparametric approach. *Biometrics*, 70(2), 409-418.
- Wachinger, C., et al., (2015). BrainPrint: A discriminative characterization of brain morphology. *NeuroImage*, 109, 232-248.
- Ye, C., et al., (2022). Coupling of brain activity and structural network in multiple sclerosis: A graph frequency analysis study. *Journal of Neuroscience Research*, 100(5), 1226-1238.

REVIEWER COMMENTS

Reviewer #1 (Remarks to the Author):

I wish to thank the authors for addressing my comments about their work. I found their revision of outstanding quality.

I think the explanation for deriving the Laplacian from undirected graph is highly beneficial, and I also appreciate the sensitivity analysis of the low vs high frequency cut-off.

My remaining concerns are minor:

1 - Related to R1Q3: The answer in the rebuttal is satisfactory but I think mentioning that the polarity of the eigenmode is arbitrary in the manuscript would also be necessary.

2 - R1Q7: while the response is exhaustive, it does not fully address the question.

A simple subtraction of the temporal mean as described in manuscript and rebuttal does not rescale the signal in amplitude to ensure some channels of higher amplitude do not dominate the analysis (as mentioned in rebuttal). If such mean-centering preprocessing is performed (without rescaling in amplitude), the benefit is that it does not introduce an energy bias at the region level. The drawback is that if some channels have more energy than others, they dominate the analysis and the eigenmodes with strong weights in those regions will show higher energy.

However, if the regional signal is normalized such that all regions have unit variance in amplitude (i.e. via a z-score normalization, which is usual in FC analysis), the opposite bias is observed: regions of low amplitude are artificially scaled up and eigenmodes containing them will have higher energy than they should.

One way to assess which bias is being performed here would be to show the energy (or amplitude) distribution – or mean and std dev – of each region/channel, before normalization. If all channels have similar amplitude (or at least std dev), then no rescaling is needed and no bias is introduced.

3 - Fig.S6: b,c. Middle panels, color code in legend (red line) does not match graph (green line).

Reviewer #2 (Remarks to the Author):

The authors have made substantial efforts to reply to all concerns raised during the review.

However, I feel less and less enthusiastic about the eigenmodes story overall and well beyond the current paper. The additional analyses performed by the authors using null models (shuffled connectivity) clearly show the lack of specificity of the approach, as even a purely randomized connectivity graph can predict functional patterns. Of course, the predictions lag the ones from empirical data but still the gap does not appear that huge.

Also, the relative benefit of using cellular connectome against classic structural connectome is even more weaker, we are talking about few percentage of increased performance. And from a theoretical point of view I am not convinced by the comparison as the underlying technics are different (classical Laplacian vs. Laplacian designed for directed graph). What would happen if you symmetrize the directed CC before computing the classic Laplacian? As a matter of fact, the CC does not appear that much directed by visual inspection of the figures (as it is expected for cortico-cortical projections). Alternatively, although the (classic) Laplacian is fundamentally linked to symmetric graph, nothing preclude to apply it to directed graphs. To complete the investigations, it would also be interesting to see whether human connectivity predict marmoset patterns as well.

Reviewer #3 (Remarks to the Author):

The authors have done a great job in addressing my and the other Reviewers' previous comments/suggestions. The paper is now much more improved; however, I have residual comments/issues to improve the readability of the work.

1. The panel title "Reconstructed activity" for Figs. 2a, 5b, S6b, and S11c is rather misleading. It confuses with the cortical activity represented by the spatiotemporal fMRI

signal, but in reality, the one being measured is a summary statistic of the signal related to its norm over time. Various terms are also used in the manuscript such as energy and activity concentration. I suggest that the authors reconcile the terminologies and ensure that the one being used is an accurate representation of what is being quantified.

2. In one of the null tests, the authors used a randomly shuffled permutation of the BOLD-fMRI time series. I don't think this approach is stringent enough as random shuffling removes the intrinsic spatial autocorrelation within fMRI data.

3. In Figure 3, even though the two datasets lead to significantly correlated CFD patterns, there remain some considerable discrepancies in significant areas especially for the decoupling map. This is interesting and the authors could speak about what's the difference between the two datasets that could be driving the discrepancies (e.g., sample quality, scanner, etc.).

4. The analysis in Figure 4c is somewhat circular. I don't think it's appropriate to correlate CFD and gradient because they were both derived from the CC.

5. Lines 310-314 need to be edited as the message that the authors want to convey is unclear.

6. It is interesting to see that the marmoset's CC and diffusion MRI connectome eigenmodes have a very stark difference in spatial profiles. The authors could discuss this a little bit in the discussion as this is an open-ended question that future works could possibly try to explain. Maybe the authors could also quantitatively show this difference as a supplementary figure, say by plotting a mode-by-mode correlation matrix.

7. There are several typos and grammatical errors throughout the text. One example would be Line 566 (GFT was written as GTT).

Reviewer #3 (Remarks on code availability):

Details of the code are sufficiently provided.

Reviewer #1 (Remarks to the Author):

I wish to thank the authors for addressing my comments about their work. I found their revision of outstanding quality. I think the explanation for deriving the Laplacian from undirected graph is highly beneficial, and I also appreciate the sensitivity analysis of the low vs high frequency cut-off.

We thank you for your feedback and appreciation of our manuscript. We addressed the points you brought up.

My remaining concerns are minor:

Q1: Related to R1Q3: The answer in the rebuttal is satisfactory but I think mentioning that the polarity of the eigenmode is arbitrary in the manuscript would also be necessary.

Response:

Thanks for your suggestions. We have added that the sign (polarity) of the CC eigenmode is pretty arbitrary in the revision as below (P7, Lines 126-127).

“Note that the sign (polarity) of CC eigenmode is arbitrary.”

Q2: R1Q7: while the response is exhaustive, it does not fully address the question.

A simple subtraction of the temporal mean as described in manuscript and rebuttal does not rescale the signal in amplitude to ensure some channels of higher amplitude do not dominate the analysis (as mentioned in rebuttal). If such mean-centering preprocessing is performed (without rescaling in amplitude), the benefit is that it does not introduce an energy bias at the region level. The drawback is that if some channels have more energy than others, they dominate the analysis and the eigenmodes with strong weights in those regions will show higher energy.

However, if the regional signal is normalized such that all regions have unit variance in amplitude (i.e. via a z-score normalization, which is usual in FC analysis), the opposite bias is observed: regions of low amplitude are artificially scaled up and eigenmodes containing them will have higher energy than they should.

One way to assess which bias is being performed here would be to show the energy (or amplitude) distribution – or mean and std dev – of each region/channel, before normalization. If all channels have similar amplitude (or at least std dev), then no rescaling is needed and no bias is introduced.

Response:

Thanks for your helpful advice. We first showed the mean, standard deviation, and activity concentration (L_2 -norm across time points) distribution of regional BOLD-fMRI signals for each marmoset (Fig. S13). We found that the standard deviations of the BOLD-fMRI signals exhibited a nonuniform distribution across regions. To overcome the activity concentration bias at the region level, we normalized the BOLD-fMRI signals by subtracting their temporal mean and dividing them by their temporal standard deviation (i.e., z-score). Z-score normalization ensures that all signals have a consistent scale, essential for fair regional comparisons.

We separated the CC eigenmodes into filters with different frequencies. Next, low- and high-frequency eigenmodes were used to spatially filter the BOLD-fMRI amplitudes for each time point, resulting in low- and high-frequency components. We further performed a nonparametric permutation test to examine the spatial significance of the cortical class-level activity concentrations.

Fig. S13. The mean, standard deviation, and activity concentration distribution of regional BOLD-fMRI signals. Regional activity concentration was defined as the L_2 -norm of BOLD-fMRI signals across all time points.

Cortical activity constrained by low-frequency eigenmodes was concentrated within dorsal-posterior cortices such as the visual (VC), somatosensory (SS), posterior parietal cortex (PPC), and posterior cingulate, medial, and retrosplenic cortices (PCC) (all $P_{SR} < 0.05$, FDR-corrected; Fig. 2c). The high-frequency components were concentrated in ventral-anterior (frontopolar-temporal) cortices, including the medial prefrontal cortex (mPFC), dorsolateral prefrontal cortex (dlPFC), ventrolateral prefrontal cortex (vlPFC), orbitofrontal cortex (OFC), lateral inferior temporal cortex (LIT), and auditory (AU) (all $P_{SR} < 0.05$, FDR-corrected; Fig. 2d). In summary, the patterns of CC eigenmodes constrain cortical activity were circumscribed by specific brain systems.

Fig. 2. c, d The spatial patterns of low- and high-frequency components. The low- and high-frequency components from each of the eleven cortical classes are presented as box plots ordered by the median value. Outliers are shown by the gray circles. Asterisks denote statistically significant activity concentration in each cortical class compared to the null permutation distributions (1000 repetitions, $*P_{SR} < 0.05$, one-tailed, FDR-corrected). AU = Auditory cortex, dlPFC = dorsolateral prefrontal cortex, LIT = lateral and inferior temporal cortex, mPFC = medial prefrontal cortex, MOT = motor and premotor cortex, OFC = orbitofrontal cortex, PCC = posterior cingulate, medial, and retrosplenic cortex, PPC = posterior parietal cortex, SS = Somatosensory cortex, vlPFC = ventrolateral prefrontal cortex, VC = Visual cortex.

Q3: Fig.S6: b,c. Middle panels, color code in legend (red line) does not match graph (green line).

Response:

We have corrected this typo in the revision. The green line indicates the reconstruction accuracy of the empirical SC graph.

Reviewer #2 (Remarks to the Author):

Q1: The authors have made substantial efforts to reply to all concerns raised during the review.

However, I feel less and less enthusiastic about the eigenmodes story overall and well beyond the current paper. The additional analyses performed by the authors using null models (shuffled connectivity) clearly show the lack of specificity of the approach, as even a purely randomized connectivity graph can predict functional patterns. Of course, the predictions lag the ones from empirical data but still the gap does not appears that huge.

Also, the relative benefit of using cellular connectome against classic structural connectome is even more weaker, we are talking about few percentage of increased performance. And from a theoretical point of view I am not convinced by the comparison as the underlying technics are different (classical Laplacian vs. Laplacian designed for directed graph). What would happen if you symmetrize the directed CC before computing the classic Laplacian? As a matter of fact, the CC does not appeared that much directed by visual inspection of the figures (as it is expected for cortico-cortical projections). Alternatively, although the (classic) Laplacian is fundamentally linked to symmetric graph, nothing preclude to apply it to directed graphs. To complete the investigations, it would also be interesting to see whether human connectivity predict marmoset patterns as well.

Response:

We appreciate your thoughtful concerns. First, we investigated the specificity of the basis sets for explaining brain function using two null models: the rewired connectome and the Moran spectral randomization (MSR) surrogate cortical activity (Faskowitz et al., 2023; Pang et al., 2023). We next compared the performance between the eigenmodes of cellular connectome (CC), symmetrized CC, and structural connectome (SC). Lastly, we examined whether human connectivity could predict marmoset activity patterns.

(1) The specificity of CC eigenmodes to explain cortical activity

The first requirement for specificity is that cortical activity should be better described by CC eigenmodes than eigenmodes derived from shuffled connectivity. To this end, we derived eigenmodes from two null models (Rubinov et al., 2010; Váša et al., 2022). In the first null model, Erdős–Rényi (ER) random graphs have the same density as the empirical

CC graph, whereas edges are randomly arranged. In the second null model, rewired graphs preserve the empirical CC graph's weight, degree, and strength distributions.

If all connectome eigenmodes ($N = 55$) are employed, BOLD-fMRI signals can be fully described by eigenmodes obtained from random graphs since the node identities are the same and reflect brain regions like the CC-derived graph. Because the random graph does not accurately reflect the topology of CC, we anticipated that this representation would be less effective.

We found that the first 20 CC eigenmodes achieved 83% reconstruction accuracy of cortical activity concentration. In all ER random graphs, the number of eigenmodes necessary to capture 83% activity concentration exceeds the empirical CC graph with a median number of necessary eigenmodes of 35 (minimum: 31, maximum: 36) (Fig. R1a, left panel). The median number of necessary eigenmodes in all rewired graphs is 33 (minimum: 32, maximum: 35) (Fig. R1a, right panel).

Similarly, the first 20 CC eigenmodes achieved 88% reconstruction accuracy of functional connectivity (FC) in marmosets. In all ER random graphs, the number of eigenmodes that is necessary to capture 88% reconstruction accuracy of FC exceeds the empirical CC graph with a median number of necessary eigenmodes of 50 (minimum: 47, maximum: 53) (Fig. R1b, left panel). The median number of necessary eigenmodes in all rewired graphs is 48 (minimum: 45, maximum: 52) (Fig. R1b, right panel). Therefore, the eigenmodes derived from shuffled connectivity yield poorer reconstruction accuracies in reconstructing cortical activity concentration and FC than CC eigenmodes, demonstrating the specificity of CC eigenmodes. These results are consistent with previous studies (Glomb et al., 2020; Rué-Queralt et al., 2021; Luppi et al., 2023).

Fig. R1. (This figure is just shown in response to reviewers). **a** Cortical activity concentration reconstruction accuracy. Reconstruction accuracy was quantified as the ratio between empirical and reconstructed cortical activity concentration. **b** FC reconstruction accuracy in marmosets (i.e., Pearson correlation coefficients between empirical and reconstructed FC matrices across edges). The red line indicates the reconstruction accuracy of the empirical CC graph. The RdPu (Reds to Purples) shading lines (*left*) indicate the reconstruction accuracy using eigenmodes derived from Erdős–Rényi (ER) random graphs (1000 repetitions). The blue shading lines (*right*) indicate the reconstruction accuracy using eigenmodes derived from rewired CC graphs (1000 repetitions).

The second requirement for specificity is that CC eigenmodes should perform poorly in explaining randomly oriented activity. We used the Moran spectral randomization (MSR) approach (Wagner et al., 2015) to generate the spatially constrained surrogate cortical activity. The accuracy of CC eigenmodes in reconstructing cortical activity concentration and FC is superior to their accuracy in reconstructing the MSR surrogate activity (Fig. 2), thus demonstrating the specificity of CC eigenmodes.

Fig. 2. Reconstruction accuracy of marmoset cortical activity concentration (a) and functional connectivity (b) achieved by CC eigenmodes. The red line indicates the reconstruction accuracy of the empirical CC graph. The green shading lines indicate the reconstruction accuracy of using empirical CC eigenmodes to reconstruct the Moran spectral randomization (MSR) surrogate cortical activity (1000 repetitions). The bottom row corresponds to the P-value, the proportion of times that the empirical reconstruction accuracy exceeded the randomized reconstruction accuracy magnitudes.

(2) The performance of CC, symmetrized CC, and SC eigenmodes

In reconstructing marmosets' cortical activity concentration and FC, we compared the reconstruction accuracy of CC, symmetrized CC, and SC eigenmodes. Similar to previous studies (Pang et al., 2023; Patil et al., 2023), we used a graph signal processing (GSP) framework, but the underlying anatomical eigenmodes differed.

We first constructed a weighted asymmetrical matrix A (55 source areas \times 55 target areas), where A_{ij} denotes the direct connection weight (FLNe value) of source area index i and injected target area index j . This CC matrix has 747 paired bidirectional projections, 360 unidirectional projections (only one direction), and 1100 unconnected edges (Fig. R2). For bidirectional connections, we compared the FLNe weights (i.e., $A_{ij} - A_{ji}$) for each direction to assess a degree of asymmetry (Fig. R3). The CC matrix has 502 bidirectional connections with a degree of asymmetry between -0.03 and 0.03.

The alternative simple method is to symmetrize the weight matrix A to $A_{sym} = \frac{A_{ij} + A_{ji}}{2}$. Then, symmetrized CC eigenmodes were derived from the normalized undirected graph

Laplacian. The fact that the graphs described by A and A_{sym} have radically different structural properties (Seabrook et al., 2023), however, is a significant disadvantage. For instance, there is no guarantee that the stationary distributions of the random walks on these two graphs are similar. Indeed, according to previous studies (Johns et al., 2007; Marques et al., 2020; Seabrook et al., 2023), symmetrizing A dramatically erases structural information from a directed graph.

A more rigorous method is the normalized directed graph Laplacian described in Chung (2005). This Laplace operator is defined as a self-adjoint operator using the transition probability operator and the Perron vector. Notably, applications such as spectral clustering, graph embedding, and classification have used the normalized directed graph Laplacian. These applications may employ the random walk operator to express the directionality and edge density of the graph (Seabrook et al., 2023).

Furthermore, we reconstructed SC matrices using probabilistic diffusion tractography. A group-level SC matrix was generated by averaging all individual SC matrices. Because the connection density of CC and SC differed (the density of CC was 0.6842; the density of SC was 0.9912), we thresholded the SC matrix to construct a connection matrix that matched the density of CC (Fig. R2).

The CC, symmetrized CC, and SC eigenmodes were derived from graph Laplacian (directed or undirected). It is worth noting that the undirected Laplacian is a particular case of the directed Laplacian. Contrasting these different basis sets allows us to distinguish anatomically accurate descriptions of function activity. A direct comparison of the reconstruction accuracy of these distinct basis sets revealed that CC eigenmodes numerically outperform symmetrized CC and SC eigenmodes in capturing cortical activity concentration (Fig. R4a) and FC (Fig. R4b). Hence, CC eigenmodes could provide a more compact description of cortical activity than symmetrized CC and SC eigenmodes.

As neural tracing provides information on projection directionality, CC has historically been able to shape brain function more effectively than SC (Knock., 2009; Melozzi., 2019).

Fig. R2. (This figure is just shown in response to reviewers). **Marmoset cellular connectome (CC) (left), symmetrized CC (middle), and structural connectome (SC) (right) matrix.**

Fig. R3. (This figure is just shown in response to reviewers). **Asymmetry of bidirectional connection in CC matrix.**

Fig. R4. (This figure is just shown in response to reviewers). **The performance of CC, symmetrized CC, and SC eigenmodes in reconstructing marmosets' cortical activity concentration and FC.**

(3) Using human connectivity to predict marmoset activity patterns

We next attempted to use the human SC eigenmodes to capture intrinsic brain activity and FC in the marmoset brain. We included diffusion MRI datasets from the 100 unrelated subjects ($n = 100$, 54 females, 22-36 years) provided by the Human Connectome Project (HCP) (<https://db.humanconnectome.org/>) (Van Essen et al., 2013). The probabilistic

diffusion tractography was used to reconstruct SC matrices based on the HCP-MMP1.0 parcellation (Glasser et al., 2016). Then, we extracted a group-level SC matrix of 11 homologous areas. Finally, we reconstructed marmoset cortical activity at each time point via human SC eigenmodes, and further generated an area-to-area FC matrix.

Not surprisingly, marmoset activity patterns can be expressed ideally using all human SC eigenmodes of homologous areas ($N = 11$). The first four marmoset CC eigenmodes outperformed human SC eigenmodes, with reconstruction accuracy of activity concentration and FC reaching 83% (AUC=0.55) and 87% (AUC=0.43) of marmosets' cortical activity concentration and FC, respectively (Fig. R5).

Fig. R5. (This figure is just shown in response to reviewers). **Using human connectivity to predict marmoset activity patterns.** **a** Marmoset cellular connectome (CC) and human structural connectome (SC) of 11 homologous areas. The performance of marmoset CC and human SC eigenmodes in reconstructing marmosets' cortical activity concentration (**b**) and FC (**c**). Asterisks denote statistically significant reconstruction accuracy compared to the null permutation distributions (1000 repetitions, $*P < 0.05$, one-tailed, FDR-corrected).

Bibliographies in the Response to Reviewer #2:

- Faskowitz, J., et al., (2023). Commentary on Pang et al. (2023) *Nature*. *bioRxiv*, Preprint at <https://doi.org/10.1101/2023.07.20.549785>.
- Glasser, M. F., et al., (2016). A multi-modal parcellation of human cerebral cortex. *Nature*, 536(7615), 171-178.
- Glomb, K. et al. (2020). Connectome spectral analysis to track EEG task dynamics on a subsecond scale. *NeuroImage*, 221, 117137.
- Johns, J., & Mahadevan, S. (2007). Constructing basis functions from directed graphs for value function approximation. In *Proceedings of the 24th international conference on Machine learning* (pp. 385-392).
- Knock, S. A. et al. (2009). The effects of physiologically plausible connectivity structure on local and global dynamics in large scale brain models. *Journal of Neuroscience Methods*, 183(1), 86-94.
- Luppi, A. I. et al. (2023). Distributed harmonic patterns of structure-function dependence orchestrate human consciousness. *Communications biology*, 6(1), 117.
- Marques, A. G. et al. (2020). Signal processing on directed graphs: The role of edge directionality when processing and learning from network data. *IEEE Signal Processing Magazine*, 37(6), 99-116.
- Melozzi, F. et al. (2019). Individual structural features constrain the mouse functional connectome. *Proceedings of the National Academy of Sciences*, 116(52), 26961-26969.
- Patil, K. R., Jung, K., & Eickhoff, S. B. (2023). Commentary on Pang et al. (2023) *Nature*. *bioRxiv*, Preprint at <https://doi.org/10.1101/2023.10.06.561240>.
- Pang, J. C., et al., (2023). Geometric constraints on human brain function. *Nature*, 618, 566-574.
- Pang, J. C., et al., (2023). Reply to: Commentary on Pang et al. (2023) *Nature*. *bioRxiv*, Preprint at <https://doi.org/10.1101/2023.10.06.560797>.
- Rué-Queralt, J. et al. (2021). The connectome spectrum as a canonical basis for a sparse representation of fast brain activity. *NeuroImage*, 244, 118611.
- Rubinov, M., & Sporns, O. (2010). Complex network measures of brain connectivity: uses and interpretations. *Neuroimage*, 52(3), 1059-1069.
- Seabrook, E., & Wiskott, L. (2023). A Tutorial on the Spectral Theory of Markov Chains. *Neural Computation*, 35(11), 1713-1796.
- Van Essen, D. C., et al., (2013). The WU-Minn human connectome project: an overview. *Neuroimage*, 80, 62-79.
- Váša, F., & Mišić, B. (2022). Null models in network neuroscience. *Nature Reviews Neuroscience*, 23(8), 493-504.
- Wagner, H. H., & Dray, S. (2015). Generating spatially constrained null models for irregularly spaced data using Moran spectral randomization methods. *Methods in Ecology and Evolution*, 6(10), 1169-1178.

Reviewer #3 (Remarks to the Author):

The authors have done a great job in addressing my and the other Reviewers' previous comments/suggestions. The paper is now much more improved; however, I have residual comments/issues to improve the readability of the work.

We greatly appreciate your insightful comments on our manuscript. We addressed the points you brought up.

Q1: The panel title "Reconstructed activity" for Figs. 2a, 5b, S6b, and S11c is rather misleading. It confuses with the cortical activity represented by the spatiotemporal fMRI signal, but in reality, the one being measured is a summary statistic of the signal related to its norm over time. Various terms are also used in the manuscript such as energy and activity concentration. I suggest that the authors reconcile the terminologies and ensure that the one being used is an accurate representation of what is being quantified.

Response:

We have modified the panel title "Reconstructed activity" to "Reconstructed activity concentration" for Figs. 2a, 5b, S6b, and S12c.

We defined the L_2 -norm of BOLD-fMRI signal $f_{v_i}(t)$ across all time points to measure the cortical activity concentration of the graph frequency component at the brain area v_i : $E_{v_i} = \|f_{v_i}(t)\|_2$. We used the L_2 -norm because it interprets energy for each graph frequency component (Medaglia et al., 2018; Huang et al., 2018; Preti et al., 2019). As a result, the term "activity concentration" was used in the revision.

Q2: In one of the null tests, the authors used a randomly shuffled permutation of the BOLD-fMRI time series. I don't think this approach is stringent enough as random shuffling removes the intrinsic spatial autocorrelation within fMRI data.

Response:

We used the Moran spectral randomization (MSR) approach (Wagner et al., 2015) to generate the spatially constrained surrogate cortical activity. MSR principally relies on an inverse of the distance matrix W between brain regions. The eigenvectors of W provide an estimate of the autocorrelation in the brain and are used to impose a similar spatial structure on random, normally distributed surrogate data.

In the revision, we performed a spatial permutation test using the MSR method to generate spatially constrained null models for fMRI data. The MSR surrogate cortical activity was performed using the singleton procedure implemented in BrainSpace (Vos de Wael et al., 2020) (<http://github.com/MICA-MNI/BrainSpace>). The revision has been amended as follows:

Page 6 in the Supplementary Information:

“**Moran spectral randomization.** The spatially constrained surrogate cortical activity was generated by the Moran spectral randomization (MSR) approach¹⁵. This method was initialized with eigenvectors of inverse distance matrix between brain regions. The surrogate cortical activity was performed using the singleton procedure implemented in BrainSpace¹⁶ (<http://github.com/MICA-MNI/BrainSpace>). Then, the surrogate cortical activity was projected onto eigenmodes of the original connectome to reconstruct the activity patterns.”

Fig. 2. Reconstruction accuracy of marmoset cortical activity concentration (a) and functional connectivity (b) achieved by cellular connectome (CC) eigenmodes. The red line indicates the reconstruction accuracy of the empirical CC graph. The green shading lines indicate the reconstruction accuracy using empirical CC eigenmodes to reconstruct the Moran spectral randomization (MSR) surrogate cortical activity (1000 repetitions).

Q3: In Figure 3, even though the two datasets lead to significantly correlated CFD patterns, there remain some considerable discrepancies in significant areas especially for the decoupling map. This is interesting and the authors could speak about what’s the difference between the two datasets that could be driving the discrepancies (e.g., sample quality, scanner, etc.).

Response:

We conducted a spatial correlation of the CFD pattern between the NIH and ION site datasets. The results showed a statistically significant positive correlation ($\rho = 0.83$, $P_{\text{SMASH}} < 0.001$). Furthermore, the Dice coefficient was 0.73 and 0.70 for significantly decoupled and coupled areas, respectively. The differences in the scanner magnetic fields and the scanning parameters of the two datasets may drive some discrepancies.

The data acquisition procedure from both centers followed the same animal training protocol, 8-element radiofrequency coil, and MRI scanning protocols. Awake marmosets were scanned in horizontal MRI scanners (ION, 9.4 T/30 cm; NIH, 7 T/30 cm). rsfMRI data were collected using 2D gradient echo EPI sequence with the following parameters: TR = 2000 ms, TE = 18 ms (ION) or 22.2 ms (NIH), flip angle = 70.4° , FOV = 28×36 mm, matrix size = 56×72 , 38 axial slices, slice thickness = 0.5 mm, 512 volumes per scan. Two sets of spin-echo EPI with opposite phase-encoding directions (LR and RL) were also collected for EPI-distortion correction with the following parameters: TR = 3000 ms, TE = 37.69 ms (ION) or 36 ms (NIH), flip angle = 90° , FOV = 28×36 mm, matrix size = 56×72 , 38 axial slices, slice thickness = 0.5 mm, 8 volumes for each set. Furthermore, for each rsfMRI session, a T2-weighted structural image was scanned for co-registration purposes with the following parameters: TR = 8000 ms (ION) or 6000 ms (NIH), TE = 10 ms (ION) or 9 ms (NIH), flip angle = 90° , FOV = 28×36 mm, matrix size = 112×144 , 38 axial slices, slice thickness = 0.5 mm. In addition, rsfMRI collection for the marmosets concerning the NIH dataset (all males) and the ION cohort (12 males vs. 1 female) was sex-biased.

Furthermore, we evaluated the data quality across two datasets. The image quality metrics for the temporal information included head motion and temporal Signal-to-Noise Ratio (tSNR) (Fig. R6). We found that head-motion levels are similar across datasets (two-sample test, $P=0.17$). We compared tSNR across the sites/scanners, which shows no significant differences between the two datasets ($P=0.37$). Thus, the NIH and ION datasets have no significant difference in the above quality assurance measurements.

We have added the following text to the main text:

Page 13 in the main text (Lines 235-236):

“The differences in the scanner magnetic fields and the scanning parameters of the two datasets may drive some discrepancies.”

Page 22 in the main text (Lines 429-431):

“Fifth, BOLD-fMRI collection for the marmosets concerning the discovery dataset (all males) and replication cohort (12 males vs. 1 female) was sex-biased.”

Fig. R6. (This figure is just shown in response to reviewers). **Similar quality measurements of the ION and the NIH datasets.** (*left*) Head motions of the NIH and ION datasets. The boxplot presents average head motion (weighted Euclidean norm of six motion parameters) across time points of two datasets, which found no significant differences between the two datasets. (*right*) Temporal Signal-to-Noise Ratio (tSNR) of the NIH and ION datasets. The tSNR values were calculated by the mean value of gray matter voxels divided by the standard deviation of background noises.

Q4: The analysis in Figure 4c is somewhat circular. I don't think it's appropriate to correlate CFD and gradient because they were both derived from the CC.

Response:

Thanks for your idea. We have removed Figure 4c in the revision as below:

Fig. 4. Marmoset cellular-functional decoupling (CFD) is related to microstructures and macroscale hierarchical organization. **a, b** The pattern of regional CFD negatively correlates with microstructures, including marmoset myelin content (T1w/T2w ratio) and neuronal counts. **c** Patterned CFD positively correlates with the second functional gradient (FG2). The significance of correlation coefficients (P_{SMASH}) is evaluated using 1000 spatial autocorrelation-preserving surrogate brain maps.

Q5: Lines 310-314 need to be edited as the message that the authors want to convey is unclear.

Response:

Page 17 in the main text (Lines 313-316):

We revised these sentences to “Extended previous structural-functional coupling focused on a single “summary process”^{8,10,11}, we considered these couplings contribute to multiple repertoires through eigenmode decomposition^{12,14}. Eigenmodes provide a powerful framework for connecting brain anatomy with the spatiotemporal patterns of neural dynamics.”

Q6: It is interesting to see that the marmoset's CC and diffusion MRI connectome eigenmodes have a very stark difference in spatial profiles. The authors could discuss this a little bit in the discussion as this is an open-ended question that future works could possibly try to explain. Maybe the authors could also quantitatively show this difference as a supplementary figure, say by plotting a mode-by-mode correlation matrix.

Response:

The first CC eigenmode was uniformly distributed throughout the brain. The second one reflected a dorsal-ventral dimension. The third CC eigenmode showed the dimension between the anterior cingulate and other brain areas. The fourth CC eigenmode represented a gradient axis from the sensorimotor cortex to the visual cortex, resembling the marmosets' principal structural gradient. With increasing eigenvalue (or frequency), the irregularity and localization of CC eigenmode patterns increased (Fig. S4c).

Despite certain similarities, the spatial patterns of CC eigenmodes were distinct from SC eigenmodes in marmosets (Fig. S4c and Fig. S6a). Hence, we calculated the Pearson correlation to quantify the spatial pattern difference between CC and SC eigenmodes (Fig. S7). The first SC eigenmode was uniformly distributed across the brain. The second SC eigenmode reflected a dorsal-ventral dimension, with the highest correlation coefficient with the second CC eigenmode ($r = 0.5763$, $P_{\text{SMASH}} < 0.001$). The third SC eigenmode showed the dimension between the sensorimotor areas and other brain areas, with the highest correlation coefficient with the third CC eigenmode ($r = -0.7358$, $P_{\text{SMASH}} < 0.001$). The fourth SC eigenmode has the highest correlation coefficient with the second CC eigenmode ($r = -0.5609$, $P_{\text{SMASH}} < 0.001$).

The reasons that the spatial patterns of CC and SC eigenmodes are different may be as follows:

1) The techniques for producing the connectome are different. The CC eigenmodes rely on cellular-level retrograde neuroanatomical tract-tracing connectome. The SC eigenmodes rely on short- and long-range white matter fiber connections reconstructed from dMRI.

2) The methods used to derive eigenmodes are different. The CC eigenmodes produced from Laplacian's normalized directed graph may capture the projection directionality and edge density, emphasizing the importance of projection directionality information. The SC eigenmodes obtained from the undirected graph Laplacian of dMRI-based networks

capture local gray matter and white matter fiber connections.

As a result, different methods of generating anatomical connections and deriving eigenmodes may lead to different spatial patterns of CC and SC eigenmodes. We have added the following text to the Discussion section (P18, Lines 329-331).

“Different methods of generating anatomical connections and deriving eigenmodes may lead to different spatial patterns of CC and SC eigenmodes.”

Fig. S4c and Fig. S6a. The spatial patterns of CC eigenmodes and SC eigenmodes.

Fig. S7. The correlation matrix between cellular connectome (CC) eigenmodes and structural connectome (SC) eigenmodes. We calculated the Pearson correlation to quantify the spatial pattern difference between CC and SC eigenmodes. Bluer and redder colors represent CC and SC eigenmodes with similar spatial distribution.

Q7: There are several typos and grammatical errors throughout the text. One example would be Line 566 (GFT was written as GTT).

Response:

We have carefully checked and revised typos and grammatical errors throughout the manuscript.

Reviewer #3 (Remarks on code availability):

Details of the code are sufficiently provided.

Response:

We thank you for your positive appraisal of our work.

Bibliographies in the Response to Reviewer #3:

- Huang, W., et al., (2018). A graph signal processing perspective on functional brain imaging. *Proceedings of the IEEE*, 106(5), 868-885.
- Medaglia, J. D., et al., (2018). Functional alignment with anatomical networks is associated with cognitive flexibility. *Nature Human Behaviour*, 2(2), 156-164.
- Preti, M. G., & Van De Ville, D. (2019). Decoupling of brain function from structure reveals regional behavioral specialization in humans. *Nature Communications*, 10(1), 4747.
- Vos de Wael, R., et al. (2020). BrainSpace: a toolbox for the analysis of macroscale gradients in neuroimaging and connectomics datasets. *Communications Biology*, 3(1), 103.
- Wagner, H. H., & Dray, S. (2015). Generating spatially constrained null models for irregularly spaced data using Moran spectral randomization methods. *Methods in Ecology and Evolution*, 6(10), 1169-1178.

REVIEWERS' COMMENTS

Reviewer #1 (Remarks to the Author):

Thank you for addressing my previous comment about arbitrary polarity.

On Q2 related to the normalization of BOLD timeseries, unfortunately it seems that my concern has not been understood. I don't think the new Supplementary Figure S13 adds much useful information because it reports mean, std and energy concentration at subject level while I was referring to region level. I don't think reporting this S13 is necessary but that's to the authors to decide.

However, the authors have now clarified that the BOLD timeseries are z-score normalized prior to conducting the GSP analysis. As explained previously, this introduces an energy bias (or distortion) when then comparing the activity concentration in low and high graph frequencies.

Despite that this maybe usual practice in GSP analysis, I think it is worth mentioning.

Reviewer #1 (Remarks on code availability):

The code is available on github and is decently documented, while still far from professional software engineering standards. The presence of a demo that runs on a subset of the whole dataset is appreciable and shows the effort by the authors to make to analysis reproducible by others.

I parsed thoroughly the functions related to the GSP analysis and find it is accordance with the reported results and Methods, without detectable mistakes.

Running the code would require licensing to third party software (e.g. Matlab) and possible dependencies. Also some path in the code are hard-coded so it would likely require modifications and debugging (which is not uncommon), therefore I have not tested it.

Reviewer #2 (Remarks to the Author):

I would like to thank the authors for addressing all concerns raised during the review. I find the paper strengthened, even if it does not change my skepticism about the specificity of such approach.

Reviewer #3 (Remarks to the Author):

The authors have adequately addressed my previous comments and I support the acceptance of the paper.

Reviewer #1 (Remarks to the Author):

Q1: Thank you for addressing my previous comment about arbitrary polarity.

On Q2 related to the normalization of BOLD timeseries, unfortunately it seems that my concern has not been understood. I don't think the new Supplementary Figure S13 adds much useful information because it reports mean, std and energy concentration at subject level while I was referring to region level. I don't think reporting this S13 is necessary but that's to the authors to decide.

However, the authors have now clarified that the BOLD timeseries are z-score normalized prior to conducting the GSP analysis. As explained previously, this introduces an energy bias (or distortion) when then comparing the activity concentration in low and high graph frequencies.

Despite that this maybe usual practice in GSP analysis, I think it is worth mentioning.

Response:

Thanks for your helpful advice. At the region level, we first showed the temporal mean, standard deviation, and activity concentration (L_2 -norm across time points) distribution of regional BOLD-fMRI signals for each marmoset (Supplementary Fig. 13). The standard deviations of the BOLD-fMRI signals exhibited a nonuniform distribution across regions. To overcome the activity concentration bias at the region level, we normalized the BOLD-fMRI signals by subtracting their temporal mean and dividing them by their temporal standard deviation (i.e., z-score). Note that if all regions have similar amplitudes (or at least standard deviation), normalization is unnecessary to avoid bias when comparing the activity concentration in low and high graph frequencies.

We have added the following text to the main text:

Page 19 in the main text (Lines 430-432):

“Note that normalization is unnecessary to avoid introducing bias if the signals in all regions have similar amplitudes (or at least standard deviation).”

Reviewer #1 (Remarks on code availability):

Q2: The code is available on github and is decently documented, while still far from professional software engineering standards. The presence of a demo that runs on a subset of the whole dataset is appreciable and shows the effort by the authors to make to analysis reproducible by others.

I parsed thoroughly the functions related to the GSP analysis and find it is accordance with the reported results and Methods, without detectable mistakes.

Running the code would require licensing to third party software (e.g. Matlab) and possible dependencies. Also some path in the code are hard-coded so it would likely require modifications and debugging (which is not uncommon), therefore I have not tested it.

Response:

We thank the reviewer for all the positive comments. We optimized the code and changed the file read/write path to a relative one. Details of the code are sufficiently provided. The code used to conduct the main results in this study is available at https://github.com/weiliao81/Marmoset_CFD and on Zenodo (<https://doi.org/10.5281/zenodo.10728317>).

Reviewer #2 (Remarks to the Author):

I would like to thank the authors for addressing all concerns raised during the review. I find the paper strengthened, even if it does not change my skepticism about the specificity of such approach.

Response:

We thank you for your positive comments of our manuscript.

Reviewer #3 (Remarks to the Author):

The authors have adequately addressed my previous comments and I support the acceptance of the paper.

Response:

We thank you for your positive appraisal of our work.